# Ecological ReGional Ocean Model with vertically resolved sediments (ERGOM SED 1.0): Coupling benthic and pelagic biogeochemistry of the south-western Baltic Sea

Hagen Radtke[1], Marko Lipka[2], Dennis Bunke[3,4], Claudia Morys[5,6], Jana Woelfel[7], Bronwyn Cahill[1,8], Michael E. Böttcher[2], Stefan Forster[5], Thomas Leipe[9], Gregor Rehder[7], and Thomas Neumann[1]

[1]Leibniz Institute for Baltic Sea Research Warnemuende (IOW), Department of Physical Oceanography and Instrumentation, Seestr. 15, 18119 Warnemünde, Germany

[2]Leibniz Institute for Baltic Sea Research Warnemuende (IOW), Department of Marine Geology, Geochemistry and Isotope Biogeochemistry Group, Seestr. 15, 18119 Warnemünde, Germany

[3]Leibniz Institute for Baltic Sea Research Warnemuende (IOW), Department of Marine Geology, Paleooceanography and Sedimentology Group, Seestr. 15, 18119 Warnemünde, Germany

[4]Current address: Leipzig University, Institute of Geophysics and Geology, Talstr. 15, 04013 Leipzig, Germany

[5]University of Rostock, Institute for Biosciences, Albert-Einstein-Str. 3, 18059 Rostock, Germany

[6]Current address: Royal Netherlands Institute for Sea Research (NIOZ), Department of Estuarine and Delta Systems, and Utrecht University, Korringaweg 7, 4401 NT Yerseke, The Netherlands

[7]Leibniz Institute for Baltic Sea Research Warnemuende (IOW), Department of Marine Chemistry, Working group on Trace Gas Biogeochemistry, Seestr. 15, 18119 Warnemünde, Germany

[8]Current address: Freie Universität Berlin, Institute for Space Science, Carl-Heinrich-Becker-Weg 6-10, 12165 Berlin, Germany

[9]Leibniz Institute for Baltic Sea Research Warnemuende (IOW), Department of Marine Geology, Microanalysis Group, Seestr. 15, 18119 Warnemünde, Germany

*Correspondence to:* Hagen Radtke (hagen.radtke@io-warnemuende.de)

**Abstract.** Sediments play an important role in organic matter mineralisation and nutrient recycling, especially in shallow marine systems. Marine ecosystem models, however, often only include a coarse representation of processes beneath the sea floor. While these parametrisations may give a reasonable description of the present ecosystem state, they lack predictive capacity for possible future changes, which can only be obtained from mechanistic modelling.

5    This paper describes an integrated benthic-pelagic ecosystem model developed for the German Exclusive Economic Zone (EEZ) in the Western Baltic Sea. The model is a hybrid of two existing models: the pelagic part of the marine ecosystem model ERGOM and an early diagenetic model by Reed et al. (2011). The latter one was extended to include the carbon cycle, a determination of precipitation and dissolution reactions which accounts for salinity differences, an explicit description of adsorption of clay minerals and an alternative pyrite formation pathway. We present a one-dimensional application of the

10   model to seven sites with different sediment types. The model was calibrated with observed pore water profiles and validated with results of sediment composition, bioturbation rates and bentho-pelagic fluxes gathered by in situ incubations of sediments (benthic chambers). The model results generally give a reasonable fit to the observations, even if some deviations are observed, e.g. an overestimation of sulphide concentrations in the sandy sediments. We therefore consider it a good first step towards a three-dimensional representation of sedimentary processes in coupled pelagic-benthic ecosystem models of the Baltic Sea.

# 1 Introduction

## 1.1 Importance of the bentho-pelagic coupling

Shallow coastal waters are the most dynamic part of the ocean due to the various effects of natural forcing and anthropogenic activities, they are characterised by the processing and accumulation of land-derived discharges (nutrients, pollutants etc.) which influence not only the coastal ecosystem but also the adjacent deeper sea areas. Shallow marine ecosystems often differ significantly from those in the deeper parts of the sea (Levinton, 2013). One important reason for this is the influence of sedimentary processes on the pelagic ecosystem. This influence can take place in a number of different functional ways, including:

- Remineralisation of organic matter produced in the water column fuels the subsequent release of nutrients and enhances the productivity of these regions (Berner, 1980).

- At the same time, nutrients can be buried in the sediment in a particulate form (Sundby et al., 1992) or be removed by denitrification (Seitzinger et al., 1984).

- Sulphate reduction in the sediments may lead to a release of toxic hydrogen sulphide (Hansen et al., 1978).

- Benthic biomass and primary production of benthic microalgae exceeds that of the phytoplankton in the overlying waters (Glud et al., 2009; Pinckney and Zingmark, 1993; Colijn and De Jonge, 1984) and represents a major food source for benthic organisms (Cahoon et al., 1999). In shallow regions, benthic primary production oxygenates the water column and competes with the pelagic one for nutrients (Cadée and Hegeman, 1974).

- Sediments serve as habitats for the zoobenthos, thereby affecting the overlying waters mainly via bioturbation or filtration (Gili and Coma, 1998).

- Other benthic organisms are food for opportunistic benthic/pelagic predator species, whose presence influences the pelagic system as well (Rudstam et al., 1994).

- Organisms typically inhabiting the pelagic may have benthic life stages and therefore rely on sediment properties for reproduction (Marcus, 1998).

This list, which could be continued, illustrates the importance of bentho-pelagic coupling for the functioning of shallow marine ecosystems.

## 1.2 Mechanistic sediment representation

In spite of this importance, the representation of the sediments in marine ecosystem models is often strongly oversimplified. This is understandable, since these models are constructed to answer specific research questions, and if these focus on pelagic processes, it can be desirable to represent sediment functions by the simplest-as-possible parametrisations. The drawback of

using simple parametrisations is that they are mostly obtained from the present-day state. An example for such a parametrisation could be a percentage of organic matter which is remineralised in the sediments after its deposition and returned to the water column as nutrients. When ecosystem models are used not only to understand the present, but also to estimate future ecosystem changes in response to external drivers, this causes a problem: the use of such simple parametrisations means an implicit no-change assumption. In other words, the quantitative relationships described by the parametrisation will remain unchanged in future conditions, e.g. after the construction of a fish farm or in a changing climate. It is not straightforward to estimate the error introduced into the model system if this assumption is not valid.

An alternative to empirical parametrisations is the use of mechanistic models which try to derive the functionality of the subsystem from process understanding. For nutrient recycling in the sediments, this could be an early diagenetic model which estimates the final nutrient fluxes from a set of individual diagenetic processes.

Our aim is to construct a three-dimensional fully coupled model of pelagic and sediment biogeochemistry which does not make the no-change assumption. Specifically, we want to understand:

– How changes in early diagenetic processes affect the reaction of a shallow marine ecosystem to climate change?

– If pelagic ecosystem modelling can provide realistic deposition of particulate organic matter to reproduce the local variability in early diagenetic processes?

In this paper, we report about first successful approaches of this goal: the construction of a combined benthic-pelagic biogeo-chemical model formulated in a one-dimensional, vertically resolved domain. The model is calibrated and applied to a specific area of interest, the south-western Baltic Sea. It provides the basis for the development of a three-dimensional framework.

### 1.3 Combining models of sedimentary and pelagic biogeochemistry

Marine biogeochemical models and process-resolving sediment models are very similar to each other in terms of their approach. They both try to describe a complex biogeochemical system with a limited set of state variables. Transformation processes are formulated as a parallel set of differential equations (e.g., van Cappellen and Wang, 1996). These have to obey the principle of mass conservation for any chemical element whose cycle is part of the model system. But in spite of these similarities, and even though both types of models have been extensively applied at least since the 1990s, there have not been many attempts, at least published ones, to combine them into one single benthic-pelagic model system. The review of Paraska et al. (2014), which compares existing sediment model studies, lists 83 publications of which 10 include a coupling to the water column.

In the simplest case, this coupling is only one-way: water column biogeochemistry is calculated first and then used as input for a sediment model. This type of models has e.g. been applied to the North Sea (Luff and Moll, 2004) and Lake Washington (Cerco et al., 2006). In these studies, full three-dimensional models were used for pelagic biogeochemistry investigations. The models aimed to explain regional patterns in sediment biogeochemistry.

To the best of our knowledge, the first fully coupled benthic-pelagic model system with vertically resolved benthic processes was published by Soetaert et al. (2001). They presented a modelling approach where the biogeochemistry of the Goban Spur shelf ecosystem (north-east Atlantic) was described in a horizontally integrated, one-dimensional model. In the present commu-

nication we present a similar approach, adapted to understand the role of the sediments for the ecosystem of the south-western Baltic Sea.

A number of fully coupled benthic-pelagic models have been published for different regions, each differing in the way the compartments are vertically resolved. In our study, we use several fixed-depth vertical layers both in the water column and in the sediment (Soetaert et al., 2001; Soetaert and Middelburg, 2009; Meire et al., 2013). Other studies use a two-layer sediment, where the boundary between the layers is defined by the oxic-anoxic transition rather than a fixed depth (Lee et al., 2002; Lancelot et al., 2005). The opposite is true in the model of Reed et al. (2011), where the water column is resolved with two layers only, while the sediment processes which are clearly the focus of the study, are resolved on a fine vertical grid. These one-dimensional model studies also differ in the complexity of the biogeochemical reactions involved. One of the most complex early diagenetic models was recently published by Yakushev et al. (2017). This is integrated into the Framework for Aquatic Biogeochemical Models (FABM, www.fabm.net). This generic interface allows coupling to any biogeochemical model within its framework, from one-dimensional setups (as we described before) to three-dimensional applications. Our one-dimensional approach presented here can also be seen as an intermediate step towards a fully coupled three-dimensional ecosystem model, with a vertically resolved sediment model coupled under each grid cell. The way to go from the current model to the 3-d version is already pointed out in the model description.

There are a few successful regional applications of three-dimensional setups with coupled water column and sediment biogeochemistry. Sohma et al. (2008) present such a model for Tokyo Bay, where they use it to explain the occurrence of hypoxia and to understand the carbon cycle in the bay (Sohma et al., 2018). Brigolin et al. (2011) developed a fully coupled 3-d model for the Adriatic Sea and use it to estimate the seasonal variability of N and P fluxes. The ERSEM model (Butenschön et al., 2016) is another example of two-way coupling of complex benthic and pelagic biogeochemical models, which treats sediments in a different way: Here, they are vertically resolved into three different layers (oxic, anoxic, sulphidic), and the pore water exchange between them follows a near-steady-state assumption. Another recent example is a Black Sea study by Capet et al. (2016), in which the authors apply a hybrid approach with a vertically integrated early diagenetic model. The partitioning between different oxidation pathways, typically determined by the vertical zonation, is obtained by running a one-dimensional, vertically resolved model (OMEXDIA (Soetaert et al., 1996a)) over a range of different boundary values and fitting a statistical meta-model through its output.

Our region of interest is the Baltic Sea, in the first instance, its south-western part where coastal marine sediments play an important role in the transformation and removal of nutrients from the water column. We combine two existing models which have already been successfully applied in the Baltic Sea, namely, the pelagic ecosystem model ERGOM (Neumann et al., 2017) and the early diagenetic model by Reed et al. (2011), to obtain a full benthic-pelagic model of the southwestern Baltic Sea. In the latter, several modifications were implemented as will be described.

## 1.4 The German part of the Baltic Sea and the SECOS project

The Baltic Sea is a marginal sea with only narrow and shallow connections to the adjacent North Sea. The small cross sections of these channels, the Danish Straits, and the correspondingly constrained water exchange have several implications for the Baltic Sea system:

- It is essentially a non-tidal sea.

- It is brackish due to mixing between episodically inflowing North Sea water with Baltic river waters which causes an overall positive freshwater balance.

- It shows a pronounced haline stratification.

- It is prone to eutrophication due to the accumulation of mostly river-derived nutrients.

The German Exclusive Economic Zone (EEZ) in the Baltic Sea is situated to the south of the Danish Straits. It consists of different bights, islands and peninsulas and exhibits a strong zonal gradient and a strong temporal variability in salinity. This varies from 12 to over 20 g kg$^{-1}$ north of the Fehmarn island to 7 to 9 g kg$^{-1}$ in the Arkona Sea (IOW, 2017). Even lower salinities occur in river-influenced near-coastal areas. Most of the sediment area is characterised by erosion or transport bottoms which only intermittently store deposited material before it is transported further into the central basins of the Baltic Sea (Emeis

et al., 2002). Still, during this storage period, organic material is already partly mineralised and inorganic nitrogen is partly removed from the ecosystem by denitrification processes (Deutsch et al., 2010). This transformation of a bioavailable substance into a non-reactive form and its subsequent removal is one example of the type of ecosystem services (e.g., Haines-Young and Potschin, 2013) that coastal sediments can perform.

Understanding and quantifying the scope and scale of such sedimentary services in the German Baltic Sea has been the

aim of the SECOS project (The Service of Sediments in German Coastal Seas, 2013 - 2019). The project contained a strong empirical part, including several interdisciplinary research cruises focused on sediments characterisation. Seven study sites were selected, based on different granulometric parameters, each of them representative of a larger area. These were sampled several times in order to capture the effect of seasonality on biogeochemical functioning (see Figure 1). The sampled stations include three sandy sites: Stoltera (ST), Darss Sill (DS) and Oder Bank (OB), three mud sites: Lübeck Bight (LB), Mecklenburg

Bight (MB) and Arkona Basin (AB) and a silty site: Tromper Wiek (TW). The TW site has both an intermediate grain size and an intermediate organic matter content, compared to the sandy and muddy sites. In this work, we focus on the development of our coupled one-dimensional benthic-pelagic model system for the German Baltic Sea. We use empirical data obtained from repeated sampling of the SECOS stations to calibrate and validate our early diagenetic model. Further work, discussing the fully coupled three dimensional application of the model to assessing sedimentary services in the German Baltic Sea will be

described in a forthcoming paper.

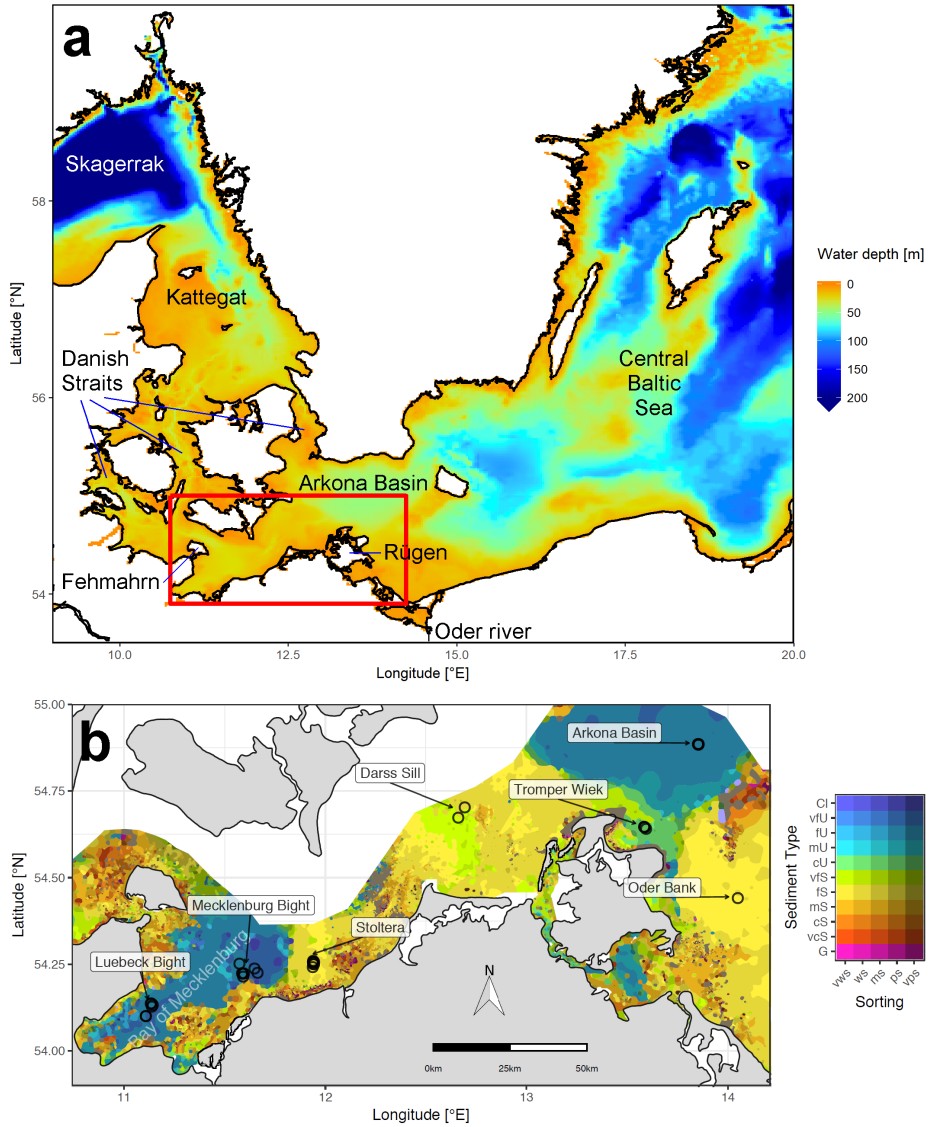

**Figure 1.** (a) Bathymetry of the Western Baltic Sea and location of our area of interest. (b) The investigation area of the SECOS project. The map shows granulometry, redrawn from Tauber (2012) and Lipka (2018), and the seven stations considered in this model study. Sediment Type: Cl = clay, vfU = very fine silt, fU = fine silt, mU = medium silt, cU = coarse silt, vfS = very fine sand, fS = fine sand, mS = medium sand, cS = coarse sand, vcS = very coarse sand, G = gravel; Sorting: vws = very well sorted, ws = well sorted, ms = moderately sorted, ps = poorly sorted, vps = very poorly sorted.

## 1.5 Differences in biogeochemistry between permeable and impermeable sediments

In the study area, different types of sediments dominated by varying grain size fractions are found ranging from sand to mud. This implies differences in the biogeochemical processes associated with organic matter mineralization and physical processes that are responsible for pore water and elemental transport in the sediment and across the sediment-water interface. Due to their relatively larger grain sizes, sand acts as a permeable substrate, which means that lateral pressure variations may induce advection of interstitial water. These pressure variations may be caused by waves or by the interaction between horizontal near-bottom currents and ripple formation. In muddy sediments, in contrast, molecular diffusion often controls the transport of dissolved species, which may, however, be superimposed by the bioirrigating activity of macrozoobenthos (Boudreau, 1997; Meysman et al., 2006).

These substantial differences cause differences in the biogeochemical properties of the substrate types. Porewater advection in permeable sediments does not only transport solutes but also particulate material. Fresh and labile organic matter (POC and DOC) from the fluff layer can be quickly transported into permeable sediments, the latter in this way acting as a kind of bioreactor. The typically low contents of reactive organics in sand led for a long time to the consideration of sands as "geochemical deserts" (Boudreau et al., 2001). In parallel, the low content of clay minerals and associated organic matter is often accompanied by lower microbial cell numbers when compared to muddy substrates (e.g., Llobet-Brossa, 1998; Böttcher et al., 2000). It has, however, been shown that microbial turnover rates also in sands may be high (Werner et al., 2006; Al-Raei et al., 2009). Actually, the supply of fresh organic material may lead to fast microbial degradation rates comparable to those of the organic-rich muddy sediments where more refractory organic material is accumulating at depth. The high mixing rates of pore water in the sands then bring together reactants for secondary reactions like coupled nitrification-denitrification, which makes these areas an effective biological filter, even if pore water concentrations are low compared to impermeable sediments. In our area of investigation, oxygen fluxes and sulphate reduction rates are comparable between sandy and muddy sites, while the organic content differs by an order of magnitude (Lipka et al., 2018b).

## 1.6 Fluff layer representation

As mentioned earlier, the transport of fluffy layer material from coast to basin areas is an important process in our region of interest. Previous studies with a pelagic ecosystem model (Radtke et al., 2012), which includes fluff layer dynamics, support this experimental finding and highlight the role of this mechanism for the overall nutrient exchange between coasts and basins. For this reason, we explicitly include the fluff layer in our model as a third compartment in addition to water column and sediment. This approach, which is similar to Lee et al. (2002), is in contrast to most other coupled benthopelagic models. We see the explicit representation of the fluff layer dynamics as one of the major advantages of our model.

## 1.7 Article structure

This article is structured as follows. In Section 2 we present a description of the model and the processes which are included. In Section 3, we summarise which empirical data were used and give a brief explanation on how they were obtained. In Section 4,

we describe how these data were used to fit the model to the different stations, since the seven stations mentioned before serve as the test case for our model. The model results are shown and discussed in Section 5, where we provide a summary of the scope of model application and its limitations. The paper ends with Section 6, in which conclusions and an outlook toward the model's future application within a three-dimensional ecosystem model framework are given.

## 2 Model description

In this section, we give a description of the combined benthic-pelagic model. We start in Section 2.1 with a brief introduction to the two ancestor models it descended from. The model is a purely biogeochemical, not a physical model, so section 2.2 describes how the physics affecting the biogeochemical processes are prescribed. We then explain the model compartments and state variables in Section 2.3. Before giving the full model equations in Section 2.5, we first explain the vertical transport processes which occur in these equations in Section 2.4.

The core of model is obviously the biogeochemical processes represented within it. Their description therefore forms the major part of this manuscript. Biogeochemical processes in the water column are described in Section 2.6 and those in the sediment follow in Section 2.7. The carbonate system is the same in both compartments and is described separately in Section 2.8. Since most of the biogeochemical processes included in our model are already contained in preceding models in exactly the same way, we decided to only give a qualitative description of them in the main text. The quantitative details, including the values of the model constants we used, are presented in a separate, complete description in the supplementary material. In contrast, we give a detailed and quantitative description of the "new" processes in the main text, i.e. those that are less common or those that differ from the ancestor models, since we assume that this will be the most interesting part for the majority of readers. The supplementary material also contains a table of the model constants and the sensitivities of the model results to changes in the individual parameter values.

The model description is completed by giving details on numerical aspects in Section 2.9. Finally, in Section 2.10, we give a short note on the procedure by which we automatically generate the model code from a formal description of the model processes.

### 2.1 Ancestor models

The combined benthic-pelagic model is based on two ancestors:

- The water column part is based on ERGOM, an ecological model developed originally for the Baltic Sea (Neumann, 2000). It has been continuously developed since its first publication, the latest improvements include introducing refractory dissolved organic nitrogen (Neumann et al., 2015), and transparent exopolymers (Neumann et al., 2017). From the start, ERGOM contained three functional groups of phytoplankton, representing large-cell (diatom) and small-cell (flagellate) primary producers as well as diazotroph cyanobacteria, and the ability to simulate hypoxic/anoxic conditions. ERGOM is typically used in a three-dimensional context as a part of marine ecosystem models. With some modifications, it has been applied for different ecosystems such as the North Sea (Maar et al., 2011) and the Benguela upwelling system

(Schmidt and Eggert, 2016). It is an intermediate-complexity model for the lower trophic levels up to zooplankton and has been applied for a broad range of scientific questions.

- The sediment part is based on a model developed for a study on the effect of seasonal hypoxia on sedimentary phosphorus accumulation in the Arkona Sea (Reed et al., 2011). This model is, as many others of its kind, a descendant of the van Cappellen and Wang (1996) model, which focused on the sedimentary iron and manganese cycle, and the mineralisation pathways of oxic mineralisation, denitrification, and sulphate reduction. An extensive literature survey (combined with model fitting to observations) allowed the estimation of a large quantity of model constants such as solubility products and half-saturation constants. These were later on inherited by several early diagenetic models, including the one presented in this article. These models solve the diagenetic equations, typically applied at a well-defined single site as a one-dimensional setup.

  Like the present one, the model by Reed et al. (2011) is a prognostic model and solves the time-dependent equations rather than making a steady-state assumption.

## 2.2 Physical parameters used in the model simulations

Since our model is a purely biogeochemical model, it requires a physical environment in which it is embedded. In a final, three-dimensional application, this will be a hydrodynamic host model, and the biogeochemical model described in this communication will be coupled into it. Since we do an intermediate step first and run the model in one-dimensional setups, we need to provide physical quantities as model input. The variables which influence the biogeochemical processes in the water column are

- temperature,

- salinity,

- light intensity,

- bottom shear stress and

- vertical turbulent diffusivity.

These are prescribed by forcing files[1] which need to be provided in order to run the one-dimensional model. We obtain these data from a three-dimensional model simulation of the Baltic Sea ecosystem (Neumann et al., 2017). This simulation was performed using the Modular Ocean Model (MOM) version 5.1 (Griffies, 2018). The model had a horizontal resolution of 3 n.m. and a vertical resolution of 2 m and covered the entire Baltic Sea. Open boundary conditions were applied in the Skagerrak at the transition to the North Sea. The model was driven by atmospheric forcing data from the coastDat dataset (Weisse et al., 2009) which were extended in time using data from the German Weather Service (Schulz and Schattler, 2014).

---

[1]`physics/temperature.txt`, `physics/salinity.txt`, `physics/light_at_top.txt`, `physics/bottom_stress.txt`, `physics/diffusivity.txt`, found in the subdirectories `stations/station_??` in the supplementary material

The ERGOM ecosystem model, as described in the previous section, was implemented in the physical host model, so it produced a hindcast simulation of both physics and biogeochemistry of the Baltic Sea ecosystem. We extracted model output from the simulated year 2015 at the different locations as input for the 1-d model. Since we run the 1-d model for a longer period, the physical forcing is repeated every year.

## 2.3 Model compartments and state variables

The one-dimensional model consists of four compartments as shown schematically in Figure 2:

1. The water column,

2. a fluff layer deposited on the sediment surface,

3. the sedimented solids, and

4. the pore water between them.

The water column and the sediment are vertically resolved, the former in layers of 2 m depth such that their number depends on the water depth of the specific site, the latter in 22 layers increasing in depth from 1 mm at the sediment surface to 2 cm at the bottom of the modelled sediment in 22 cm depth. These specific numbers are not intrinsic to the model but can be changed in the input files[2]. The current choice of 22 cm for the sediment depth was taken according to the availability of pore water data.

The chosen vertical resolution must be seen as a compromise between speed and accuracy. Especially for the 3-d application, we want to keep the numerical effort of the calculations as small as possible. A comparison to a run with double resolution is shown in Appendix E, it shows minor deviations between the resolutions.

Sediment porosity is prescribed[3] and site-specific. As a simplifying assumption, accumulating organic material does not change the porosity. Similarly, the amount of material accumulated in the fluff layer does not change the remaining volume in the bottom water cell.

The tracers (model state variables) present in each of the compartments are listed in Table 1. All of the tracers have a fixed stoichiometric composition which is shown in Appendix A. Where stoichiometric ratios change, such as during detritus decomposition, more than one tracer is needed. This means we can check mass conservation at design time of the model by formulating it in a process-based way as outlined in Radtke and Burchard (2015). To check this mass conservation, the chemical reaction equations need to be formulated in a complete way, which is why "virtual tracers" such as water may be included in the process formulation, even if they do not occur as state variables in the model.

Total alkalinity is a parameter describing the buffering capacity of a solution against adding acids, it describes the amount of a strong acid that needs to be added to titrate it to a pH of 4.3. In our model, it is represented as a "combined tracer", which

---

[2]`physics/cellheights.txt, physics/sed_cellheights.txt`
[3]`physics/sed_inert_ratio.txt`

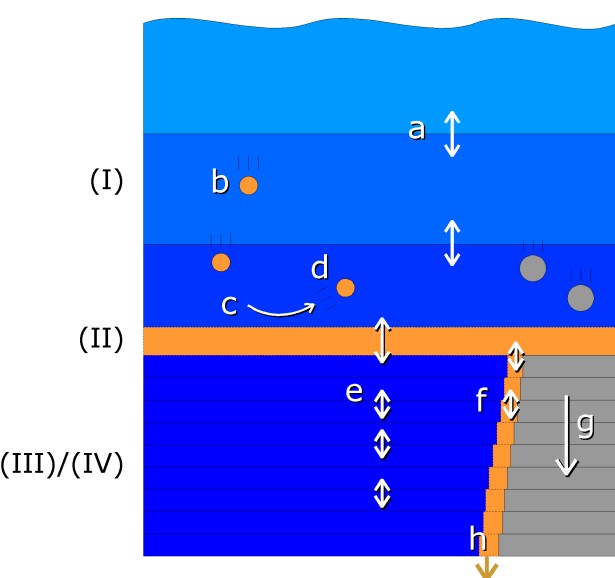

**Figure 2.** Schematic view of the compartments and vertical exchange processes in the model. Compartments: (I) water column, (II) fluff layer, (III) pore water, (IV) solid sediment. Both water column and sediment consist of several vertically stacked grid cells. Vertical transport processes: a = turbulent mixing, b = particle sinking, c = sedimentation, d = resuspension, e = bioirrigation combined with molecular diffusion, f = bioturbation, g = sediment growth, h = burial. Bioactive solid material is shown in orange, bioinert solid material in grey and water in blue.

means that its rate of change depends on its constituents ($OH^-$, $H_3O^+$, $PO_4^{3-}$) which are actively produced or consumed. The reasoning behind this is explained in Section 2.8.

The state variables will not be discussed one-by-one here, but rather in the section about the biogeochemical processes (Sections 2.6 and 2.7), where their role in the ecosystem will be explained.

## 2.4    Transport processes

The processes which transport the tracers vertically are schematically shown in Figure 2. Their detailed implementation is discussed here.

Horizontal exchange (transport) is neglected in our one-dimensional model. This is obviously an inadequate approximation for the water column processes, as we do not consider basins, but rather single stations, some of which are situated in proximity to river mouths, where lateral transport processes have a major impact (Schneider et al., 2010; Emeis et al., 2002; Christiansen et al., 2002). We solve this issue in the future application of the biogeochemical model in a three-dimensional model system (Cahill et al., in prep.).

In this model, we are not specifically interested in the water column as such but rather see it as being responsible for delivering the right amount of sedimenting detritus at the right time. To obtain this, we relax the wintertime nutrients in the surface layer to a realistic value. This may be seen as a parametrisation of a lateral exchange process. In addition, transport

**Table 1.** Tracers used in the ERGOM SED v1.0 model

| name | W | F | S | P | description | unit |
|---|---|---|---|---|---|---|
| t_lpp | + | | | | large-cell phytoplankton | mol kg$^{-1}$ (N units) |
| t_spp | + | | | | small-cell phytoplankton | mol kg$^{-1}$ (N units) |
| t_cya | + | | | | diazotroph cyanobacteria | mol kg$^{-1}$ (N units) |
| t_zoo | + | | | | zooplankton | mol kg$^{-1}$ (N units) |
| t_det_? | + | | | | detritus, N+C, fast decaying (1) to inert (6) | mol kg$^{-1}$ (N units) |
| t_detp_? | + | | | | phosphate in detritus, fractions 1 to 6 | mol kg$^{-1}$ (N units) |
| t_don | + | | | | autochthonous dissolved organic nitrogen | mol kg$^{-1}$ |
| t_poc | + | | | | particulate organic carbon | mol kg$^{-1}$ |
| t_ihw | + | | | | suspended iron hydroxide | mol kg$^{-1}$ |
| t_ipw | + | | | | suspended phosphate bound to Fe-III | mol kg$^{-1}$ |
| t_mow | + | | | | suspended manganese oxide | mol kg$^{-1}$ |
| t_n2 | + | | | + | dissolved molecular nitrogen | mol kg$^{-1}$ |
| t_o2 | + | | | + | dissolved molecular oxygen | mol kg$^{-1}$ |
| t_dic | + | | | + | dissolved inorganic carbon | mol kg$^{-1}$ |
| t_alk | + | | | + | total alkalinity | mol kg$^{-1}$ |
| t_nh4 | + | | | + | ammonium | mol kg$^{-1}$ |
| t_no3 | + | | | + | nitrate | mol kg$^{-1}$ |
| t_po4 | + | | | + | phosphate | mol kg$^{-1}$ |
| t_h2s | + | | | + | hydrogen sulphide | mol kg$^{-1}$ |
| t_sul | + | | | + | elemental sulphur | mol kg$^{-1}$ |
| t_so4 | + | | | + | sulphate | mol kg$^{-1}$ |
| t_fe2 | + | | | + | ferrous iron | mol kg$^{-1}$ |
| t_ca2 | + | | | + | dissolved calcium | mol kg$^{-1}$ |
| t_mn2 | + | | | + | dissolved manganese-II | mol kg$^{-1}$ |
| t_sil | + | | | + | silicate | mol kg$^{-1}$ |
| t_ohm_quickdiff | + | | | + | OH- ions with realistically quick diffusion | mol kg$^{-1}$ |
| t_ohm_slowdiff | + | | | + | OH- ions which move unrealistically slow with alkalinity | mol kg$^{-1}$ |
| t_sed_? | | + | + | | sedimentary detritus N+C, fractions 1 to 6 | mol m$^{-2}$ (N units) |
| t_sedp_? | | + | + | | phosphate in sedimentary detritus, fractions 1 to 6 | mol m$^{-2}$ (N units) |
| t_ihs | | + | + | | iron hydroxide in the sediment | mol m$^{-2}$ |
| t_ihc | | + | + | | iron hydroxide in the sediment - crystalline phase | mol m$^{-2}$ |
| t_ips | | + | + | | iron-bound phosphate in the sediment | mol m$^{-2}$ |
| t_ims | | + | + | | iron monosulphide | mol m$^{-2}$ |
| t_pyr | | + | + | | pyrite | mol m$^{-2}$ |
| t_mos | | + | + | | manganese oxide in the sediments | mol m$^{-2}$ |
| t_rho | | + | + | | rhodochrosite | mol m$^{-2}$ |
| t_i3i | | + | + | | potentially reducible Fe-III in illite-montmorillonite mixed layer minerals | mol m$^{-2}$ |
| t_iim | | + | + | | Fe-II adsorbed to illite-montmorillonite mixed layer minerals | mol m$^{-2}$ |
| t_pim | | + | + | | phosphate adsorbed to illite-montmorillonite mixed layer minerals | mol m$^{-2}$ |
| t_aim | | + | + | | ammonium adsorbed to illite-montmorillonite mixed layer minerals | mol m$^{-2}$ |
| h2o | | virtual | | | water molecule | |
| h3oplus | | virtual | | | hydronium ion | |
| ohminus | | virtual | | | hydroxide ion | |
| i2i | | virtual | | | structural Fe-II in illite-montmorillonite mixed-layer minerals | |

W: Water column, F: Fluff layer, S: Solid sediment, P: Pore water, ?: reactivity classes 1 to 6.

of fluff layer material away from or towards the modelled location is a lateral process included in the model. The physical processes which are explicitly included in our model are described here.

### 2.4.1 Turbulent mixing

The vertical exchange due to turbulent mixing in the water column is prescribed externally[4] by a turbulent diffusivity. In our case, it is taken from a three-dimensional MOM5 model run Neumann et al. (2017). In this model setup, turbulent vertical mixing is estimated by the KPP turbulence scheme (K profile parametrisation, Large et al., 1994), which considers both local mixing and, in case of unstable stratification, (non-local) convection. We only take into account the local part of the mixing and apply it to all tracers in the water column.

### 2.4.2 Particle sinking

In our model, suspended particulate matter sinks at a constant rate through the water column. We choose 4.5 m day$^{-1}$ for detritus, 1 m day$^{-1}$ for manganese and iron oxides, including the phosphate adsorbed by them, and 0.5 m day$^{-1}$ for large-cell phytoplankton and particulate organic carbon. In contrast, cyanobacteria are not sinking but, due to their positive buoyancy, they show an upward movement of 0.1 m day$^{-1}$. In reality, the sinking rate differs between individual particles; the currently chosen average values are a result of fitting the previous ERGOM model with the simplified sediment representation to observations.

### 2.4.3 Sedimentation and resuspension

Shear stress at the bottom determines whether erosion or sedimentation takes place. We apply the combined shear stress of currents and waves calculated by the same MOM5 model as the turbulent mixing. If this shear stress $\tau$ is below a critical value of $\tau_c = 0.016$ N m$^{-2}$ (Christiansen et al., 2002), the sinking suspended matter accumulates in the fluff layer compartment. If it is exceeded, the fluff layer material is resuspended into the lowest water cell at a constant relative rate $r_{ero} = 6$ day$^{-1}$.

In our model, no material will ever be resuspended from the sediment itself, which starts below the fluff layer. This means that our model is incapable of realistically capturing extreme events like storms or bottom trawling which winnow the upper layers of the sediment, removing organic material, which has a lower sinking velocity, by separating it from the heavier mineral components (Bale and Morris, 1998). It also neglects a washout, that is, the removal of organic matter from the sediment pores by advective transport of pore water by strong bottom currents (Rusch et al., 2001). In our model, sediment reworking by currents and waves is not explicitly represented, but rather parametrised together with the bioturbation process. This process allows a bi-directional exchange of particulate material between the sediment and the fluff layer, see Section 2.4.5. The upward component of the transport represents winnowing of sediments (Bale and Morris, 1998).

---

[4]`physics/diffusivity.txt`

### 2.4.4 Bioerosion

In environments with oxic bottom waters, we assume that in addition to waves and currents, macrofaunal animals or demersal fish can resuspend organic material from the fluff layer by active movements (Graf and Rosenberg, 1997). Therefore, under oxic conditions, we assume that $r_{biores} = 3\,\%\,\mathrm{day}^{-1}$ of the fluff material is resuspended independently from the shear stress conditions. This number was estimated from calibration of a three-dimensional Baltic Sea ecosystem model (Neumann and Schernewski, 2008) where the process proved to be critical for transporting organic matter to the deep basins below a depth of approx. 60 m. In these depths, a resuspension due to wave-induced shear stress is no longer possible.

### 2.4.5 Bioturbation

Bioturbation describes the movement and mixing of particles inside the sediment caused by the zoobenthos.[5] In fact, it is difficult to discriminate what causes the vertical mixing of particles; also physical effects like bottom shear may have the same effect. We therefore include them in our "bioturbation" process.

We consider bioturbation to act as a vertical diffusivity $D_{B,solids}(z)$ on the concentrations of the different solid species in the sediment. This implies we exclude non-local mixing processes, even if they may be important in nature (Soetaert et al., 1996b), and try to represent them by local mixing. We only take intraphase mixing into account, which means we assume that the porosity $\Phi(z)$ remains constant over time.

The diffusivity $D_{B,solids}(z)$ is also applied to describe the transport between the uppermost sediment layer and the fluff, which is caused by benthic organisms. In reality, the fluff layer may strongly differ in its compaction (porosity) depending on the turbulence conditions. However, we assume it to be perfectly compacted ($\phi = 0$) to be able to apply the above equation to describe the exchange process, and therefore assume a thickness of 3 mm. This is not a physical assumption but rather a numerical trick which we use to transport the fluff material into the sediments. In reality, the fluff layer may be up to a few centimetres thick, and the incorporation of organic matter is done by macrofaunal activities (e.g., van de Bund et al., 2001).

The value of 3 mm describes a volume estimate of SPM (suspended particulate matter) taken from this region: typical SPM concentrations in the lowermost 40 cm of the water column are about 8 mg/l higher compared to the value 5 m above the sea floor (Christiansen et al., 2002). As the density of these particles is just slightly higher than that of the surrounding water, we can estimate their volume at approximately $3\,\mathrm{l\,m}^{-2}$ which gives 3 mm of height if perfectly compacted. We see this explicit treatment of the fluff layer as a major advantage compared to a deposition of sinking particles directly into the surface sediments. We regard it as essential for the application of the model in a three-dimensional setting.

The vertical structure of bioturbation intensity, $D_{B,solids}(z)$, is parametrised vertically as follows:

---

[5]While bioturbation in reality causes both a transport of solids and solutes, we use the term "bioturbation" in the model to describe the transport of solids only, while the transport of solutes is done by the "bioirrigation" process.

$$
D_{B,solids}(z) \quad = \quad \begin{cases} D_{B,solids,max} & \text{for } z < z_{full} \\ D_{B,solids,max} \exp\left(-\frac{z - z_{full}}{z_{decay}}\right) & \text{for } z_{full} < z < z_{max} \\ 0 & \text{for } z_{max} < z \end{cases} \tag{1}
$$

In the uppermost part of the sediment, we assume a constant bioturbation rate. Below that, it decays exponentially with depth until it reaches a maximum depth, which may be below the bottom of our model. So, we externally prescribe (a) the maximum mixing intensity[6] and (b) three length scales describing the vertical structure of bioturbation[7], which are the depth down to which the maximum mixing rate is applied ($z_{full}$), the length scale of exponential decay of the mixing rate below this depth ($z_{decay}$), and the maximum depth of mixing ($z_{max}$).

The present formulation of the model has no explicit dependence of bioturbation depth on the availability of oxidants, i.e. bioturbation will take place in oxic as well as in sulphidic environments; adding this dependence should be essential if the model was applied to sulphidic areas.

### 2.4.6 Bioirrigation

Bioirrigation describes the mixing of solutes within the pore water and the exchange with the bottom water. We describe it as a mixing intensity $D_{B,liquids}(z)$. The vertical profile of bioirrigation intensity is assumed identical to that of bioturbation. The maximum bioirrigation rate is assumed constant in time and prescribed externally[8].

### 2.4.7 Molecular diffusion

Molecular diffusion in the sediment can be described by the equation

$$
\phi(z)\frac{\partial}{\partial t}c(z,t) \quad = \quad D_0(z)\frac{\partial}{\partial z}\left(\frac{\phi(z)}{\theta(z)^2}\frac{\partial c(z,t)}{\partial z}\right), \tag{2}
$$

(Boudreau, 1997). Here $D_0$ describes the molecular diffusivity in a particle-free solution, which is effectively reduced by the effect of hydrodynamic tortuosity $\theta$. This describes the effect that the solutes need to travel a longer path as the direct way may be obstructed by solid particles. It is estimated from porosity by $\theta^2 = 1 - 2.02\ln(\phi)$ (Boudreau, 1997).

A diffusive exchange between the pore water and the overlying bottom water is controlled by the thickness of a diffusive boundary layer. While in reality this relates to the viscous sublayer thickness and is therefore inversely related to the velocity of the bottom water (Boudreau, 1997), we for simplicity assume a constant diffusive boundary layer thickness of 3 mm.

In reality, the diffusive boundary layer thickness is on the order of 1 mm at low bottom shear situations and becomes even shallower if the bottom shear increases (e.g., Gundersen and Jorgensen, 1990). We choose a larger value because we need to account for the transport through the fluff layer as well. A future model version might include a dependence of this parameter on the bottom shear stress.

---

[6]physics/sed_diffusivity_solids.txt
[7]physics/sed_depth_bioturbation.txt
[8]physics/sed_diffusivity_porewater.txt

Molecular diffusivities for the different solute species are calculated from water viscosity following Boudreau (1997). The water viscosity is determined from salinity and temperature (assumed to be identical to that in the bottom water cell). A problem occurs with the combined tracers DIC and total alkalinity, as they do not represent a specific ion but rather a set of different species with different molecular diffusivities. For simplicity, we approximate DIC diffusivity to be that of the $HCO_3^-$ ion, the most common one at the pH values we expect. For total alkalinity, we take a two-step approach: in the first step, we also take the diffusivity of the $HCO_3^-$ ion. But this is an underestimate especially for the $OH^-$ ions which increase in their concentration as the solution becomes alkaline. To take their higher diffusivity into account, we introduce two additional tracers, `t_ohm_slowdiff` and `t_ohm_quickdiff`. Before the molecular diffusion is applied during a model time step, they are both set equal to the $OH^-$ concentrations. During the diffusion time step, the former diffuses with the reduced $HCO_3^-$ diffusion rate, the latter with the $OH^-$ diffusivity. So afterwards, total alkalinity is corrected by adding the difference of the two, `t_ohm_quickdiff-t_ohm_slowdiff`. This results in a smoothed alkalinity profile.

### 2.4.8 Sediment accumulation

In nature, sediments grow upwards as new particulate matter is deposited onto them. In our model, this process is taken into account, but represented as a downward advection of material in the sediment. So, our coordinate system moves upward with the sediment surface. We assume that the sediment growth is supplied by terrigenous, bioinert material, and prescribe[9] a growth rate from literature for the mud stations only (Table 7). We do not assume sediment growth for the sand and silt stations.

We use a simple Euler-Forward advection to move the material from each grid cell into the cell below. Material leaving the model through the lower boundary is lost. Except for organic carbon, we assume that a part of it is mineralised, as will be explained in Section 2.7.1. In the top cell, new organic material from the fluff layer enters by sediment growth.

### 2.4.9 Parametrisation of lateral transport

The Baltic Sea sediments can be classified as accumulation, transport and erosion bottoms (Jonsson et al., 1990). The lateral transport of matter is characterised by the advection of fluff layer material from the transport and erosion bottoms in the shallower areas to the accumulation bottoms in the deep basins (Christiansen et al., 2002). As this process is not represented in our 1-d model setups, we need to parametrise it.

For the sandy and silty sediments, we assume a transport away from the site. This is described by a constant removal rate for all material deposited in the fluff layer. For the mud stations, we assume a transport of organic material towards the site. This is described by a constant input of detritus. Our model contains six detritus classes which degrade at different rates, as will be explained later, in Section 2.6.4. We assume that the quickest-degradable part of the detritus is already mineralised in the shallow coastal areas, before its lateral migration to the mud stations, and therefore exclude the first two classes from this artificial input.

In the 3-d version of the model, these processes are no longer required, as the material is dynamically removed from the shallow sites and transported to deeper ones by advection.

---

[9]`physics/sed_inert_deposition.txt`

## 2.5 Model equations

### 2.5.1 Equations of motion

In this subsection, we will describe the equations of motion solved by the model. The equations in the water column can be derived from the assumption that the vertical (upward) flux of a tracer can be described by an advective and a diffusive flux which follows Fick's law:

$$F_z^{wat}(z,t) \quad = \quad w \cdot c^{wat}(z,t) - D^{wat}(z,t)\frac{\partial}{\partial z}c^{wat}(z,t)\,, \tag{3}$$

where $c^{wat}(z,t)$ denotes the tracer concentration and $D^{wat}$ is the turbulent diffusivity given as external forcing[10]. For particulate matter, the constant $w$ describes its vertical velocity relative to the water, which is negative if the particles are sinking. For dissolved tracers, $w$ is set to zero. We further assume that the water itself does not move vertically. In this case, conservation of mass yields an advection-diffusion equation:

$$\begin{aligned}
\frac{\partial}{\partial t}c^{wat}(z,t) \quad &= \quad -\frac{\partial}{\partial z}F_z^{wat}(z,t) + q_c^{wat}(z,t) \\
&= \quad -w\frac{\partial}{\partial z}c^{wat}(z,t) + \frac{\partial}{\partial z}\left(D^{wat}(z,t)\frac{\partial}{\partial z}c^{wat}(z,t)\right) + q_c^{wat}(z,t)\,,
\end{aligned} \tag{4}$$

where $q_c^{wat}(z,t)$ describes the biogeochemical sources minus sinks of the considered state variable.

The equations in the sediment are different because we need to take porosity into account and treat dissolved tracers (in the pore water) and solid tracers differently. For the pore water tracers, the upward flux is given by

$$F_z^{pw}(z,t) \quad = \quad -\phi(z) \cdot D^{pw}(z,t)\frac{\partial}{\partial z}c^{pw}(z,t)\,, \tag{5}$$

where $\phi(z)$ is the porosity of the sediment (the ratio between pore water volume and total volume) which we assume as constant in time. The concentration $c^{pw}(z,t)$ relates to the pore water volume only. The effective diffusivity $D^{pw}$ is the sum of two contributions, the effective molecular diffusivity $\frac{D_0}{\theta^2}$ and the effective (bio)irrigation diffusivity $D_{B,liquids}(z)$. The advection-diffusion equation is then given by

$$\phi(z)\frac{\partial}{\partial t}c^{pw}(z,t) \quad = \quad \frac{\partial}{\partial z}\left(\phi(z) \cdot D^{pw}(z,t)\frac{\partial}{\partial z}c^{pw}(z,t)\right) + q_c^{pw}(z,t)\,, \tag{6}$$

which is a well-known early diagenetic equation (Boudreau, 1997). For the solid-state tracers, their concentration $c^{sed}(z,t)$ relates to the volume of the solids only, and the flux is given by

$$F_z^{sed}(z,t) \quad = \quad (1-\phi(z))w(z)c^{sed}(z,t) - (1-\phi(z)) \cdot D^{sed}(z,t)\frac{\partial}{\partial z}c^{sed}(z,t)\,, \tag{7}$$

where $w(z)$ is a velocity for a virtual vertical downward transport. It results from sediment growth due to deposition of particulate material, but as we keep the sediment-water interface at a constant position in our model, we need to describe the increasing depth in which we find individual sediment particles as a downward advection. Volume conservation of the

---

[10]physics/diffusivity.txt

particulate material requires that we can write $w(z)$ as

$$w(z) \quad = \quad \frac{w_0}{1 - \phi(z)}, \tag{8}$$

such that the vertical velocity gets smaller in depths where the sediment is more compacted, and $w_0$ describes a theoretical velocity which would occur at perfect compaction ($\phi = 0$)[11]. The advection-diffusion equation then reads

$$5 \quad (1 - \phi(z))\frac{\partial}{\partial t}c^{sed}(z,t) \quad = \quad -w_0\frac{\partial}{\partial z}c^{sed}(z,t) + \frac{\partial}{\partial z}\left((1 - \phi(z)) \cdot D_{B,solids}(z)\frac{\partial}{\partial z}c^{sed}(z,t)\right) + q_c^{sed}(z,t). \tag{9}$$

Practically, we do not store the concentration $c^{sed}(z,t)$ (mol m$^{-3}$) as a state variable but rather the quantity of the tracer per area in a specific layer, $C^{sed}(k,t)$ (mol m$^{-2}$), where $k$ is a vertical index. The transformation is straightforward,

$$C^{sed}(k,t) \quad = \quad \int\limits_{z_{bot,k}}^{z^{top,k}} (1 - \phi(z))\, c^{sed}(z,t)\, dz. \tag{10}$$

For particulate tracers, we also consider a storage in the fluff layer, $C^{fluff}(t)$, which is measured in mol m$^{-2}$. The equation
for $C^{fluff}(t)$ is derived in the following subsection.

### 2.5.2 Boundary conditions

Boundary conditions are required for the partial differential equations given above. We give two boundary conditions for the water column concentrations: one at the sea surface, $z_{surf}$ and one at the sediment-water interface, $z_0$. We also give two boundary conditions for the sediment concentrations: one at the sediment-water interface, $z_0$, and one at the lower model
boundary, $z_{bot}$. We start describing the boundary conditions from bottom to top for the dissolved tracers, and then continue describing them from top to bottom for the particulate / solid-phase state variables.

The pore water tracers have a zero-flux boundary condition at the bottom of the model:

$$F_z^{pw}(z_{bot},t) \quad = \quad 0. \tag{11}$$

An exception to the zero-flux boundary is the parametrisation of sulphide production in the deep which will be discussed later.
At the sediment-water interface, we assume that the dissolved tracers are exchanged between pore water and water column via a diffusive boundary layer of a depth $\Delta z_{bbl}$. So, our upper boundary condition for the pore water tracers is given by

$$F_z^{pw}(z_0,t) \quad = \quad -\phi(z_0) \cdot D^{pw}(z_0,t)\frac{c^{wat}(z_0,t) - c^{pw}(z_0,t)}{\Delta z_{bbl}}. \tag{12}$$

This flux can be directed into or out of the sediment, depending on where the concentration is larger.

To satisfy mass conservation, the vertical flux applied as the lower boundary condition for the dissolved-species concentra-
25 tions in the water column depends on the upward flux from the sediment:

$$F_z^{wat}(z_0,t) \quad = \quad F_z^{pw}(z_0,t) + \tilde{Q}_c^{fluff}(t). \tag{13}$$

---

[11]physics/sed_inert_deposition.txt

The additional term $\tilde{Q}^{fluff}(t)$ represents the sources minus sinks of the dissolved state variable which are caused by biogeochemical transformations of the fluff layer material. At the sea surface, we apply a zero-flux condition, both for dissolved and for particulate state variables:

$$F_z^{wat}(z_{surf}, t) \quad = \quad 0. \tag{14}$$

An exception is only made for tracers which are modified by gas exchange with the atmosphere, e.g. oxygen.

Now the boundary conditions for the particulate state variables are different. The reason is that the water column and the sediment do not directly interact, but we consider the fluff layer as an intermediate layer between the two. Particulate material which sinks to the bottom is deposited in the fluff layer, from where it is incorporated into the sediments.

At the bottom of the water column, there can be two possible situations.

– If the bottom shear stress is lower than the critical shear stress, we assume a deposition of particulate material. This sinking material $(w < 0)$ vanishes from the water column because of sedimentation. It appears in the fluff layer.

– If the bottom shear stress exceeds the critical shear stress, particulate material from the fluff layer is eroded and enters the water column.

In both cases, we additionally consider the bioresuspension process which was described above in Section 2.4.4. We can

therefore formulate the boundary condition for particulate material as

$$F_z^{wat}(z_0, t) \quad = \quad \begin{cases} \min(w, 0) \cdot c^{wat}(z_0, t) + r_{biores}(t) \cdot C^{fluff}(t) & \text{for } \tau(t) \leq \tau_c \\ r_{ero} \cdot C^{fluff}(t) + r_{biores}(t) \cdot C^{fluff}(t) & \text{for } \tau(t) > \tau_c \end{cases} . \tag{15}$$

The fluff interacts with the surface sediment layer in two ways. Firstly, sediment growth means an incorporation of fluff layer material into the surface sediments. Secondly, bioturbation which is considered as a diffusion-analog mixing leads to an exchange of particulate material between fluff layer and surface sediment. So, the boundary condition for solids at the sediment

surface is given by

$$F_z^{sed}(z_0, t) \quad = \quad w_0 \frac{C^{fluff}(t)}{\Delta z_{fluff}} - (1 - \phi(z_0)) \cdot D^{sed}(z_0, t) \frac{\frac{C^{fluff}(t)}{\Delta z_{fluff}} - c^{sed}(z_0, t)}{\Delta z_{fluff}} . \tag{16}$$

Here, $\Delta z_{fluff}$ represents a virtual thickness of the fluff layer assuming it was perfectly compacted, see the discussion in Section 2.4.5. In this way, the benthofaunal processes of incorporating fluff layer material into the surface sediments can be simply described as a diffusion-analog flux of particulates. The opposite processes which cause a removal of fine-grained

material from the sediments, winnowing or washout, can in the same way be described as a diffusion process, in this case upward. This occurs in the model especially when the fluff layer material is resuspended during periods of high bottom shear and the concentration $C^{fluff}(t)$ is correspondingly low.

The concentration change in the fluff layer is then defined by mass conservation, and is simply given by

$$\frac{\partial}{\partial t} C^{fluff}(t) \quad = \quad F_z^{sed}(z_0, t) - F_z^{wat}(z_0, t) + Q_c^{fluff}(t) \tag{17}$$

for all particulate state variables. Here, $Q_c^{pw}(t)$ describes the sources minus sinks term from the biogeochemical transformations of the considered state variable.

Finally, the burial of particulate material at the lower model boundary can be described by the following boundary condition:

$$F_z^{sed}(z_{bot},t) \quad = \quad w_0 c^{sed}(z_{bot},t). \tag{18}$$

So, we assume the particulate material to be buried forever when it leaves the model domain. An exception, as mentioned before, is the parametrisation of deep sulphide formation which is described in Section 2.7.

## 2.6 Biogeochemical processes in the water column

In this section, we describe the biogeochemical processes acting in the water column. These are mostly identical to previously published ERGOM versions (e.g., Neumann and Schernewski, 2008; Neumann et al., 2015), which contained a more simple,
vertically integrated sediment model. As in the previous section, we provide the quantitative description including the model constants in the online supplement.

A reaction network table giving the reaction equations including their stoichiometric coefficients is given in Table 2.

### 2.6.1 Primary production and phytoplankton growth

There are three classes of phytoplankton in the model, representing large-cell and small-cell microalgae as well as diazotroph
cyanobacteria. Their growth is determined by a class-specific maximum growth rate, but contains two limiting factors for nutrients and light. The light limitation is a saturation function with optimal growth at a class-specific optimum level or at 50% of the surface radiation. The short wave light flux at the surface is taken from a dynamically down-scaled ERA40 atmospheric forcing (Uppala et al., 2005), using the regional Rossby Centre Atmosphere model (RCA). Nutrient limitation is a quadratic Michaelis-Menten term for DIN (nitrate + ammonium) or phosphate, depending on which one is limiting, based on Redfield
stoichiometry. Diazotroph cyanobacteria are only limited by phosphate and not by DIN, but they are only allowed to grow in a specific salinity range. Cyanobacteria and small-cell algae also require a minimum temperature to grow (Wasmund, 1997; Andersson et al., 1994).

However, according to Engel (2002) although nutrients are limiting an enhanced polysaccharide exudation could be the result of a cellular carbon overflow, whenever nutrient acquisition limits biomass production but not photosynthesis. These
transparent exopolymers are included in our model, they are assumed to have a constant sinking velocity.

### 2.6.2 Phytoplankton respiration and mortality

We assume a constant respiration of phytoplankton which is proportional to its biomass. As the model maintains the Redfield ratio, the degradation of biomass (catabolism) goes along with an excretion of ammonium and phosphate. This simplified description of phytoplankton growth does not describe day/night metabolism or temperature dependence. A small fraction
of the nitrogen is released as dissolved organic nitrogen (DON). In the model, this represents the DON fraction which is less utilisable by phytoplankton, while the fraction with high bioavailability is considered to be part of the ammonium state variable.

**Table 2.** Reaction network table for the processes in the water column. See Table A1 for the composition of state variables. Processes marked with a * also take place in the pore water.

| number | forward (backward) reaction | equation |
|---|---|---|
| 1 | p_no3_assim_lpp | $1.1875\,H_3O^+ + 6.4375\,H_2O + 6.625\,CO_2 + 0.0625\,PO_4^{3-} + NO_3^- \rightarrow 8.625\,O_2 + \texttt{t\_lpp}$ |
| 2 | p_nh4_assim_lpp | $NH_4^+ + 0.0625\,PO_4^{3-} + 6.625\,CO_2 + 7.4375\,H_2O \rightarrow \texttt{t\_lpp} + 6.625\,O_2 + 0.8125\,H_3O^+$ |
| 3 | p_no3_assim_spp | $1.1875\,H_3O^+ + 6.4375\,H_2O + 6.625\,CO_2 + 0.0625\,PO_4^{3-} + NO_3^- \rightarrow 8.625\,O_2 + \texttt{t\_spp}$ |
| 4 | p_nh4_assim_spp | $NH_4^+ + 0.0625\,PO_4^{3-} + 6.625\,CO_2 + 7.4375\,H_2O \rightarrow \texttt{t\_spp} + 6.625\,O_2 + 0.8125\,H_3O^+$ |
| 5 | p_n2_assim_cya | $7.9375\,H_2O + 6.625\,CO_2 + 0.0625\,PO_4^{3-} + 0.5\,N_2 + 0.1875\,H_3O^+ \rightarrow 7.375\,O_2 + \texttt{t\_cya}$ |
| 6 | p_lpp_resp_nh4 | $\texttt{t\_lpp} + 6.625\,O_2 + 0.8125\,H_3O^+$ $\rightarrow 0.1\,\texttt{t\_don} + 0.9\,NH_4^+ + 0.0625\,PO_4^{3-} + 6.625\,CO_2 + 7.4375\,H_2O$ |
| 7 | p_spp_resp_nh4 | $0.8125\,H_3O^+ + 6.625\,O_2 + \texttt{t\_spp}$ $\rightarrow 7.4375\,H_2O + 6.625\,CO_2 + 0.0625\,PO_4^{3-} + 0.9\,NH_4^+ + 0.1\,\texttt{t\_don}$ |
| 8 | p_cya_resp_nh4 | $0.8125\,H_3O^+ + 6.625\,O_2 + \texttt{t\_cya}$ $\rightarrow 7.4375\,H_2O + 6.625\,CO_2 + 0.0625\,PO_4^{3-} + 0.1\,\texttt{t\_don} + 0.9\,NH_4^+$ |
| 9 | p_lpp_graz_zoo | $\texttt{t\_lpp} \rightarrow \texttt{t\_zoo}$ |
| 10 | p_spp_graz_zoo | $\texttt{t\_spp} \rightarrow \texttt{t\_zoo}$ |
| 11 | p_cya_graz_zoo | $\texttt{t\_cya} \rightarrow \texttt{t\_zoo}$ |
| 12 | p_zoo_resp_nh4 | $0.8125\,H_3O^+ + 6.625\,O_2 + \texttt{t\_zoo}$ $\rightarrow 7.4375\,H_2O + 6.625\,CO_2 + 0.0625\,PO_4^{3-} + 0.9\,NH_4^+ + 0.1\,\texttt{t\_don}$ |
| 13 | p_don_rec_nh4 | $\texttt{t\_don} \rightarrow NH_4^+$ |
| 14 | p_lpp_mort_det_? | $\frac{1}{3}\,H_3O^+ + \texttt{t\_lpp} \rightarrow \frac{1}{3}\,H_2O + \frac{1}{3}\,NH_4^+ + \frac{2}{3}\,\texttt{t\_det\_?} + \frac{2}{3}\,\texttt{t\_detp\_?}$ |
| 15 | p_spp_mort_det_? | $\frac{1}{3}\,H_3O^+ + \texttt{t\_spp} \rightarrow \frac{1}{3}\,H_2O + \frac{1}{3}\,NH_4^+ + \frac{2}{3}\,\texttt{t\_det\_?} + \frac{2}{3}\,\texttt{t\_detp\_?}$ |
| 16 | p_cya_mort_det_? | $\texttt{t\_cya} + \frac{1}{3}\,H_3O^+ \rightarrow \frac{2}{3}\,\texttt{t\_detp\_?} + \frac{2}{3}\,\texttt{t\_det\_?} + \frac{1}{3}\,NH_4^+ + \frac{1}{3}\,H_2O$ |
| 17 | p_zoo_mort_det_? | $\frac{1}{3}\,H_3O^+ + \texttt{t\_zoo} \rightarrow \frac{1}{3}\,H_2O + \frac{1}{3}\,NH_4^+ + \frac{2}{3}\,\texttt{t\_det\_?} + \frac{2}{3}\,\texttt{t\_detp\_?}$ |
| 18 | p_nh4_nit_no3* | $H_2O + 2\,O_2 + NH_4^+ \rightarrow 2\,H_3O^+ + NO_3^-$ |
| 19 | p_h2s_oxo2_sul* | $H_2S + 0.5\,O_2 \rightarrow S + H_2O$ |
| 20 | p_h2s_oxno3_sul* | $0.4\,H_3O^+ + 0.4\,NO_3 + H_2S \rightarrow 0.2\,N_2 + 1.6\,H_2O + S$ |
| 21 | p_sul_oxo2_so4* | $3\,H_2O + 1.5\,O_2 + S \rightarrow 2\,H_3O^+ + SO_4^{2-}$ |
| 22 | p_sul_oxno3_so4* | $1.2\,H_2O + 1.2\,NO_3 + S \rightarrow 0.6\,N_2 + 0.8\,H_3O^+ + SO_4^{2-}$ |
| 23 | p_fe2_ox_ihw | $0.25\,O_2 + 4.5\,H_2O + Fe^{2+} \rightarrow Fe(OH)_3^{susp} + 2\,H_3O^+$ |
| 24 | p_po4_ads_ipw (p_ipw_diss_po4) | $Fe(OH)_3^{susp} + PO_4^{3-} \leftrightarrow FePO4 + 3\,OH^-$ |

Due to simplification, in our model phytoplankton experiences a constant background mortality, although we know this is far away from reality where it is species-specific and depends on abiotic (e.g. nutrient, light etc.) and biotic conditions. An additional mortality is generated by grazing of zooplankton as described next.

### 2.6.3 Zooplankton processes

Zooplankton is only represented as one bulk state variable. It grows by assimilating any type of phytoplankton, however, it has a smaller food preference for the cyanobacteria class compared to the other classes. The uptake becomes limited by a Michaelis-Menten-function if the zooplankton´s food approaches a saturation concentration. Feeding can only take place in oxic waters and is temperature-dependent. It shows a maximum at an optimum temperature and a double-exponential decrease when this temperature is exceeded.

Both zooplankton respiration and mortality represent a closure term for the model. They are meant to include the respiration and mortality of the higher trophic levels (fish) which feed on zooplankton, therefore we use a quadratic closure. Mortality is additionally enhanced under anoxic conditions which, however, do not occur in our study area.

### 2.6.4 Mineralisation processes

The description of detritus[12] differs from the previous ERGOM versions. We have split the detritus into six classes, depending on its degradability. This degradability is described as a decay rate constant, which ranges from 0.065 $\text{day}^{-1}$ for the first class to $1.6 \cdot 10^{-5}$ $\text{day}^{-1}$ for the fifth class, while the last one is assumed to be completely bioinert. This type of model is known as a "multi-G model" (Westrich and Berner, 1984).

Details on the specific choice of the classes are given in Appendix B.

The mineralisation is, however, temperature dependent by a $Q_{10}$ rule (Thamdrup et al., 1998; Sawicka et al., 2012), as it is realised by microbial processes; the values given above are valid at 0°C. The 0°C choice is somewhat arbitrary. Actually, the model is not very sensitive to this choice, as an enhanced baseline temperature, meaning a lower decomposition rate of each class, would be compensated for by a shift in the class composition, leaving higher concentrations of quickly-degradable detritus classes which means overall a very similar total decomposition rate, see Appendix B.

When organic detritus is created by plankton mortality, it is partitioned into the different classes in a constant ratio. This ratio was determined from a fit of the multi-G model to an empirical relation between detritus age and its relative decay rate which was proposed by Middelburg (1989). The fraction of non-decaying detritus was estimated from empirically determined carbon burial rates in the Baltic Sea (Leipe et al., 2011).

The chemical composition of detritus is, in contrast to phyto- and zooplankton, not determined by the Redfield ratio. It is enriched in carbon and phosphorus by 50 %, such that it has a C:N:P ratio of 159:16:1.5. This resembles detritus compositions as they were determined in sediment traps and by investigating fluffy layer material in the Baltic Sea (Heiskanen and Leppänen, 1995; Emeis et al., 2000, 2002; Struck et al., 2004).

In the water column, detritus can be mineralised by three different oxidants: oxygen, nitrate and sulphate. They are utilised in this order; if the preferential oxidant's concentration declines, the specific pathway is reduced by a Michaelis-Menten limiter and the next pathway takes over such that the total mineralisation is held constant. In all pathways, DIC, ammonium and phos-

---

[12]Throughout the manuscript, we use the term "detritus" in its biological meaning. Here it describes dead particulate organic material only, as opposed to its use in geology, where the term includes deposited mineral particles.

phate are released. Nitrate reduction also produces molecular nitrogen (heterotrophic denitrification), while sulphate reduction generates hydrogen sulphide.

Mineralisation of particulate organic carbon in transparent exopolymers takes place via the same pathways, but only releases DIC. DON is also mineralised after some time and decays to ammonium (which may represent the transformation to bioavailable DON compounds).

### 2.6.5 Reoxidation of reduced substances

In the presence of oxygen, ammonium is nitrified to nitrate (e.g., Guisasola et al., 2005). The intermediate step, formation of nitrite, is omitted in the model. Hydrogen sulphide can be reoxidised by oxygen or by nitrate (chemolithoautotrophic denitrification) (e.g., Bruckner et al., 2013). This takes place as a two-step process via the formation of elemental sulphur (Jørgensen, 2006). All reoxidation processes exponentially increase their rates with temperature.

In the sediments, we additionally assume that $Fe^{2+}$ can be produced as a reduced substance. If it is released from the sediments and enters the water column, it can be reoxidised by oxygen, creating suspended iron oxyhydroxides.

### 2.6.6 Adsorption and desorption reactions

Dissolved phosphate can be adsorbed to iron oxyhydroxide particles suspended in the water column. In the same way, phosphate adsorbed to iron oxyhydroxide particles can be released if the ambient concentration of phosphate is low. The process is identical to the one in the sediments and is discussed in Section 2.7.5 in detail.

## 2.7 Biogeochemical processes in the fluff layer, sediment and pore water

In this section, we qualitatively describe the sedimentary biogeochemical processes contained in the model. For a quantitative description including the model constants, we refer to the online supplementary material. Figure 3 gives a schematic overview of the processes considered in the sediment model. As every model, the chosen set of biogeochemical processes and variables does not aim at completeness in its representation of reality, but rather at the strongest possible simplification which still retains the required complexity to describe the processes we are interested in. For this reason, we do not, for example, consider methane formation explicitly.

The stoichiometry of the processes included in the model is shown in three reaction network tables:

– Primary redox reactions are given in Table 3.

– Secondary redox reactions are given in Table 4.

– Adsorption/desorption and precipitation/dissolution reactions are given in Table 5.

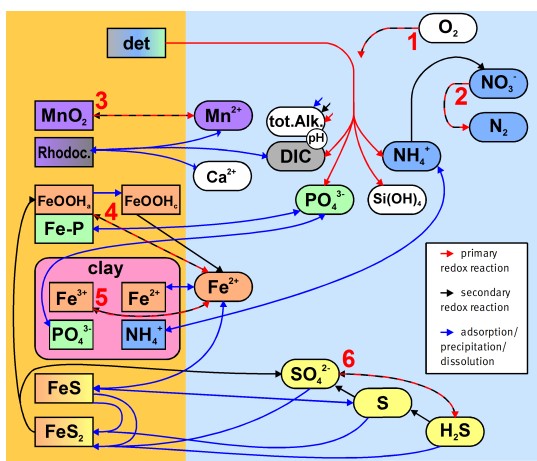

**Figure 3.** Simplified sketch of state variables and processes in the sediment model. Boxes to the left and right indicate sediment and pore water state variables, respectively. pH is not a state variable but calculated from DIC and total alkalinity. Red arrows show primary redox processes, driven by oxidation of organic carbon. The red numbers indicate the order in which the oxidants are utilised. Black arrows show secondary redox reactions, which means reoxidation of reduced substances. Blue arrows show adsorption/desorption or precipitation/dissolution reactions, which may depend on pH. Abbreviations: det = detritus, Rhodoc. = rhodochrosite, tot.Alk. = total alkalinity, DIC = dissolved inorganic carbon

### 2.7.1 Mineralisation in general

The mineralisation of detritus is the dominant biogeochemical process in the sediments, as the oxidation of the carbon therein is the major supply of chemical energy for microbes.

As in the water column, oxidants are utilised in a specific order, and a smooth transition to the next mineralisation pathway occurs when the preferred one gets exhausted. However, the number of possible oxidants is increased in the sediment, as here also solid components may act as electron acceptors. The order in which they are utilised is (Boudreau, 1997):

1. oxygen

2. nitrate

3. manganese oxide

4. iron oxyhydroxide

5. Fe-III contained in clay minerals

6. sulphate

After sulphate is exhausted, typically the formation of methane would start. This process is omitted in the current model, as we designed our model for the top 22 cm of the south-western part of the Baltic Sea, where we do not expect sulphate to be

**Table 3.** Reaction network table for the primary redox reactions in the sediment and the fluff layer. See Table A1 for the composition of state variables.

| number | forward (backward) reaction | equation |
|---|---|---|
| 25 | `p_sed_?_resp_nh4` | $H_3O^+ + 9.9375\,O_2 + $`t_sed_?` $\rightarrow 0.9375\,Si(OH)_4 + 10.9375\,H_2O + 9.9375\,CO_2 + NH_4^+$ |
| 26 | `p_sed_?_denit_nh4` | `t_sed_?` $+ 8.95\,H_3O^+ + 7.95\,NO_3^-$ $\rightarrow 9.9375\,CO_2 + NH_4^+ + 3.975\,N_2 + 22.8625\,H_2O + 0.9375\,Si(OH)_4$ |
| 27 | `p_sed_?_mnred_mn2` | `t_sed_?` $+ 40.75\,H_3O^+ + 19.875\,MnO_2$ $\rightarrow 0.9375\,Si(OH)_4 + 70.5625\,H_2O + 19.875\,Mn^{2+} + NH_4^+ + 9.9375\,CO_2$ |
| 28 | `p_sed_?_irred_ims` | $39.75\,H_2S + 39.75\,Fe(OH)_3 + H_3O^+ + $`t_sed_?` $\rightarrow 9.9375\,CO_2 + NH_4^+ + 39.75\,FeS + 110.3125\,H_2O + 0.9375\,Si(OH)_4$ |
| 29 | `p_sed_?_irredips_ims` | $39.75\,FePO_4 + 239.78125\,H_3O^+ + 129.1875\,H_2O + $`t_sed_?`$ + 39.75\,H_2S$ $\rightarrow 0.9375\,Si(OH)_4 + 9.9375\,CO_2 + NH_4^+ + 39.75\,FeS + H_2O$ $+ 39.75\,PO_4^{3-} + 358.03125\,H_3O^+$ |
| 30 | `p_sed_?_irred_iim` | $79.5\,H_3O^+ + $`t_sed_?`$ + H_3O^+ + 39.75\,Fe(OH)_3 + 79.5\,OH^-$ $\rightarrow 0.9375\,Si(OH)_4 + 188.8125\,H_2O + H_2O + 39.75\left(Fe^{2+} + 2OH^-\right)^{ads-clay}$ $+ NH_4^+ + 9.9375\,CO_2$ |
| 31 | `p_sed_?_irredips_iim` | $39.75\,FePO_4 + $`t_sed_?`$ + 80.5\,H_3O^+ + 49.6875\,H_2O + 79.5\,OH^-$ $\rightarrow 0.9375\,Si(OH)_4 + 9.9375\,CO_2 + 39.75\,PO_4^{3-} + NH_4^+ +$ $39.75\left(Fe^{2+} + 2OH^-\right)^{ads-clay}$ $+ H_2O + 119.25\,H_3O^+$ |
| 32 | `p_i3i_?_irred_i2i` | `t_sed_?`$ + H_3O^+ + 39.75\left(Fe^{3+} + 3OH^-\right)^{in-clay}$ $\rightarrow 0.9375\,Si(OH)_4 + 30.8125\,H_2O + 39.75\left(Fe^{2+} + 2OH^-\right)^{in-clay} + NH_4^+$ $+ 9.9375\,CO_2$ |
| 33 | `p_sed_?_sulf_nh4,` `p_sed_?_sulfdeep_nh4` | $10.9375\,H_3O^+ + 4.96875\,SO_4^{2-} + $`t_sed_?` $\rightarrow 20.875\,H_2O + 4.96875\,H_2S + 9.9375\,CO_2 + NH_4^+ + 0.9375\,Si(OH)_4$ |
| 34 | `p_sedp_?_remin_po4,` `p_sedp_?_sulfdeep_po4` | `t_sedp_?`$ + 0.28125\,H_2O$ $\rightarrow 0.28125\,H_3O^+ + 0.09375\,PO_4^{3-}$ |

limiting. This depth restriction is based on the limited length of the sediment cores taken in the empirical part of our research project. We do, however, describe the process implicitly, since we assume that a part of the organic carbon which leaves the model domain through the lower boundary will be transformed to methane, which, as it diffuses upward will be oxidised by sulphate and generate $H_2S$. Therefore, we parametrise this process by a conversion from sulphate to hydrogen sulphide at the lower boundary.

As in the water column, we distinguish between six different classes of detritus with different basic mineralisation rates.

**Table 4.** Reaction network table for the secondary redox reactions in the sediment and in the fluff layer.

| number | forward (backward) reaction | equation |
|---|---|---|
| 35 | p_fe2_ox_ihs | $Fe^{2+} + 4.5\,H_2O + 0.25\,O_2 \rightarrow 2\,H_3O^+ + Fe(OH)_3$ |
| 36 | p_ihs_red_iim, p_ihc_red_iim | $H_2S + 8\,Fe(OH)_3 \rightarrow 8\left(Fe^{2+} + 2OH^-\right)^{ads-clay} + 2\,H_2O + SO_4^{2-} + 2\,H_3O^+$ |
| 37 | p_ihs_red_ims, p_ihc_red_ims | $9\,H_2S + 8\,Fe(OH)_3 \rightarrow 8\,FeS + 18\,H_2O + SO_4^{2-} + 2\,H_3O^+$ |
| 38 | p_mn2_ox_mos | $Mn^{2+} + 0.5\,O_2 + 3\,H_2O \rightarrow 2\,H_3O^+ + MnO_2$ |
| 39 | p_ims_form2_pyr | $0.5\,H_3O^+ + 0.25\,SO_4^{2-} + 0.75\,H_2S + FeS \rightarrow FeS_2 + 1.5\,H_2O$ |
| 40 | p_pyr_oxmos_ihs | $1.25\,H_2O + 1.25\,H_3O^+ + FeS_2 + MnO_2 \rightarrow Mn^{2+} + Fe(OH)_3 + 1.625\,H_2S + 0.375\,SO_4^{2-}$ |
| 41 | p_pyr_oxo2_ihs | $4\,H_2O + FeS_2 + 0.25\,O_2 \rightarrow Fe(OH)_3 + 0.5\,H_3O^+ + 0.25\,SO_4^{2-} + 1.75\,H_2S$ |
| 42 | p_imm_oxo2_ihs | $0.25\,O_2 + \left(Fe^{2+} + 2OH^-\right)^{ads-clay} + 0.5\,H_2O \rightarrow Fe(OH)_3$ |
| 43 | p_i2i_oxo2_i3i | $0.5\,H_2O + 0.25\,O_2 + \left(Fe^{2+} + 2OH^-\right)^{in-clay} \rightarrow \left(Fe^{3+} + 3OH^-\right)^{in-clay}$ |
| 44 | p_aim_nit_no3_sed | $2\,O_2 + (NH_3)^{ads-clay} \rightarrow H_3O^+ + NO_3^-$ |
| 45 | p_fe2_mnox_ihs | $MnO_2 + 2\,Fe^{2+} + 6\,H_2O \rightarrow 2\,H_3O^+ + Mn^{2+} + 2\,Fe(OH)_3$ |
| 46 | p_h2s_mnox_so4 | $0.25\,H_2S + 1.5\,H_3O^+ + MnO_2 \rightarrow Mn^{2+} + 2.5\,H_2O + 0.25\,SO_4^{2-}$ |
| 47 | p_i3i_redh2s_i2i | $H_2S + 8\left(Fe^{3+} + 3OH^-\right)^{in-clay} -> 8\left(Fe^{2+} + 2OH^-\right)^{in-clay} + 2\,H_2O + 2\,H_3O^+ + SO_4^{2-}$ |
| 48 | p_ims_oxo2_ihs | $FeS + 2.25\,O_2 + 4.5\,H_2O \rightarrow SO_4^{2-} + Fe(OH)_3 + 2\,H_3O^+$ |

Details on the specific choice of the classes are given in Appendix B.

These rates are only controlled by temperature, not by the specific oxidant which is available. There is an ongoing controversy as to what determines the rate of sedimentary carbon decay, whether it is the oxidant (and therefore the accessible energy per mole of carbon) or the degradability of the detrital carbon itself (Kristensen et al., 1995; Arndt et al., 2013). In leaving out the explicit dependence of the oxidant, we do not favour the latter theory; we chose to adopt the decay rates proposed by Middelburg (1989), which may implicitly take the effect of the oxidant into account[13].

Sedimentary organic phosphorus (OP) may degrade faster than the corresponding nitrate and carbon, an effect known as preferential P mineralisation (Ingall and Jahnke, 1997). We include this by introducing additional state variables t_detp_n for each class n of detritus, describing the OP concentration, as well as a constant factor pref_remin_p which describes a redox-dependent ratio between the mineralisation speeds of OP and organic carbon and nitrogen. This factor is set equal to

---

[13]Middelburg's equation states that material which is decomposed later will be decomposed slower. This may be because the material itself is different, or because the oxidant is different. The Middelburg model includes both effects, and splitting them in a mechanistic model would mean preferring one theory or the other. So what we do assume if we just apply the Middelburg model is that the time which a particle spends in the oxic zone, the anoxic zone, and the sulphidic zone is similar in our setting to Middelburg's experiments. In this case, the Middelburg model will include the correct slowing-down of degradation caused by the less efficient oxidant.

**Table 5.** Reaction network table for adsorption/desorption and precipitation/dissolution processes in the sediment and in the fluff layer.

| number | forward (backward) reaction | equation |
|---|---|---|
| 49 | p_po4_ads_ips (p_ips_diss_po4) | $PO_4^{3-} + Fe(OH)_3 \leftrightarrow 3\,OH^- + FePO_4$ |
| 50 | p_fe2_prec_ims (p_ims_diss_fe2) | $2\,OH^- + H_2S + Fe^{2+} \leftrightarrow 2\,H_2O + FeS$ |
| 51 | p_fe2_prec_iim (p_iim_diss_fe2) | $2\,OH^- + Fe^{2+} \leftrightarrow \left(Fe^{2+} + 2\,OH^-\right)^{ads-clay}$ |
| 52 | p_ims_trans_iim (p_iim_trans_ims) | $2\,H_2O + FeS \leftrightarrow H_2S + \left(Fe^{2+} + 2\,OH^-\right)^{ads-clay}$ |
| 53 | p_mn2_prec_rho (p_rho_diss_mn2) | $1.6\,CO_2 + Mn^{2+} + 0.6\,Ca^{2+} + 4.8\,H_2O \leftrightarrow 3.2\,H_3O^+ + MnCO_3(CaCO_3)_{0.6}$ |
| 54 | p_po4_ads_pim (p_pim_lib_po4) | $PO_4^{3-} + 3\,H_3O^+ \leftrightarrow \left(PO_4^{3-} + 3\,H^+\right)^{ads-clay} + 3\,H_2O$ |
| 55 | p_nh4_ads_aim (p_aim_lib_nh4) | $OH^- + NH_4^+ \leftrightarrow H_2O + (NH_3)^{ads-clay}$ |

1 under oxic conditions and greater than 1 under anoxic conditions (Jilbert et al., 2011). This approach follows Reed et al. (2011).

### 2.7.2 Specific mineralisation processes

Here we describe the implementation of the primary redox reactions, indicated by the red numbers in Figure 3.

Oxic mineralisation and heterotrophic denitrification are formulated in the same way as in the water column, see Section 2.6.4.

The next pathway is the reduction of Mn-IV to Mn-II which produces dissolved manganese.

The reduction of iron oxyhydroxides should produce dissolved Fe-II. This, however, may precipitate very quickly, especially where hydrogen sulphide is present. So for numerical reasons, we combine these reactions, and the reduced Fe-III is directly

converted into iron monosulphide or considered as adsorbed by clay minerals, as we describe below in 2.7.3.

Some clay minerals, especially sheet silicates which are abundant in the German part of the Baltic Sea (Belmans et al., 1993), contain structural iron which is available for redox reactions (e.g., Jaisi et al., 2007). We prescribe a station-specific content of these minerals given in Table 7 and assume that they contain a small amount (0.1 mass-%) of reducible iron, because a particle analysis of sheet silicates from the area of interest (Leipe, unpublished data), showed slightly lower iron contents in

the sulphidic zone compared to the surface area.

The primary redox reaction follows process 32 in Table 3, we describe it in detail since it is a new process added to our model. Mineralisation of organic carbon under reduction of structural iron in sheet silicates takes place at a rate of

$$\texttt{p\_i3i\_k\_irred\_i2i} \quad = \quad \texttt{t\_sed\_k} \cdot r_k \cdot exp(\tau \cdot T) \cdot (1 - l_{o2}) \cdot (1 - l_{no3}) \cdot (1 - l_{mos}) \cdot (1 - l_{ihs}) \cdot l_{i3i} \,, \tag{19}$$

where $\texttt{t\_sed\_k}$ is the amount of detritus per area in a specific sediment layer, $r_k$ is a basic reactivity for this class (see Appendix B), $\tau = 0.15$ K$^{-1}$ is a temperature sensitivity constant for the mineralisation, and $T$ is the temperature in °C. The limitation functions $l_?$ are of Ivlev type, e.g.

$$l_{o2} \quad = \quad 1 - exp\left(\frac{-[O_2]}{O_{2,min}}\right) \,, \tag{20}$$

they have a value close to one at high concentrations of the corresponding oxidant and become zero if this oxidant is exhausted. Here, $O_{2,min}$ denotes a threshold concentration at which the limitation occurs, but the Ivlev function generates a soft transition between the different oxidation pathways, so they can occur simultaneously. The product of the last five factors in equation (19) means this process will run at a substantial rate only if

- oxygen concentration is low,

- nitrate concentration is low,

- manganese oxides in the sediment are depleted,

- iron oxyhydroxides in the sediment are depleted but

- reducible structural Fe-III in the clay minerals is still abundant.

Sulphate reduction produces hydrogen sulphide. As discussed above, it represents the terminal mineralisation process in our model. This process, described by processes 33 and 34 in Table 3, follows the formulation by Reed et al. (2011) in our model. Organic material leaving the lower boundary of our model because of sediment growth will also be mineralised by this process. We assume that a fraction of the buried material will be mineralised by either sulphate reduction or methanogenesis, the rest being buried. For the methane produced, we assume that it will not enter the model domain but rather be oxidised by sulphate, producing $H_2S$ below the model domain. We assume for simplicity that all these reactions happen instantaneously, which results in the same net reaction as the sulphate reduction.

### 2.7.3 Precipitation and dissolution reactions

Solids can precipitate from a solution when it becomes supersaturated. This happens in an aqueous solution when the actual ion activity product exceeds the respective solubility product and a critical degree of supersaturation is reached (e.g., Sunagawa, 1994; Böttcher and Dietzel, 2010).

Diagenetic models often simplify the calculation by multiplying the concentrations rather than the activities (e.g., van Cappellen and Wang, 1996). The resulting product is then proportional to the actual solubility product as long as the ionic strength

of the solution does not change. As the ionic strength of sea water is almost completely defined by its salinity (Millero and Leung, 1976), this assumption is well justified for most marine environments. The Baltic Sea, however, is a brackish sea with strong spatial and temporal changes of bottom salinity, especially in the western part (e.g., Leppäranta and Myrberg, 2009). For this reason, we take the activity coefficients, which transform concentrations to activities, into account. This is done by using

the Davies equation (Davies, 1938), which determines the individual activity coefficient $a_i$ as (Stumm and Morgan, 2012)

$$\log_{10}(a_i) \quad = \quad -Az_i^2 \left( \frac{\sqrt{I}}{1+\sqrt{I}} - 0.3 \cdot I \right), \tag{21}$$

where $I$ is the ionic strength expressed in mol l$^{-1}$, $z_i$ is the ion charge, and $A$ is the Davies parameter, calculated after Kalka (2018):

$$A \quad = \quad 1.82 \cdot 10^6 \, \mathrm{l}^{-0.5} \left( \frac{\epsilon(T,S)T_K}{1\,\mathrm{F\,K\,m^{-1}}} \right)^{-1.5}. \tag{22}$$

Here, $\epsilon$ is the dielectric constant of water calculated after Gadani et al. (2012)[14], and $T_K$ is the temperature in K.

    Ca-Rhodochrosite precipitates at elevated concentrations of manganese and carbonate. Its solubility product is composition dependent, as the Ca:Mn ratio varies (Böttcher, 1997; Böttcher and Dietzel, 2010). For Baltic Sea muds where ratios around 0.6 occur, an effective solubility product (including the effect of oversaturation) of $10^{-9.5}$ to $10^{-9}$ M$^2$ can be deduced from Jakobsen and Postma (1989). In our model, the reaction follows process 53 in Table 5. 10% of the dissolved manganese will

precipitate per day if the solution is undersaturated. Saturation is calculated by the formula (Jakobsen and Postma, 1989)

$$s_{rho} \quad = \quad \frac{a_2 \mathtt{t\_mn2} \cdot a_2 \left[ CO_3^{2-} \right]}{10^{-9.5}\,\mathrm{mol^2\,kg^{-2}}}, \tag{23}$$

where $\left[ CO_3^{2-} \right]$ is the concentration of carbonate ions which depends on DIC concentration and pH, see the description of the carbon cycle in Section 2.8. The term $a_2$ is the activity coefficient for ions with a charge of two, see eq. (21). The term in the denominator is the solubility product for rhodochrosite (Jakobsen and Postma, 1989).

If the solution becomes undersaturated, rhodochrosite will be dissolved again. Then, process 53 is reversed, and a fixed amount of $10^{-6}$ mol kg$^{-1}$ day$^{-1}$ of manganese-II is released until saturation is reached.

    Iron monosulphide precipitates on contact with dissolved Fe-II and sulphide, depending on pH, with a solubility product taken from Morse et al. (1987). But, as stated in Section 2.7.2, we assume for numerical reasons that this process takes place directly after Fe-III reduction. The solubility product is then used in an inverse way to determine the equilibrium concentration

of dissolved Fe-II at the current pH, sulphide concentration, and salinity:

$$a_2 Fe_{eq,ims}^{2+} \quad = \quad 10^{-2.95} \frac{a_1 [H_3 O^+]}{a_1 [HS^-]}, \tag{24}$$

where the $a_i$ are the Davies activity coefficients, see Eq. (21), and $10^{-2.95}$ mol l$^{-1}$ is the solubility product for iron monosulphide (Morse et al., 1987; Theberge and Iii, 1997).

    We then assume a precipitation or dissolution of iron monosulphide which relaxes the present concentration of Fe-II against

this equilibrium. This is in agreement with a pore water chemistry model for the central Baltic Sea (Kulik et al., 2000), which

---

[14]https://www.mathworks.com/matlabcentral/fileexchange/26294-calculated-dielectric-constant-of-sea-water/content/dielec.m

states that dissolved iron concentrations in the pore water are buffered by iron sulphides (mackinawite and greigite). The dissolution of iron monosulphide also takes place if clay minerals in the same grid cell are capable of adsorbing additional Fe-II. This process is described in Section 2.7.5.

As a simplification, we neglect the change in porosity which would be caused by precipitation (or dissolution) of any solids.

### 2.7.4 Pyrite formation

Pyrite ($FeS_2$) is a crystalline compound formed in early diagenesis (Rickard and Luther, 2007). Its formation from iron monosulphide is included in most early diagenetic models. This process is not a simple precipitation process, but rather a redox process. While both sulphide and iron monosulphide contain sulphur of oxidation state -2, the redox state of S in pyrite is -1. This implies that an electron acceptor is required to create pyrite. A generally accepted mechanism for pyrite creation is the use of zero-valent sulphur from polysulphides, this may be created by oxidation of sulphate with Fe-III. However, this process alone cannot explain the high degrees of pyritisation in Baltic deep sediments (Boesen and Postma, 1988).

An additional pathway which does not rely on elemental sulphur, but instead reduces hydrogen sulphide to hydrogen gas, has been proposed by Drobner et al. (1990), Rickard and Luther (1997) and Rickard (1997). Similar to how it was done in early diagenetic models (e.g., Wijsman et al., 2002), we include this pathway and therefore assume that whenever iron monosulphide and $H_2S$ are present, pyrite is formed from them. The generated $H_2$ will be consumed by sulphate-reducing bacteria (Stephenson and Stickland, 1931), so in the net reaction, sulphate acts as the electron acceptor.

In our model, the reaction therefore follows process 39 in Table 4. The speed of the transformation process is determined by

$$\texttt{p\_ims\_form2\_pyr} \quad = \quad \texttt{t\_ims} \cdot \texttt{t\_h2s} \cdot k_{pyr}, \tag{25}$$

where $\texttt{t\_ims}$ is the concentration of iron monosulfide, $\texttt{t\_h2s}$ is the concentration of hydrogen sulphide, and $k_{pyr}$ is a kinetic constant for conversion of iron monosulphide to pyrite. Its value of $8.9 \text{ kg mol}^{-1} \text{ day}^{-1}$ was adopted from Wijsman et al. (2002) and Rickard and Luther (1997) which use $8.9 \text{ l mol}^{-1} \text{ day}^{-1}$.

### 2.7.5 Adsorption balances

Adsorption in our model takes place on the surfaces of two particle types: iron oxyhydroxides and clay minerals. Adsorption on silicate particles is not explicitly represented in the model, but parametrised by a reduction of the effective diffusivity of phosphate and ammonium, following Boudreau (1997).

Iron oxyhydroxides adsorb dissolved phosphate. This is a well-known process responsible for the sedimentary retention of phosphate derived from mineralisation processes (e.g., Sundby et al., 1992). As both phosphate and hydroxide ions can occupy the adsorption sites at the surface, adsorption is less efficient in alkaline environments. In our model, we use a formula from Lijklema (1980) which describes the adsorbed P:Fe ratio at a given phosphate concentration and pH.

$$\frac{P_{ads}}{Fe} \quad = \quad 0.298 - 0.0316 \, pH + 0.201 \sqrt{DIP/1 \, \text{mmol l}^{-1}} \tag{26}$$

Here, $DIP$ gives the dissolved phosphate concentration. But we use it inversely. We calculate an equilibrium concentration for dissolved phosphate at the current P:Fe ratio and pH.

$$DIP_{eq} \quad = \quad \max\left(\frac{1}{0.201}\frac{P_{ads}}{Fe} - 1.483 + 0.157\,pH, 0\right)^2 \cdot 1\,\mathrm{mmol\,l^{-1}} \tag{27}$$

If the current concentration of dissolved phosphate is above this equilibrium concentration, adsorption takes place and $PO_4$ in the pore water is decreased. If it is below the equilibrium concentration, desorption takes place. The maximum function is added to treat situations when both $pH$ and the amount of adsorbed phoshate get so low that the formula by Lijklema (1980) gives no real solution for $DIP$. In this case, we assume that all currently dissolved phosphate will become adsorbed. The model processes `p_po4_ads_ips` and `p_ips_diss_po4` will change the phosphate concentration in the pore water by

$$\frac{\partial}{\partial t}\texttt{t\_po4} \quad = \quad \texttt{k\_ips\_dissolution} \cdot (DIP_{eq} - \texttt{t\_po4}) + \dots, \tag{28}$$

where `k_ips_dissolution`$= 0.1\,\mathrm{day^{-1}}$ is a reaction rate constant we assume. We chose this probably unrealistically low value for reasons of numerical stability.

Following Reed et al. (2011), we define two classes of iron oxyhydroxides. The first one is fresh, amorphous and adsorbs phosphate. The second one is a more crystalline phase, for which we assume no adsorption. The first phase is transformed to the second one with a constant rate in time, implying a continuous phosphate release.

Clay minerals, due to their large surface area, can also adsorb pore water species. We allow an adsorption of phosphate, ammonium and Fe-II. For simplicity, we assume that the ratio of adsorbed species to clay mass is proportional to the pore water concentration, until a saturation threshold is exceeded. For Fe-II, this proportionality constant is derived from Jaisi et al. (2007), for ammonium from Raaphorst and Malschaert (1996), and for phosphate from Edzwald et al. (1976). In all three cases, we calculate a pore water concentration which is in equilibrium with the current ratio of adsorbed species to clay mineral mass. Then adsorption or release processes take place to relax the present pore water concentration towards the equilibrium value.

To calculate $Fe^{2+}_{eq,clay}$, we first determine the mass of clay minerals present in a specific model layer per square meter. This is done by the formula

$$m_{clay} \quad = \quad \rho_{clay} \cdot \Delta z \cdot (1 - \phi) \cdot r_{clay}. \tag{29}$$

Here, $\rho_{clay} = 2.7 \cdot 10^3\,\mathrm{kg\,m^{-3}}$ gives the density of montmorillonite (Osipov, 2012), $\Delta z$ (m) gives the thickness of the layer, $(1 - \phi)$ gives the ratio between volume of the solids and total volume of the sediments, and $r_{clay}$ is the volume fraction of clay minerals among the total volume of solids. So, $m_{clay}$ has a unit of $\mathrm{kg\,m^{-2}}$. In the next step, we find out how much iron gets adsorbed to clay depending on the dissolved concentration. For Fe-II concentrations much smaller than $1\,\mathrm{mmol\,l^{-1}}$ as we observe them in our sediments, we can linearise the adsorption isotherms given by Jaisi et al. (2007) and obtain

$$Fe^{2+}_{ads,clay} \quad = \quad \alpha q_{max}^{mass}\left[Fe^{2+}\right]m_{clay}, \tag{30}$$

where $q_{max}^{mass}$ is a mass-specific sorption capacity ($\mathrm{mol\,kg^{-1}}$), $\alpha$ is a binding energy constant ($\mathrm{l\,mol^{-1}}$) and $\left[Fe^{2+}\right]$ is the concentration of iron in the pore water ($\mathrm{mol\,l^{-1}}$). We can rearrange the equation to obtain the equilibrium concentration of

dissolved Fe-II:

$$Fe^{2+}_{eq,clay} = \frac{Fe^{2+}_{ads,clay}}{\alpha q^{mass}_{max} m_{clay}} .$$ (31)

For the product $\alpha \cdot q^{mass}_{max}$, Jaisi et al. (2007) find values between 500 and 3000 l kg$^{-1}$ for different types of clay, which means 1 kg of clay added to 0.5 to 3 m$^3$ of water would adsorb the same amount of $Fe^{2+}$ as would remain in the solution. We adopt a value of 1000 l kg$^{-1}$ for our model.

For numerical reasons, we allow an immediate precipitation of the desorbed Fe-II as iron monosulphide in case of over-saturation, leaving out the intermediate transformation to dissolved Fe-II. The inverse is also true, if iron monosulphide is dissolved, the released Fe-II may directly be adsorbed by the clay minerals instead of being released to the pore water first. This is described by process 52 in Table 5.

For the adsorption isotherm of phosphate on clay minerals, we follow the study by Edzwald et al. (1976). They give maximum adsorption capacities $m_{P,ads,max}/m_{clay}$ between of 0.09 mg g$^{-1}$ for $P$ on Kaolinite to 2.58 mg g$^{-1}$ for Illite. These values were obtained at a pH close to 7.5, and pH dependence of adsorption differs between the different clay minerals. Since the composition of the clay minerals is unknown to us, we choose a conservative value of 0.2 mg g$^{-1}$, this could be adapted when such data are available. In contrast to Fe-II adsorption, a half-saturation of P adsorption is already reached at concentrations around 1 mg l$^{-1}$, which corresponds to approx. 0.03 mmol l$^{-1}$. We model this saturation in a very simple way by a linear dependency of dissolved and adsorbed phosphate below a threshold concentration of dissolved phosphate, and a constant amount of adsorbed phosphate if the threshold is exceeded:

$$PO^{3-,ads}_{4,eq} = \frac{1}{M_P} m_{clay} \frac{m_{P,ads,max}}{m_{clay}} \min\left( \frac{\texttt{t\_po4}}{PO^{3-,sat}_4}, 1 \right),$$ (32)

where $M_P$ = 31 g mol$^{-1}$ is the molar mass of phosphate, $m_{clay}$ is the mass of clay per square meter in the given grid cell, see Eq. (29), $\frac{m_{P,ads,max}}{m_{clay}} = 0.2$ mg g$^{-1}$ is the assumed maximum adsorption capacity, and $PO^{3-,sat}_4 = 0.03$ mmol l$^{-1}$ is the concentration of dissolved phosphate at which we assume this saturation is reached. We then define an adsorption and a desorption reaction following process 49 in Table 5. The adsorption process is assumed to happen instantaneously, but for numerical reasons we limit the process rate by demanding that at maximum (a) 10% of the dissolved phosphate is removed per day or (b) 10% of the lack of adsorbed phosphate with reference to the equilibrium concentration is precipitated. This artificial deceleration of the precipitation process had to be included to avoid numerical difficulties. The desorption process works in a similar way. On maximum, 10% of the adsorbed phosphate which exceeds the equilibrium concentration is released per day, or 10% of the saturation concentration $PO^{3-,sat}_4$, whichever is less.

For ammonium adsorption to clay minerals, the processes are in principle identical to those of phosphate. Since the adsorption is weak compared to that of phosphorus (in the range below 1 $\mu$mol g$^{-1}$, Raaphorst and Malschaert, 1996), we, however, neglect the effect by setting the maximum amount of adsorbable ammonium to zero in our present setup. So while the model is able to include the dynamics of ammonium adsorption to clay minerals, we make no use of it in the present application.

**Table 6.** Reaction network of secondary redox reactions in the sediment, giving the possible reoxidation processes in the presence of the oxidants listed in the first row.

| reoxidation by | $O_2$ | $NO_3^-$ | $MnO_2$ | $Fe(OH)_3$ |
|---|---|---|---|---|
| $NH_4^+ \rightarrow NO_3^-$ | + | | | |
| $NH_4^{+(ads-clay)} \rightarrow NO_3^-$ | + | | | |
| $Mn^{2+} \rightarrow MnO_2$ | + | | | |
| $Fe^{2+} \rightarrow Fe(OH)_3$ | + | | + | |
| $Fe^{2+(ads-clay)} \rightarrow Fe(OH)_3$ | + | | | |
| $Fe^{2+(in-clay)} \rightarrow Fe^{3+(in-clay)}$ | + | | | |
| $H_2S \rightarrow SO_4^{2-}$ | + | + | + | + |
| $FeS \rightarrow Fe(OH)_3 + SO_4^{2-}$ | (+) | | | |
| $FeS_2 \rightarrow Fe(OH)_3 + 2SO_4^{2-}$ | + | | + | |

### 2.7.6 Reoxidation of reduced substances

Reduced substances can be reoxidised if the appropriate oxidant is present in a sufficient concentration. Table 6 gives a summary of the redox reactions implemented in our model which will be described one by one in this section.

Ammonium is oxidised to nitrate in the presence of oxygen. The rate of this process is proportional to both the ammonium and the oxygen concentration and, as in the water column, increases exponentially with temperature.

Dissolved manganese-II will be oxidised in the presence of oxygen and precipitates as manganese oxide. This is also assumed to be a second-order process proportional to both precursor concentrations.

Dissolved Fe-II is oxidised by oxygen in a pH-dependent way. The rate of this process is proportional to the Fe-II and oxygen concentration, as well as to the square of the hydronium ion concentration. It is also influenced by temperature and ionic strength, as described by Millero et al. (1987). For numerical reasons, we also allow a direct oxidation of Fe-II adsorbed to clay minerals. Alternatively, dissolved Fe-II can be oxidised by reducing manganese. This process follows Reed et al. (2011). The generated Fe-III immediately precipitates as iron oxyhydroxide.

Structural iron in clay minerals can be reoxidised as well. We only allow this process in the presence of oxygen, when it transforms back to Fe-III, which is kept bound in the clay minerals.

This reaction follows process 43 in Table 4. The process runs at the speed of

$$\texttt{p\_i2i\_oxo2\_i3i} \quad = \quad \max(\texttt{i3i\_max} - \texttt{t\_i3i}, 0) \cdot \texttt{r\_i2i\_ox} \cdot l_{o2} \, . \tag{33}$$

The oxidation occurs only until the maximum amount of reducible Fe-III in the clay material, `i3i_max`, is reached. It occurs at a rate of `r_i2i_ox`$= 0.1$ day$^{-1}$, a somewhat arbitrary value indicating that the process is typically fast compared to the

vertical transport of clay minerals. It is Ivlev-limited by a factor $l_{o2} = 1 - exp\left(\frac{-[O_2]}{O_{2,min}}\right)$ with $O_{2,min} = 2.0 \cdot 10^{-5}$ mol kg$^{-1}$, consistent with the concentration at which carbon oxidation becomes limited.

Hydrogen sulphide can reduce any of the previously mentioned oxidants, being converted to sulphate. The reaction with oxygen or nitrate is carried out as a two-step reaction. The intermediate species formed in these reactions is elemental sulphur, which can be further oxidised to sulphate. These processes follow the same kinetics as in the water column, see Section 2.6.5. Hydrogen sulphide can alternatively react with manganese oxides or iron oxyhydroxides, producing dissolved Mn-II or Fe-II. For the generated Fe-II, we, however, assume either an immediate precipitation to iron monosulphide or an immediate absorption to clay minerals, whichever is more favourable. We assume these reactions to be proportional to both the concentration of sulphide and the metal oxides. Hydrogen sulphide can also reduce structural Fe-III in the clay minerals, the Fe-II will in this case remain in the clay.

This reaction follows process 47 in Table 4. The process runs at a speed of

$$\texttt{p\_i3i\_redh2s\_i2i} \quad = \quad \max(\texttt{i3i\_max} - \texttt{t\_i3i}, 0) \cdot \texttt{r\_i2i\_ox} \cdot l_{i3i}. \tag{34}$$

The model parameter $\texttt{r\_i2i\_ox}$ describes a relative speed of $0.1$ day$^{-1}$ at which $H_2S$ is reoxidised by this process, a somewhat arbitrary value just expressing our assumption that the process is fast compared to vertical transport of the clay minerals. The model shows low sensitivity to this rate parameter, as shown in the supplementary material. The process is Ivlev-limited by the factor $l_{i3i}$.

Iron monosulphide is typically not directly oxidised but dissolves at low sulphide concentrations. However, if it is exposed to oxygen, we assume a complete oxidation to Fe-III and sulphate.

Finally, pyrite can be oxidised in the presence of oxygen or manganese-IV, but in marine environments not by Fe-III (Schippers and Jørgensen, 2002). We assume a complete oxidation to sulphate and iron oxyhydroxides.

## 2.8 Carbon cycle

The carbon cycle in this model is included, following Millero (1995) and Dickson et al. (2007). Four parameters describe the state of the dissolved carbonate system in the water:

– pH

– total alkalinity (TA)

– dissolved inorganic carbon concentration (DIC)

– $CO_2$ partial pressure (pCO$_2$)

Knowledge of any two of them allows the determination of the other two parameters. We use TA and DIC as state variables. The reason for this is that both pH and pCO$_2$ can be changed by quick equilibrium reactions with a proton transition which occurs faster than our model time step allows, while TA and DIC cannot. For details on these reactions, see Dickson et al. (2007).

The DIC concentration can increase by mineralisation of organic carbon and decrease when DIC is assimilated by phytoplankton. Also, it can be modified by $CO_2$ exchange with the atmosphere. Calcification and carbonate dissolution are not considered in our model. Total alkalinity changes if acidic or alkaline substances are added or removed. The substances occurring in our model approach which change alkalinity are $OH^-$, $H_3O^+$ and $PO_4^{3-}$ ions. The effect of dissolved organic matter on total alkalinity (Kuliński et al., 2014; Ulfsbo et al., 2015) is neglected in the present model, it may be included in a future version.

The tracer value changes by 1 unit if (see Table 1)

– `ohminus` is changed by 1 unit or

– `h3oplus` is changed by -1 unit or

– `t_po4` is changed by 0.5 units.

As the pH (for adsorption and precipitation reactions) and $pCO_2$ (for gas exchange with the atmosphere) are of particular importance, we need to derive these from the state variables. This is done in an iterative procedure. Starting with a guessed pH value (from the last model time step), we aim to correct it until it is consistent with the given values of `t_alk` and `t_-dic`. To perform this correction, we calculate the fractionation of DIC into the different species ($CO_2$, $HCO_3^-$, $CO_3^{2-}$). From this, we determine a carbonate alkalinity as $\left[HCO_3^-\right] + 2\left[CO_3^{2-}\right]$, where square brackets denote a concentration. This can be determined by (Dickson et al., 2007)

$$A_{CO2} \quad = \quad \mathtt{t\_dic}\frac{k_{1,CO2}\left[H_3O^+\right] + 2k_{2,CO2}}{\left[H_3O^+\right]^2 + k_{1,CO2}\left[H_3O^+\right] + 2k_{2,CO2}} , \tag{35}$$

where $[H3O^+] = 10^{-pH}$ and $k_{1,CO2}$ and $k_{2,CO2}$ are the acid dissociation constants for carbonates as taken from Dickson et al. (2007). We do the same for other substances taking part in acid-base dissociation reactions (water, boron, sulphide, phosphate):

$$A_{H2O} \quad = \quad \frac{k_{H2O}}{\left[H_3O^+\right]} - \left[H_3O^+\right] , \tag{36}$$

$$A_{boron} \quad = \quad c_{boron}\frac{k_{boron}}{k_{boron} + \left[H_3O^+\right]} , \tag{37}$$

$$A_{H2S} \quad = \quad \mathtt{t\_h2s}\frac{k_{1,H2S}}{k_{1,H2S} + \left[H_3O^+\right]} , \tag{38}$$

$$A_{PO4} \quad = \quad \mathtt{t\_po4}\frac{-\left[H_3O^+\right]^3 + k_{1,PO4}k_{2,PO4}\left[H_3O^+\right] + 2k_{1,PO4}k_{2,PO4}k_{3,PO4}}{\left[H_3O^+\right]^3 + k_{1,PO4}\left[H_3O^+\right]^2 + k_{1,PO4}k_{2,PO4}\left[H_3O^+\right] + k_{1,PO4}k_{2,PO4}k_{3,PO4}} . \tag{39}$$

$$\tag{40}$$

The dissociation constants $k$ are taken from Dickson et al. (2007), and the total boron concentration is calculated from salinity as (Moberg and Harding, 1933)

$$c_{boron} \quad = \quad 0.000416\,\mathrm{mol\,kg^{-1}} \cdot \frac{S}{35\,\mathrm{g\,kg^{-1}}} , \tag{41}$$

where $S$ denotes salinity.

The sum of all their alkalinities should then match the known total alkalinity, but a difference occurs because the approximated pH was incorrect.

$$\Delta A \quad = \quad \texttt{t\_alk} - A_{CO2} - A_{H2O} - A_{boron} - A_{H2S} - A_{PO4} \tag{42}$$

So, we do a Newton iteration to find an improved pH estimate.

This is done by calculating the derivative

$$\frac{d\Delta A}{dpH} \quad = \quad \frac{d\Delta A}{d\left[H_3O^+\right]} \cdot \frac{d\left[H_3O^+\right]}{dpH} = \frac{d\Delta A}{d\left[H_3O^+\right]} \cdot (-\ln(10))\left[H_3O^+\right] \tag{43}$$

and obtaining the new $pH$ estimate as

$$pH^{new} \quad = \quad pH - \frac{\Delta A}{\frac{d\Delta A}{dpH}} \cdot \tag{44}$$

We use a fixed number of ten iteration steps for a better parallel performance of the code.

Finally, we can calculate $pCO2$ as

$$pCO2 \quad = \quad \frac{\texttt{t\_dic}/k_{0,CO2}}{1 + \frac{k_{1,CO2}}{[H_3O^+]} + \frac{k_{1,CO2}k_{2,CO2}}{[H_3O^+]^2}} \tag{45}$$

## 2.9 Numerical aspects

The equations which determine the temporal evolution of the state variables are solved by a mode splitting method, i.e. concentration changes due to physical and biogeochemical processes are applied alternately in separate sub-timesteps. For a discussion

of this method and alternatives we refer to Butenschön et al. (2012).

### 2.9.1 Numerics of physical processes

Vertical diffusion is done explicitly by multiplying each vertical tracer vector by a diffusion matrix. This includes turbulent mixing in the water column as well as pore water diffusion, bioturbation (faunal solid transport) and bioirrigation (faunal solute transport). This diffusion matrix is tridiagonal, and for a small time step, which is in our case limited by the thin layers at the

top of the sediment, a Euler-Forward method can be applied. Larger time steps could be split into smaller Euler-Forward steps, which means a repeated multiplication by the tridiagonal matrix. We instead use an efficient algorithm to calculate powers of the tridiagonal matrix (Al-Hassan, 2012), and perform the multiplication only once.

### 2.9.2 Numerics of biogeochemical processes

The sources and sinks for the different tracers are calculated from the process rates. These not only include biogeochemistry,

but also parametrisations for lateral transport processes as well as sedimentation and resuspension.

To calculate the changes of a tracer concentration with time, we form the sum of the processes consuming or producing it (Radtke and Burchard, 2015).

$$\frac{\partial}{\partial t} T_i = \sum_k p_k (q_{ik} - s_{ik}) .$$ (46)

Here $T_i$ represents the concentration of tracer $i$, $p_k$ is the rate at which process $k$ runs, and $q_{ik}$ (and $s_{ik}$) is the stoichiometric ratio in which process $k$ produces (or consumes) tracer $i$. In order to ensure both non-negativity of the tracer concentrations and mass conservation, we apply the positive Euler-Forward method from Radtke and Burchard (2015). It is a clipping method which, in the case where a tracer concentration becomes negative during one Euler-forward time step, it first executes a partial time step until this tracer is zero. Then the rest of the time step is continued without the processes consuming this tracer, i.e. they are switched off. More than two partial time steps may be needed if more than one tracer is exhausted.

## 2.10  Automatic code generation

The model code is not hand-written. Instead, the model is described in a formal way in terms of its tracers, constants and processes in a set of text files. The model code is then generated by a "code generation tool" (CGT) which fills in this information into a code template file. The advantage is that the same biogeochemical equations can in this way be integrated into different models. While the current version is written in Pascal, the three-dimensional version in MOM5 has been created as a Fortran code. The CGT is open-source software and can be downloaded at www.ergom.net.

## 3  Observed data used for model applications

We use four different observational datasets for model calibration and validation. The data used are (a) pore water profiles for different dissolved species, (b) sediment elemental composition, (c) estimates of bioturbation intensity and (d) bentho-pelagic fluxes measured in benthic chamber lander incubations.

## 3.1  Selected stations

All data were collected at seven different stations in the Southern Baltic Sea (see Fig. 1, we always present the stations from west to east). The mud stations LB and MB are situated in the Mecklenburg Bight, a trough-like bay in the south-western Baltic Sea where salinities are up to 20 g kg$^{-1}$. Stations ST and DS are on sandy substrate, the latter one is situated in only 22 m depth near the major sill which impedes the transport of the more saline North Sea water into the inner part of the Baltic Sea. Station AB is situated in the central Arkona Basin in 45 m depth. The Arkona Basin is the most western basin of the inner Baltic Sea. It accumulates organic matter not only from local primary production, but also laterally imported particles from coastal areas experiencing strong eutrophication, especially from the Odra River (Christiansen et al., 2002). Station TW is a silt station with a median grain size around 40 $\mu$m. The last station, the sandy station OB on the Oder Bank, is not a place of organic matter deposition, but rather of transformation before the detritus is transported to deeper locations. The Oder Bank is a shallow sandy area strongly influenced by the Odra river plume (Voss and Struck, 1997).

All the stations were sampled during twelve cruises which took place between July 2013 and January 2016 so as to cover different seasons (Lipka, 2018). However, not every station was sampled during every cruise. The calibration of the model occurred in parallel with the sampling campaign, such that only data from the first seven cruises (until January 2015) was used for model fitting.

## 3.2 Pore water analyses

Short sediment cores with intact sediment-water interfaces were taken by a multicorer, a device which simultaneously extracts 8 sediment cores from the sea floor. Pore water was extracted at different depths by rhizones. For a detailed description of the analytical methods used, we refer to Lipka et al. (2018b). Here we just give a short summary: ammonium concentrations were measured onboard using standard photometric methods (e.g., Winde et al., 2014). The quantification of major and trace elements was done on land, following the ICP-OES method (Kowalski et al., 2012). Dissolved inorganic carbon was measured by a mass spectrometer in the gas phase after a treatment of the pore water sample with phosphoric acid. Total alkalinity was determined colorometrically after Sarazin et al. (1999). Dissolved sulphide was determined spectrophotometrically by the methylene blue technique (Cline, 1969).

Instead of directly comparing sulphate concentrations between model and reality, which change over time with salinity, we use the sulphate deficit defined as

$$\Delta SO_4 = [SO_4^*] \frac{[K]}{[K*]} - [SO_4] , \tag{47}$$

where $[SO_4]$ and $[K]$ are the measured concentrations of sulphate and potassium (the latter regarded as a passive tracer) in the pore water and $[SO_4^*]$ and $[K^*]$ are their typical concentrations in sea water of 35 g/kg salinity (Dickson and Goyet, 1994).

## 3.3 Sediment composition

Parallel sediment cores from the same multicorer casts as used for the pore water analysis were subsampled in 1 cm steps, freeze-dried under vacuum and homogenised for geochemical analyses. Total carbon (TC) as well as nitrogen (TN) and sulphur (TS) contents were measured by combustion, chromatographical separation of the released gases and their determination with a thermal conductivity detector. The total inorganic carbon (TIC) content was measured by acidic removal of carbonates and analysis of the released CO2 with a nondispersive infrared detector. The total organic carbon (TOC) content was then calculated by the subtraction of TIC from TC values. At the sand stations (ST, DS and OB), the mass fractions were measured in the fine fraction (< 63 $\mu$m) of the sediment only, assuming that the coarse fraction does not contain these elements in a significant amount. Thus, the percentage in the whole sediment was calculated by multiplying with the fine fraction ratio that was determined by laser diffractometry. Analytical details related to the devices used, their calibration as well as precision and accuracy can be obtained from Bunke (2018).

## 3.4 Bioturbation intensity estimates

In order to analyse bioturbation intensities ($D_B$), six to 24 cores per station were sliced onboard immediately after retrieval at 0.5 cm intervals to 3 cm depth and at 1 cm intervals to 10 cm for vertical chlorophyll-a (Chl-a) profiles. All samples were deep-frozen (-18°C) and stored until extraction (Sun et al., 1991). In the laboratory, the defrosted sediment samples were homogenised and three parallel subsamples of 1 cm$^3$ volume were taken from each slice. After adding 9 ml of 96% ethanol and an incubation period of 24 h in the dark, the samples were centrifuged at 4000 rpm for 5 minutes, measured photometrically (663 and 750 nm) and chlorophyll was calculated based on HELCOM (1988). The vertical chlorophyll profiles were interpreted using the bio-mixing model developed by Soetaert et al. (1996b). Experimentally derived chlorophyll decay constants of 0.01 d$^{-1}$ for mud and 0.02 d$^{-1}$ for sand (Morys, 2016) and an artificially small sedimentation rate $\omega$ of 0.00001 cm d$^{-1}$ were used, the latter just reflecting the fact that chlorophyll decay is much faster than sedimentation. The model applied may distinguish between diffusive and non-diffusive mixing. The latter mode of bioturbation was neglected in our study despite the fact that it may be the dominant particle transport process in certain areas (e.g. AB).

## 3.5 Bentho-pelagic fluxes

Total oxygen uptake (TOU) and bentho-pelagic nutrient fluxes ($NO_3^-$, $NH_4^+$, $PO_4^{3-}$) of the sediments were measured in situ with two identical benthic lander systems "Mini Benthic chamber" by courtesy of S. Sommer and P. Linke (GEOMAR Kiel, Germany) during cruises AL434, EMB076, POS475, EMB100 and EMB111 (Lipka, 2018). All systems were equipped respectively with a Plexiglas® chamber (diameter 19 cm), an electronically driven glass syringe water sampler and oxygen optodes (Aanderaa Instruments, Norway, No. 4831) in- and outside of the chamber. Oxygen concentration trend was used as the main parameter for gas fluidity and tightness. Each respective chamber covers a sediment area of 284 cm$^2$ and the chamber water volume was in the range of 5-8 l depending on sediment penetration depth. Incubation times at the seafloor ranged from 9 to 48 h. Discrete chamber water samples were gathered by up to eight glass syringes (volume approx. 45 ml) in intervals of 1-7 h, depending on total deployment time. Photo lights (SolaDive 1200) and a photo camera (GoPro Hero Black Edition 3+) were used for the observation of the chamber deployment, particularly to check for sediment disruption. The start of incubation was defined if initial concentrations inside the chamber at the beginning were close to bottom-water concentrations. Nutrient concentrations were measured with a QuAAtro multianalyser system (Seal Analytical, Southampton, UK) onboard using standard photometric methods (Grasshoff et al., 2009). Nutrient fluxes were calculated from linear increase or decrease of concentration versus time, corrected for the surface area to volume ratio of each chamber. A robust linear regression method which is tolerant to outliers was applied (Huber, 1981; Venables and Ripley, 2002), and the uncertainties of the fluxes were obtained by an ordinary bootstrapping approach (Canty and Ripley, 2017; Davison and Hinkley, 1997). The regression analysis to determine their fluxes was performed in the same way as for the nutrients. TOU was calculated by standard linear regression of $O_2$ concentration versus time (with R$^2$ values above 0.98), within a period while $O_2$ concentration did not sink below 15% of the initial concentration (Glud, 2008). Calibrations of $O_2$ optodes were performed in ambient sea water aerated for 30 min

**Table 7.** Porosity and sediment accumulation rate data used as model input and clay volume content estimated by the model based on an initial guess.

| station | porosity (rel. units) | sed. acc. rate (kg m$^{-2}$ year$^{-1}$) | source of sed. acc. rate | clay minerals content (% of volume) |
|---|---|---|---|---|
| LB | 0.91 | 0.6 | (Kersten et al., 2005) | 0.5 |
| MB | 0.91 | 0.3 | (Leipe et al., 2011) | 0.5 |
| ST | 0.40 | - | - | 0.04 |
| DS | 0.40 | - | - | 0.1 |
| AB | 0.91 | 1.1 | (Emeis et al., 2002) | 0.5 |
| TW | 0.60 | - | - | 0.1 |
| OB | 0.40 | - | - | 0.05 |

(100% atmospheric saturation) and in saturated seawater - sodium dithionite solution (0% oxygen), regularly cross-checked by Winkler titration (Winkler, 1888).

## 4  Model setup and optimisation

There are three ways in which observations feed into our model:

1. model constants which were derived in earlier studies and which our model adopted from previous models,

2. initial and boundary conditions, determining tracer concentrations at the beginning and throughout the model run, and

3. calibration data which help to confine uncertain model parameters during a repeated model calibration process.

### 4.1  Use of data as model constants

Most of the observations which help constrain our model processes enter our model indirectly, since model constants are inherited from ancestor models. Especially in van Cappellen and Wang (1996), a thorough confinement of model constants based on observations was achieved. We add to that by supplying site-specific observations for porosity for all seven stations, set to a homogeneous value per substrate type estimated from measurements within the SECOS Project (Lipka et al., 2018a). The assumption of a homogenous value is a first approximation which is motivated by the future aim to use the model in a three-dimensional context. While detailed spatial maps of surface porosity exist, vertical profiles are rare. The effect of this simplification is discussed in Appendix D. Sediment growth estimations for the three muddy sites are taken from different sources, as shown in Table 7.

## 4.2 Initial and boundary conditions

The initial conditions for most biogeochemical state variables in the water column are taken from the previous run of a three-dimensional ERGOM model as described in Section 2.2 (Neumann et al., 2017), which contained a simplified sediment model as described e.g. in Radtke et al. (2012). Concentrations of sulphate and calcium were set to salinity-determined values as described by the standard composition of sea salt (Turekian, 1968). Dissolved dinitrogen was initialised at 100% saturation (Hamme and Emerson, 2004).

In contrast, fluff and sediment were initialised empty. We allowed them to fill up with material derived from the water column during the simulated period of 100 years. While this period of 100 years is not sufficient to fill the considered 22 cm of sediment by accumulation, it is sufficient to almost reach a steady state in the pore water concentrations. While the sixth class of detritus, which is considered non-biodegradable, continues accumulating in the sediments after 100 years, those classes which affect the pore water concentrations decay on smaller time scales.

Since the model conserves nitrogen and phosphorus, the filling of the sediments would have led to a depletion in the water column. To overcome this, we relax the winter concentrations of dissolved inorganic nitrogen and phosphorus (DIN and DIP) against values obtained from the previous 3-d model run. This relaxation is applied every winter, so the nutrients required to fill the model domain are provided from an artificial external source. Their input is large at the beginning of the model run and decreases over time as the sediment reaches a state which is almost in equilibrium with the organic matter supply from the water column above.

## 4.3 Model fit to observed data

Pore water profiles from Lipka (2018) were then used to calibrate the model. The calibration included optimising model parameters for individual stations, as well as parameters for the whole model domain. Typically, this type of calibration is done manually by the modeller. But due to the large number of parameters to optimise (115 in total), we decided to do a systematic, algorithm-based optimisation.

Please note that the physical input data and initial conditions used during the model optimisation phase were taken from a preliminary, unpublished 3-d model run. It differs from the cited model version (Neumann et al., 2017) by using a less realistic light model. Furthermore, dissolved organic nitrogen is not included as a state variable. These improvements were made to the ERGOM model during the development phase of the sediment model, and the results of the sediment model show only small differences between the model versions. We use the data from the final, published 3-d model run for the results shown in this article, for reasons of reproducibility. The preliminary forcing data used during the calibration phase are, however, also given in the online supplementary material.

### 4.3.1 Penalty function

The first step in such an optimisation is to define a metric or a penalty function quantifying the misfit between model and observations. Our aim is then to minimise this function.

We chose to penalise the relative deviation between model and measurement and define the penalty function by

$$
P \quad = \quad \sum_{i=1}^{i_{max}} \sum_{j=1}^{j_{max}(i)} r_{i,j}^2 \cdot w_i \, .
\tag{48}
$$

Here, $i_{max}$ is the number of state variables we compare and $j_{max}(i)$ is the number of observations of this state variable at all depths and at all stations. The expression $r_{i,j}$ is a measure for the relative deviation between model value $m_{i,j}$ and observation $o_{i,j}$,

$$
r_{i,j} \quad = \quad \log_{10}\left(\frac{m_{i,j} + \Delta_i}{o_{i,j} + \Delta_i}\right) \, .
\tag{49}
$$

The term $\Delta_i$[15] is included to avoid huge relative errors between values which are close to zero. It denotes the random deviation in this parameter, quantified from duplicate or triplicate measurements of the same parameter in different sediment cores of the same sampling. Obviously $r_{i,j}$ becomes zero if model and observations match. Finally, the weight $w_i$ assigned to each comparison in equation (49) is defined as the ratio

$$
w_i \quad = \quad \frac{\overline{o}_{i,j}}{\Delta_i}
\tag{50}
$$

between the average observed value of the variable and the random deviation. The weight is applied to make sure that fitting the most certain variables have the highest priority.

The pore water species which are fitted are: ammonium, phosphate, silicate, sulphide, iron, manganese, the total alkalinity, and the relative sulphate deficit[16].

### 4.3.2 Optimisation strategies

After we defined a penalty function, the second step is to choose an algorithm to minimise it. Several such algorithms exist, however, our choice of methods was restricted by the relatively long runtime of a single model iteration. Since it took about 8 minutes to run a single station for 100 years, we had to choose methods which

1. needed a relatively small number of iteration steps and therefore

2. allowed for a high degree of parallelism in the individual optimisation step, in order to effectively search the 115-dimensional parameter space.

Our first choice was the Adaptative Hierarchical Recombination - Evolutionary Strategies (AHR-ES) algorithm, implemented in the R-Package calibraR (Oliveros-Ramos and Shin, 2016). We were, however, not satisfied with the optimisation result. Possibly we just failed to find out the optimal settings for the algorithm, such as the survival rate.

Therefore, our second choice was a simple alternative algorithm: our own extension of the Generalised Pattern Search (GPS) algorithm (Hooke and Jeeves, 1961). Every optimisation step consists of two sub-steps. The first substep is the most simple

---

[15]This term differs per variable, but is the same at each station and sampling depth.
[16]defined as $\Delta SO_4 [K^*] / ([SO_4^*][K])$, see eq. 47

"grid search" step in which all 115 parameters are varied by a predefined step width. We run 230 sets of 7 models in parallel, such that each parameter can be both increased and decreased. In the second substep, 230 combinations of the most successful changes are formed. Even if no single-parameter change could improve the existing solution, sometimes combinations of them can, which is the basic idea of GPS. The parameter vector with the best score which was obtained in any of the two substeps is then chosen as the starting point for the next step. If none of the two sub-steps lead to an improvement of the overall fit, the step size is reduced by a factor of 2.

The optimisation converged after 30 iteration steps and reduced the error function from 6363 (the value obtained by previous manual tuning) to 4797.

The algorithm obviously does not guarantee that we reach a global optimum, which can be seen as a drawback. The automatic method was started after manual calibration of the model. Since the optimisation method is deterministic, the local optimum is defined by this initial condition. However, in a vector space with a dimension as high as ours, it is anyway difficult to find a non-local point with a better score, no matter if it is by manual optimisation or a different search algorithm.

## 4.4 Manual correction of sand and silt stations

For the sand stations and the single silt station, the automatic optimisation resulted in an unrealistic set of parameters. The bioturbation rates were estimated as low as those of the mud stations. However, at these low bioturbation rates, the sediments failed to accumulate realistic amounts of organic matter. The pore water profiles we obtained, however, seemed to match relatively well with the observations. This was due to the fact that the realistically low concentrations of solute species were obtained by an unrealistically low incorporation of degradable particulate material into the sediments. The model assumed relatively high rates of lateral removal of fluff, such that only a small fraction of the locally produced detritus was actually processed in the sediments.

This illustrates the problem that if the diffusivity is unknown, very different transports can be caused by the same pore water gradient. We therefore decided to manually modify the solution. This modification meant raising bioturbation and bioirrigation intensity by a factor of 10 at each station. Afterwards we reduced the parameter `r_fluffy_moveaway` which describes the rate at which fluff layer material is transported to the deeper areas until realistic concentrations in the pore water profiles were reached.

This led to similar pore water profiles, but higher turnover rates and organic content in the sediments.

## 5 Model results and validation

### 5.1 Comparison to measured pore water profiles

In this section, we compare and discuss observed and simulated pore water profiles of several chemical species relevant for early diagenetic processes. Model results are taken from the last year of the 100-year simulation, which was driven by a repeated forcing every year. After this simulated period, the model almost reached a quasi-steady state, which means the annual cycle

of pore water concentrations was nearly repeated year after year at each of the stations. The sixth class of detritus, in contrast, which we defined as non-degradable, did not reach a stable concentration, but continued to accumulate in the sediment during the period of 100 years, but this continued accumulation did not influence the pore water profiles due to the fact that it was assumed to be bioinert.

### 5.1.1 Pore water profiles at mud stations

Figure 4 shows a comparison of simulated and measured pore water profiles at the three mud stations.

In the left panels, we see that the rise of alkalinity with depth is captured well by the model, except for the AB site where observations show a higher alkalinity below 10cm depth. The decline in sulphate follows the lower range of the observations.

The panels in the second column show that also the vertical profiles of ammonium and silicate are represented relatively well by the model. However, especially at the Arkona Basin station, the observed range of both ammonium and sulphide shows a strong variation (by an order of magnitude). The model does not capture that but rather sticks to the lower range of the observations. Most probably, the variability in the observations is not due to seasonality, but a consequence of spatial variability between sampling sites, since the samples were taken from two sites 23 km apart.

Surprisingly, the model is able to reflect the differently steep sulphide profiles between the stations LB and MB. While Lipka (2018) see the low sulphide concentrations which occur especially in March 2014 at MB as an indication for a preceding mixing event, our model cannot adopt this interpretation due to its limitation by a temporally constant vertical mixing. In contrast, our model suggests a higher deposition of iron oxyhydroxides at this site.

The right panels show that the modelled manganese concentrations match the observations quite well. The dissolved iron profiles show their maxima at the correct depths and a relatively large seasonal spread. The measurements show an even larger spread than the model. For the phosphate profiles, the model results mostly resemble the lowest of the measured values, except for station AB where we see a clear underestimation. This can be seen as an artefact of our fitting method, more precisely of the choice of our penalty function. Giving a penalty for the relative error means that the same absolute error is punished heavier if the observation is smaller, making the model try harder to fit low values compared to high ones.

The model results for the mud stations fit quite well, considering the fact that the real pore water profiles may be shaped by very different temporal variations. These include, for example, mixing events, changing loads of organic matter or temperature and salinity variations. Our model, not knowing the sediments' past, can only try to estimate the average conditions that might produce similar pore water concentrations.

### 5.1.2 Pore water profiles at sand stations

Figure 5 illustrates the model fit at the sandy stations.

All of the sandy stations have one major error in common: sulphide concentrations are strongly overestimated at depths below 5 cm. We suppose that the precipitation or reoxidation of sulphide is underestimated. For all other pore water species, the agreement between measured and modelled ranges is reasonable. Especially the rise of alkalinity with depth is captured

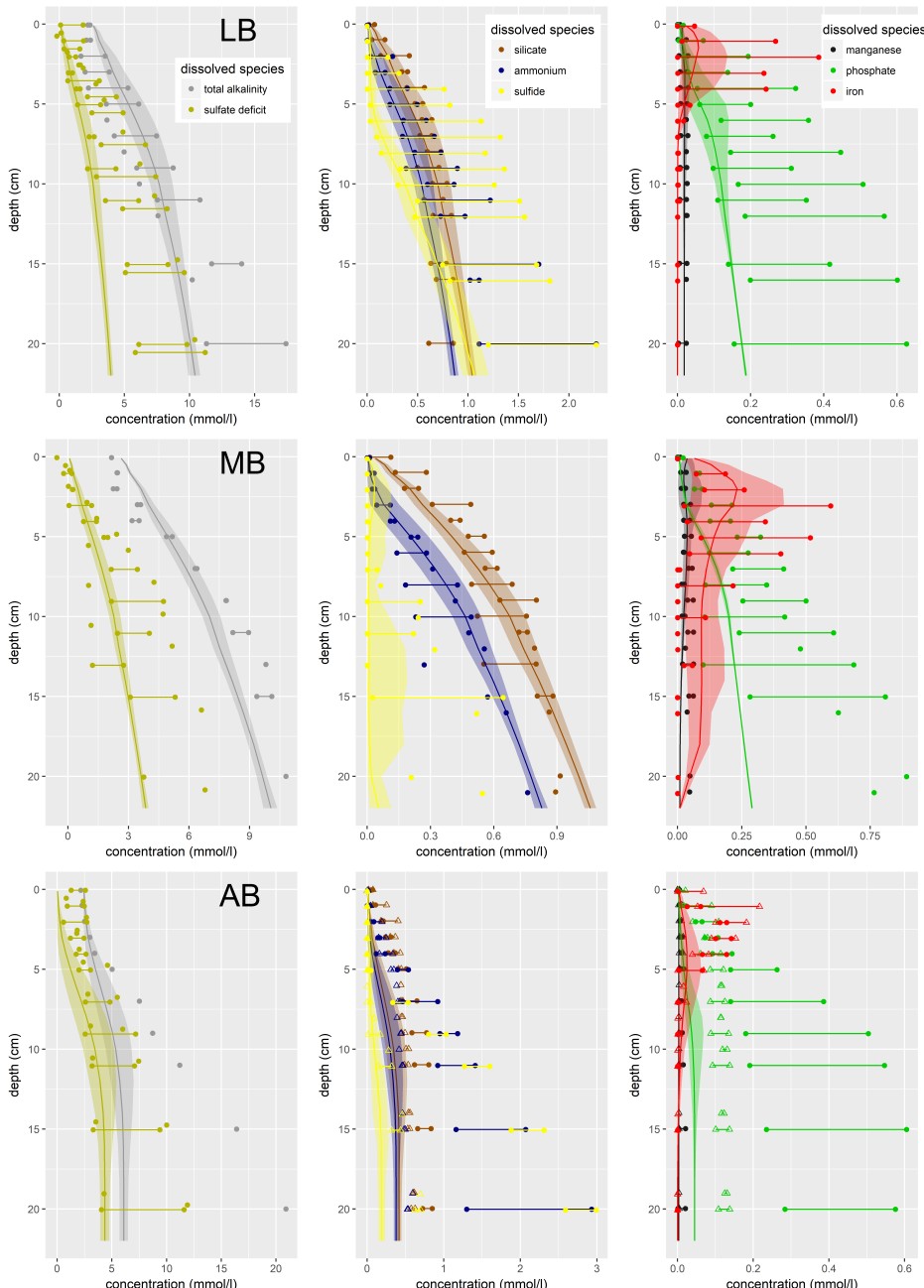

**Figure 4.** Pore water concentrations of several dissolved species at the three mud stations Lübeck Bight (top row), Mecklenburg Bight (middle row) and Arkona Basin (bottom row). Points and horizontal lines indicate the range of measurements. For station AB, empty triangles indicate the range of observations from the January 2015 cruise, which were taken at a different location in the same area, 23 km apart. Curves and shading present the model results and indicate year-average concentrations and the seasonal range. Please note the different horizontal scales.

well by the model. The sulphate deficit in the empirical data has a large uncertainty, as it is calculated as a small difference of similarly large quantities..

In our model, the sandy sites show a more pronounced seasonal cycle in the pore water profiles compared to the muddy stations. Especially iron and manganese concentrations vary considerably due to the seasonally different supply of quickly degradable organic matter and correspondingly differences in mixing intensity. While the variability in the supply of fresh organic matter is captured by the model, the variation in mixing is not. Still, the simulated ranges are supported by the variability in the observed pore water concentrations.

### 5.1.3 Pore water profiles at the silt station Tromper Wiek

For the station Tromper Wiek, we used data from two different cruises, in April and June 2014. Even if the idea in the SECOS project was to repeatedly sample the same station, the locations were approximately 6 km apart for this station, and the substrate type at the station sampled in April was sand rather than silt. The amount of sulphide in the pore waters showed a large difference between the April and the June cruise, the latter concentrations exceeding the former by a factor of 20. This reflects spatial rather than temporal variations. Some of the depth intervals were only sampled during the June cruise, which explains the different observed ranges at the different depths.

The good agreement in the profiles of ammonium and phosphate (middle and right panel in Fig. 6) suggests that total mineralisation is captured well. The left panel shows that the model estimates the sulphate deficit at the lower range of observations, but the rise of alkalinity with depth at the higher range. The model overestimates the vertical extent of Fe-II in the pore waters. However, the model reasonably reproduced the range of iron concentrations, and also the fact that dissolved manganese concentrations were always low compared to those of iron.

### 5.2 Comparison to sediment composition estimates

In Figure 7, we compare the composition of the solid parts of the sediment between model and measurements.

For the mud stations LB and MB, the modelled element concentrations show a quantitative agreement with the measurements. The main difference is that the measured values show strong vertical fluctuations, which may be the result of the deposition history. Another difference is that the vertical gradients of sulphur are considerably steeper in the model than in reality. In the mud station AB (Arkona Basin), however, the actual concentrations of all three elements are heavily underestimated. Nonetheless, the depth gradients of the concentrations match quite well, so there is perhaps just a constant offset. This might be caused by the accumulation of bioinert organic material, possibly of terrigenous origin from the Odra river.

In all sand stations (ST, DS, OB), the amount of sulphur in the sediments is underestimated. The observed sulphur in the sediments varies with depth and shows a maximum at around 10 cm depth. The fact that sulphide, in contrast, was overestimated in the pore waters, suggests that the precipitation of sulphur may be underestimated in the sandy cores. Particulate N and TOC are present in realistic quantities at the OB station. At the other two sand stations, the N and TOC observations show maxima at the top (station DS) or bottom (station ST) of the profile, which are not captured by the model. These are most likely the traces of past sedimentation or bioturbation events.

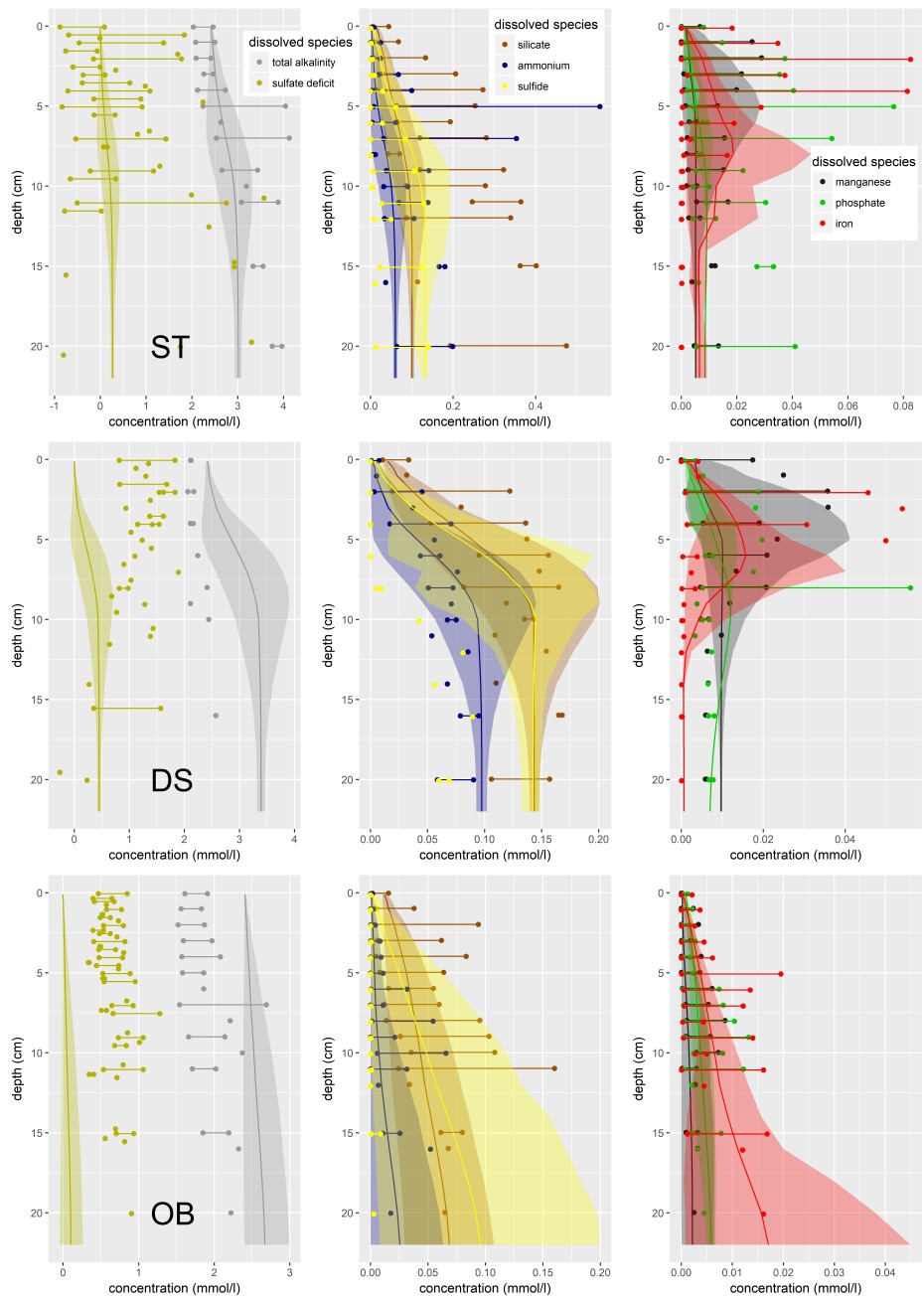

**Figure 5.** Pore water concentrations of several dissolved species at the three sand stations Stoltera (top row), Darss Sill (middle row) and Oder Bank (bottom row). Points and horizontal lines indicate the range of measurements. Curves and shading present the model results and indicate year-average concentrations and the seasonal range. Please note the different horizontal scales.

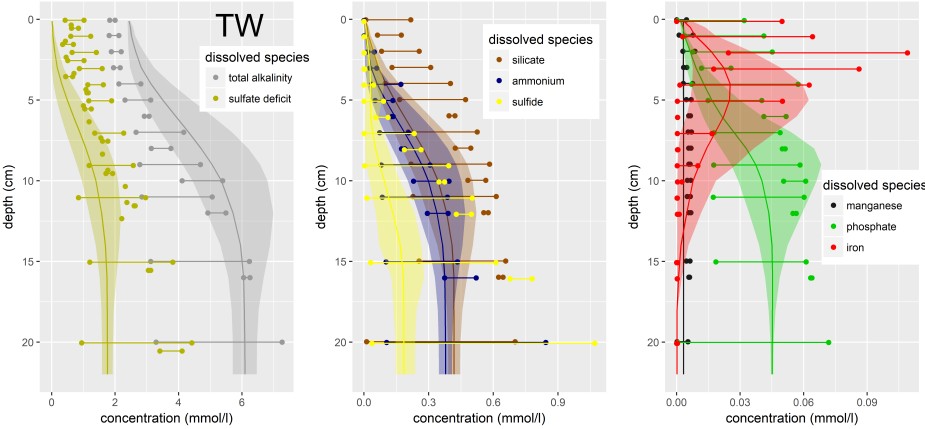

**Figure 6.** Pore water concentrations of several dissolved species at the silt station Tromper Wiek. Points and horizontal lines indicate the range of measurements. Curves and shading present the model results and indicate year-average concentrations and the seasonal range.

Reproducing subsurface TOC maxima, as they occur in permeable sediments, represents a challenge for early diagenetic models. They can be caused by different processes, such as

– non-local, fauna-driven ingestion of fluff material into a specific depth,

– washout of organic material from the surface sediment e.g. during storm events or

– lateral relocation of sediments.

### 5.3   Comparison to measured bioturbation intensities

The empirically estimated bioturbation intensities span a large range at each station. A reason for this may be that while our model assumes a temporally constant bioturbation, in reality it is highly variable. Mixing events by animals or shear stress alternate with periods without mixing (Meysman et al., 2008). Investigations of individual cores can only give snapshots of

this highly variable mixing rate.

In Figure 8a, we compare measured bioturbation diffusivities $D_B$ to those used in the model. Since the observed ranges are very large, they almost always contain the value assumed in the model. An exception is the Tromper Wiek site where exceptionally high $D_B$ values were measured. This may, however, be an artefact based on the method calculating the diffusivities. While we assume diffusion-analogue mixing in the model, also non-local mixing occurs in nature. The two processes can be

distinguished from the analysis of Chl-a profiles (Soetaert et al., 1996b; Morys et al., 2016). Figure 8b shows how often the samples from a specific site supported the hypothesis of diffusion-analogue mixing. For the station TW, it was only one fourth of the sediment cores that could be explained assuming local mixing, so non-local mixing was identified as a major process here. $D_B$ values were only calculated for cores where the observed Chl-a profiles could be explained by local mixing alone.

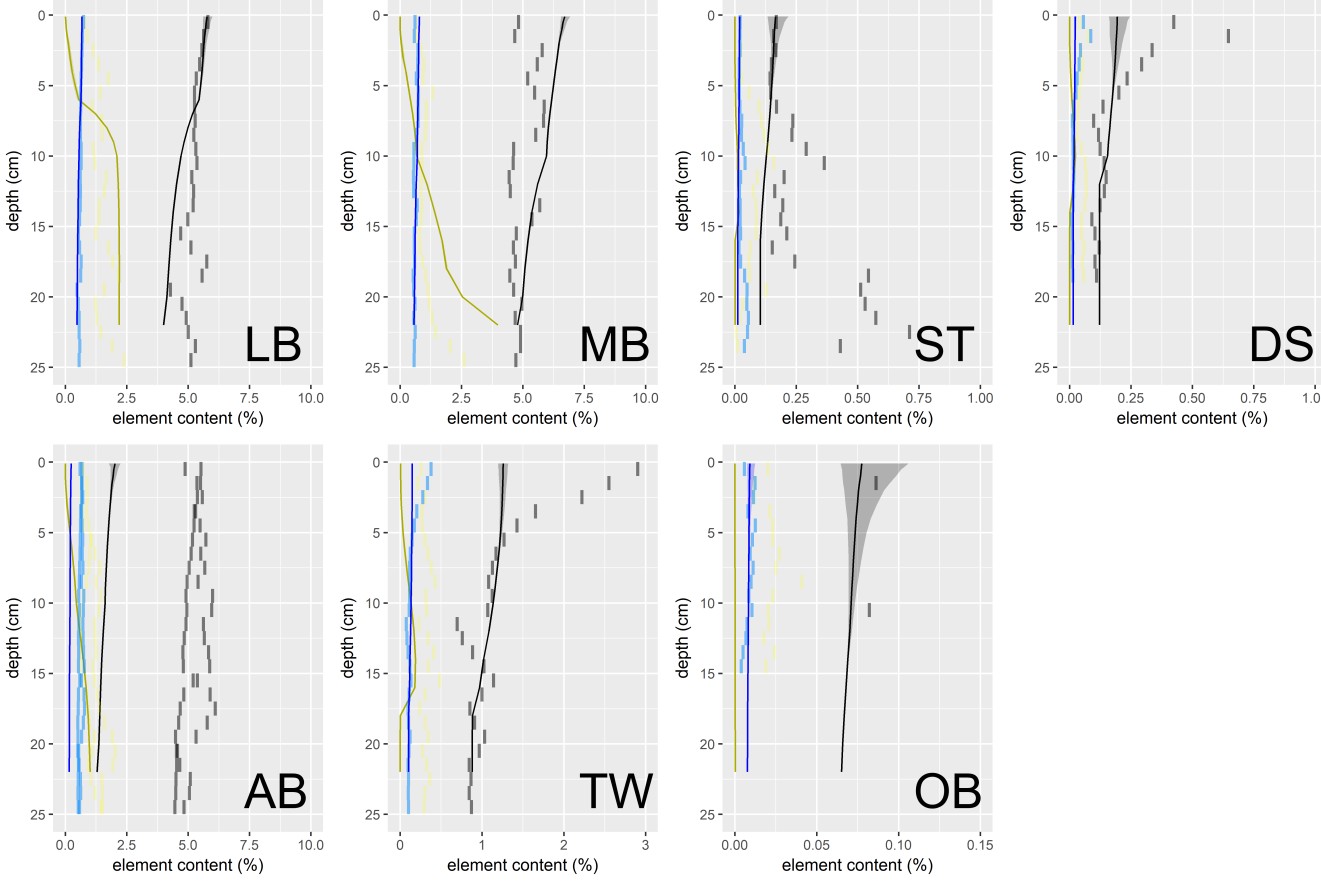

**Figure 7.** Mass fractions of nitrogen (blue), sulphur (yellow) and organic carbon (black) in the dry sediment, model results (curves and shading for seasonal range) versus measurements (vertical segments). Please note that the scales on the horizontal axes differ by a factor of 40.

The automatic model calibration yielded diffusivities at the sand and silt stations which were as low as those at the mud stations. Such weak mixing, however, could not supply enough organic matter to the sediments to reach measured element compositions. Therefore, they were corrected upwards, resulting in higher mixing at the sandy than at the muddy sites. This agrees with recent estimates of the bioturbation potential (Gogina et al., 2017; Morys et al., 2017), an index describing the ability of macrofauna to displace the sediment particles, resulting in a mixing effect. This potential was estimated to be higher in the more shallow sandy areas than in the muds. Also, measured bioturbation rates in the SECOS project were higher in the sand than in the mud (Morys, in prep.). However, the high variability of bioturbation rates within stations makes it difficult to prove a significant difference in $D_B$ between sandy and muddy areas empirically (Morys et al., 2016).

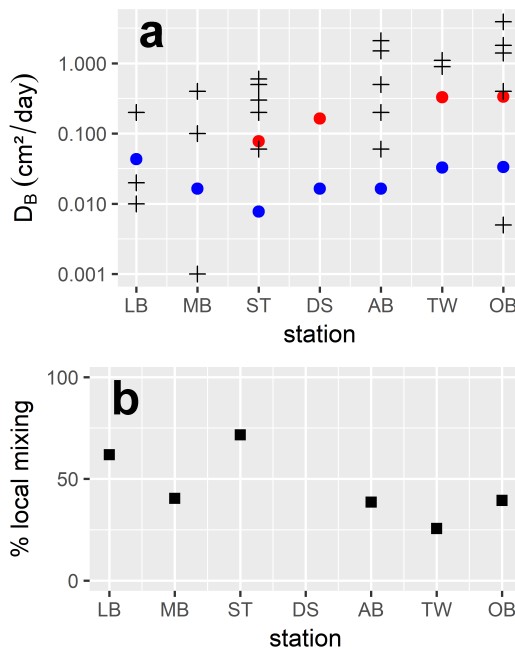

**Figure 8.** (a) Model estimations of bioturbation intensity (blue dots), manually corrected for sand and silt stations (red dots) and Chl-a-based estimates (black crosses), data from Morys et al. (2016). Black crosses represent averages over all cores from a single month where the Chl-a profiles in the sediment support the assumption of diffusion-analogue mixing. So, variation can be interpreted as temporal variability. (b) The percentage of the cores at this station whose Chl-a profiles could be explained by the assumption of a local, diffusion-analogue mixing process.

## 5.4 Comparison to measured bentho-pelagic fluxes

The net fluxes of selected pore water species ($O_2$, $NH_4^+$ and $PO_4^{3-}$) into the sediment or out of it are shown in Figures 9 and 10 for mud and sand/silt stations, respectively. An additional figure showing the fluxes of DIC, $NO_3^-$, $Fe$, $Si$ and $SO_4^{2-}$ is given in the online supplement. The figures compare modelled fluxes to observations from benthic chamber lander incubations.

5  For each of the selected species, we get two contributions in the model: the flux into or out of the sediment itself (by diffusive and by bioirrigation) and a consumption or production by mineralisation of the fluff layer material. We can distinguish between these in the model, but not in the measurements, because (a) the benthic chamber lander measures concentration changes in the bottom water only and (b) at the mud stations, the border between sediment and fluff layer is rather a smooth transition than a discrete boundary.

10  The comparison of annual-average oxygen fluxes between model and measurements shows a reasonable quantitative agreement. Taking the rather high fluctuations in the measurements into account, we cannot assume a perfect fit. The model correctly reproduces the fact that similar oxygen consumption occurs at sand and mud stations in spite of their order-of-magnitude differences in organic content and pore water concentrations, (Boudreau et al., 2001). The strong seasonality in the model, with $O_2$

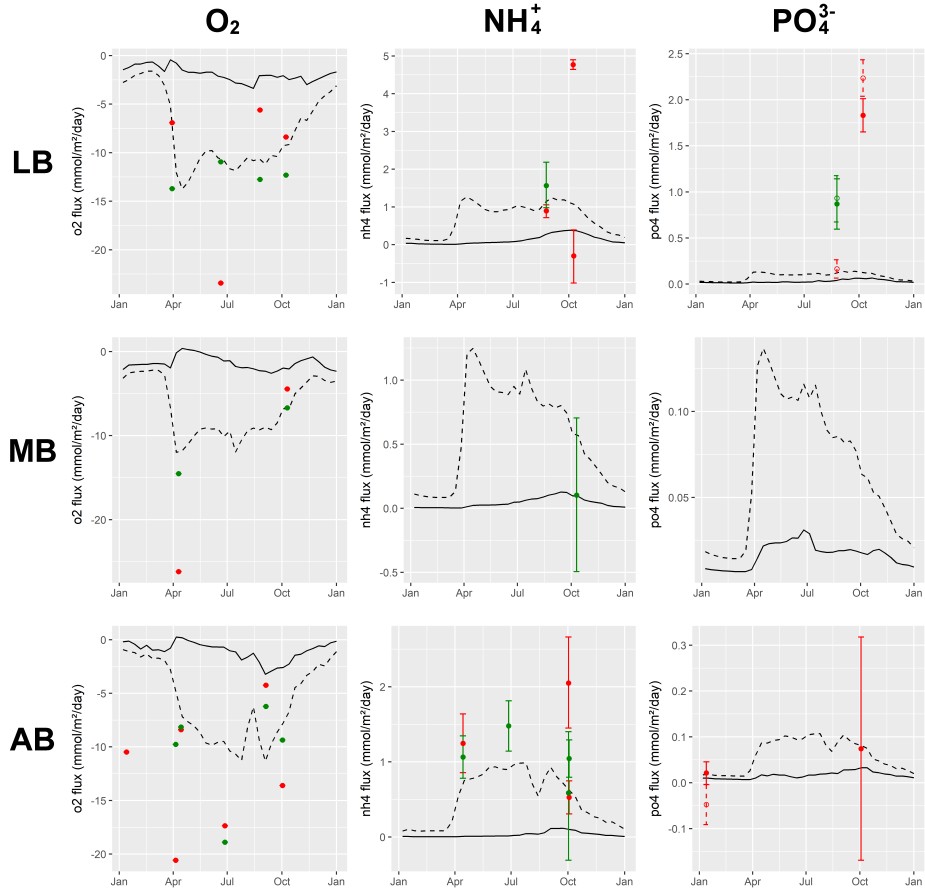

**Figure 9.** Fluxes between sediment and bottom water of selected pore water species at mud stations. Positive values denote fluxes out of the sediment. Solid line: Modelled fluxes between sediment and bottom water only. Dashed line: Fluxes including mineralisation of the fluff layer material. Dots: Measured fluxes by two benthic chambers (BC1 in red and BC2 in green). Vertical ranges: Uncertainties of these fluxes estimated by a bootstrapping method. For phosphate: full circles - estimates based on phosphate determination by photometric methods, empty circles - estimates based on P quantification by the ICP-OES method.

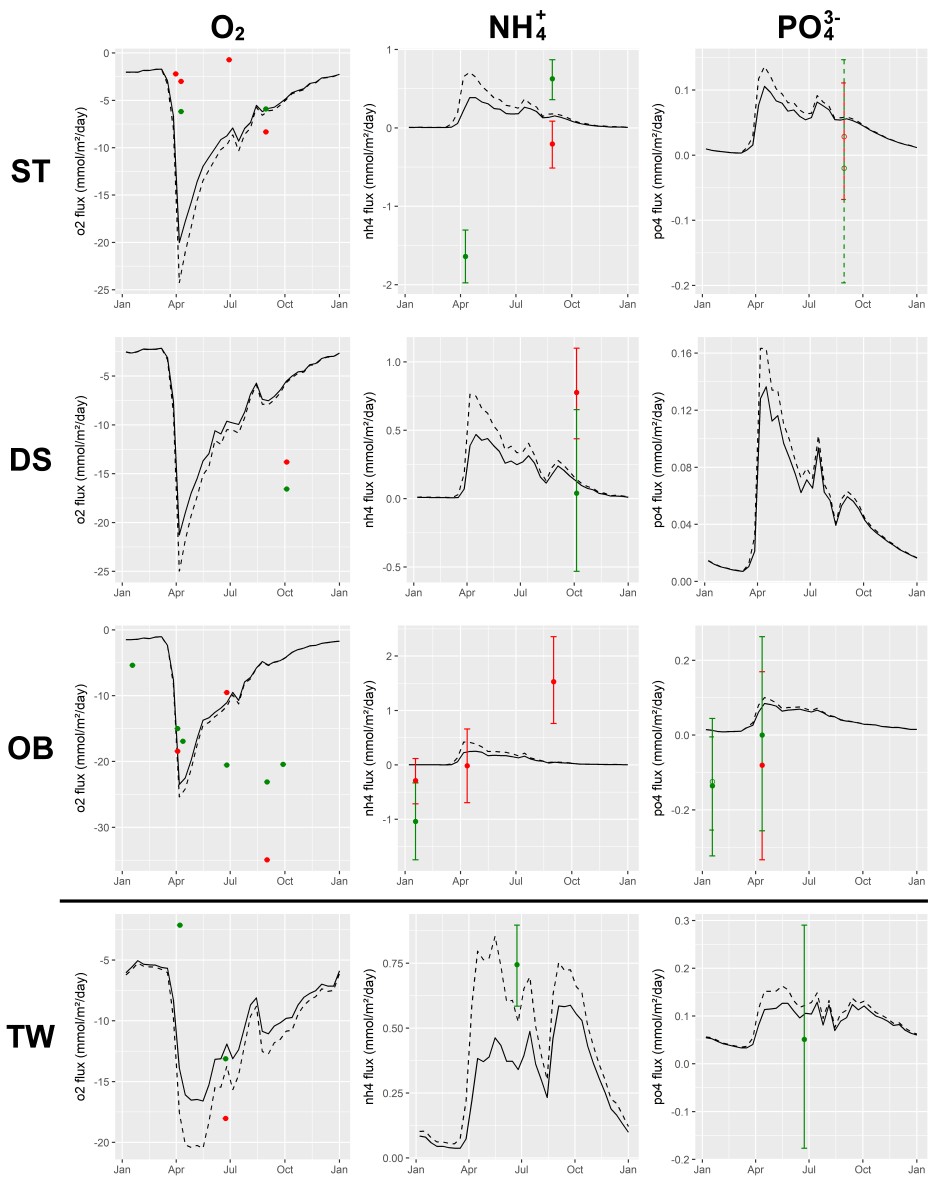

**Figure 10.** Fluxes between sediment and bottom water of selected pore water species at sand stations. Positive values denote fluxes out of the sediment. Solid line: Modelled fluxes between sediment and bottom water only. Dashed line: Fluxes including mineralisation of the fluff layer material. Dots: Measured fluxes by two benthic chambers (BC1 in red and BC2 in green). Vertical ranges: Uncertainties of these fluxes estimated by a bootstrapping method. For phosphate: full circles - estimates based on phosphate determination by photometric methods, empty circles - estimates based on P quantification by the ICP-OES method.

consumption being high during summer and low during winter, cannot be confirmed by the measurements, but we need to state that only two valid measurements exist during January and February when the modelled consumption rates are lowest. For the nutrients, we find highly variable fluxes in the observations, some of which show large relative uncertainties. The model results are in reasonable quantitative agreement with these fluxes, but again the clear seasonality in the model cannot be confirmed

empirically. Also the peak values of the observed fluxes (more than 4 mmol m$^{-2}$ day$^{-1}$ for $NH_4^+$ and 2 mmol m$^{-2}$ day$^{-1}$ for $PO_4^{3-}$) are much larger than those in the model. Our model also does not show ammonium or phosphate fluxes directed into the sediments, as they do occur in a small fraction of the observations. So, we can state that the modelled fluxes are more smooth than the observed ones. This may either reflect reduced spatiotemporal variability in the model, or artificial variability introduced into the measurements by the sediment disturbance which our incubation method causes.

## 5.5  Scope of model applicability and model limitations

In this paper, we applied our model in a one-dimensional context. The aim was to reproduce early diagenetic processes taking place in the sediments at seven exemplary sites thought to be representative for the south-western Baltic Sea by a mechanistic model. In our fully coupled model, the pelagic biogeochemistry and an assumed lateral transport supplied the organic material which drove the early diagenetic processes in the sediments. A comparison to a variety of different observations showed that the

model gives a reasonable reconstruction of sediment biogeochemistry. Still, we found differences in the details. For example, a strong overestimation of sulphide concentrations in sandy sediment pore waters most likely points to the underestimation of sulphide precipitation/reoxidation.

The analysis we show suggests that the processes most relevant for these observations are adequately represented in the model. This does not include all parts of the model. For example, the nitrogen cycle was not compared to observations, which

is due to the fact that the project SECOS in which this work was done did not focus on it and so the required observations of nitrification or denitrification rates are missing.

The ultimate aim of the model is its application in a fully coupled three-dimensional framework. A fully coupled pelagic and benthic model could answer a wide range of questions, e.g.

   – Are the strongly simplifying sediment parametrisations which we use in marine ecosystem models today consistent with

our understanding of sediment biogeochemistry, or is there a mismatch between our assumptions in the pelagic models and the sediment-water fluxes in early diagenetic models, which are directly constrained by observational data?

   – How might sedimentary services such as nutrient removal change under different conditions, and what feedbacks into pelagic biogeochemistry can be expected?

   – On which time scales can organic material stored in the sediments affect the eutrophication status of the pelagic ecosys-

tem, e.g. for how long will sedimentary nutrient release counteract nutrient abatement measures aimed at reducing the winter nutrient concentrations in the water column?

The applicability of the one-dimensional model is limited. There is little added-value in using this coupled benthic-pelagic model compared to a classical early diagenetic model, since in most cases a one-dimensional description of a pelagic ecosys-

tem will be strongly oversimplified. One could, however, imagine that it can be useful for enclosed marine areas where the horizontal exchange is limited or well-known. An application to a different area than the south-western Baltic Sea will, however, require a new model calibration, since critical parameters like bioturbation intensity might differ. We strongly discourage the use of the model as it is by just applying it to derive estimates on benthic biogeochemical process rates from pelagic biogeochemistry, unless there is a large set of benthic data available against which the model can be validated.

In cases where these data are available, we think that the model system has a high potential to serve as a starting point for detailed studies, because it can be easily modified. Adding, removing or adapting processes is very easy because of the automatic code generation principle. Only a formal mathematical formulation of the process is required, and no coding skills are needed to e.g. add additional state variables to the model system. Also a re-use of parts of the model, e.g. the explicit representation of the fluff layer, is possible.

## 6   Conclusions

In this manuscript, we describe an integrated model for ocean biogeochemistry. It simulates ocean biogeochemistry both in the water column and in the sediments.

The model was obtained by combining two ancestor models, the water column model ERGOM (Neumann et al., 2017), and the early diagenetic model used in Reed et al. (2011). A few modifications were made to the existing models, partly to include additional processes relevant for the area of interest, the south-western Baltic Sea. These model extensions include

- closing the carbon cycle in the sediments which allows the determination of pH,

- adding a specific numerical scheme for the diffusion of the tracer "total alkalinity",

- using ion activities rather than concentrations to determine precipitation and dissolution potentials, allowing us to account for salinity differences,

- the explicit description of adsorption to clay minerals, considering their mineralogy, and

- an alternative pyrite formation pathway via $H_2$ formation.

An automated model calibration approach was used to fit the model to pore water observations at seven sites in the study area. It was successful for the mud stations, but underestimated bioturbation rates and consequently the organic content of the sediment at the sand and silt sites. Therefore, these model parameters were adjusted manually at the sand and silt sites. This issue illustrates a general problem related to models of this complexity. The large quantity of unknown model parameters results in many degrees of freedom, and different types of observations are needed to constrain them. Even so, a good fit to a constrained set of observations does not guarantee that the model dynamics are captured realistically.

Applying the model in a three-dimensional framework (Cahill et al., in prep.) will reduce the degrees of freedom. For example, our model includes parametrisations for (a) lateral removal of fluff material from the sand stations and (b) lateral import of organic material at the mud stations. In a 3-d ocean model, these become intrinsically linked by the constraint of

mass conservation. Other degrees of freedom arise from the supply of oxidised iron and manganese to the individual stations. In a 3-d model, the supply and distribution of these substances would be controlled by erosion and deposition and thus determined by the model physics.

Apart from these constraints, the implementation of the model in a 3-d framework is straightforward. Physically, the coupling between different locations would be controlled by the fluff layer, its erosion and redeposition. Technically, the coupling is simplified due to the use of automatic code generation. Describing the model processes and constants in a formal way, keeping them separate from code for specific models, means it is easy to switch between different "host models". The major difficulty in going 3-d is the limited amount of validation data, such as pore water profiles and sediment-water fluxes, compared to the strong spatial and temporal variability. A first step is the application of the model to the limited area of the German EEZ for which the model is calibrated.

In the long term, biogeochemical ocean models should aim at a process-resolving description of surface sediments. This is especially true for shallow ocean areas where the efflux of nutrients from the sediment strongly influences water column biogeochemistry, like in our study area. The magnitudes of denitrification and phosphate retention, or the spatial and seasonal patterns in which oxygen consumption occurs, may strongly influence marine ecosystems.

Very often, model studies discussing "what if"-scenarios use a relatively simple sediment representation. This includes studies on nutrient abatement, human-induced stresses on ecosystems (e.g. by fish farming) or climate sensitivity analyses. But the use of a present-day parametrisation for future scenarios means a neglection of possible changes. In the context of limited data and process understanding, this implicit "no change"-assumption may be the best we can presently do. But we should be aware of the uncertainty which is introduced by this pragmatic choice. Studying the sensitivity of sediment functions to external drivers in a process-resolving sediment model can be a way to quantify these uncertainties, and possibly derive an ensemble of alternative future parametrisations.

*Code and data availability.* A source code version of the model is provided in the supplement to this article. It includes the initial conditions and physical forcing files required to reproduce the obtained results.

The code is not hand-written, but can be generated automatically from a set of text files describing the model biogeochemistry, and a code template containing the physical and numerical aspects of the model code. All three ingredients required to obtain the model source code (the text files, the code template, and the code generator program) are also included.

These components in their current and previous versions are GPLv3 licensed and can also be downloaded from our website www.ergom. net.

For the calibration and validation data used in this study, we refer to the following publications: the pore water data can be found in Lipka (2018); the sediment composition data are published in Bunke (2018); the bioturbation rate estimates are available in Morys (2016).

## Appendix A: Stoichiometric composition of model state variables

The stoichiometric composition of the model tracers is shown in Table A1.

## Appendix B: Rates of organic carbon mineralisation

The study by Middelburg (1989) relates decay rates of organic carbon to the time since the organic material was deposited. They found the following relation to be valid across time scales from days to decades:

$$\ln\left(\frac{k}{1\,\text{year}^{-1}}\right) = -0.95\ln\left(\frac{t}{1\,\text{year}}\right) - 0.81\,, \tag{B1}$$

where $k$ denotes the reactivity of organic carbon and $t$ is the time since detritus was created. In our model, we try to resemble this relation by splitting the detritus into different reactivity classes. The ratios $r_k$ of the different classes $k$ in freshly created detritus and their reactivities at $0°C$ are listed in Table A2.

We assume a faster detritus mineralisation at higher temperatures. This is controlled by a factor $\exp(T\cdot\tau)$ which we multiply with the decay rates, where $T$ is the temperature measured in $°C$ and $\tau = 0.15\,\text{K}^{-1}$ is a temperature sensitivity constant as it was used in previous versions of the ERGOM model already (Neumann and Schernewski, 2008). The effective decay rate of a quantity of fresh detritus changes over time, since the ratio between the detritus classes $k$ changes as the quickly-decaying fractions are removed faster. In Fig. A1, we illustrate the effective decay rates predicted by our model at different temperatures and compare them to the rates predicted by the Middelburg formula. We can see that (a) the class model gives a good match to the formula and (b) temperature has little influence on the overall decay rate. The latter fact can be understood as already explained in the main text: a higher temperature means a higher decomposition rate of each detritus class. This will in the sum be compensated for by a shift in the class composition. Lower concentrations of quickly-degradable detritus classes will remain, which compensates the faster decay of the less degradable classes. This means an overall very similar total decomposition rate.

## Appendix C: Quantitative influence of different model extensions

Here we use a set of sensitivity experiments to illustrate how the model refinements introduced by us influence the results. In each of these, we switch off one of our model improvements. This means that we use three simplified model versions, in which

1. total alkalinity always diffuses with the bicarbonate diffusivity, no matter how many hydroxide ions contribute to it which, in reality, diffuse faster,

2. the saturation indices for precipitation/dissolution reactions are calculated neglecting the (salinity-dependent) activity coefficients, and

3. the adsorption of ammonium, phosphate and iron onto clay minerals, as well as their reducible Fe-III content, are neglected.

As an example, we apply these reduced models to the silt station TW. (Please note that the calibration procedure was not repeated after the model modifications, but the model parameters were left unchanged.)

The results are shown in Fig. A2 in dashed lines and compared to the full model. All modifications affect the dissolved concentrations of iron in different directions. This is probably because both pore water pH and the activity coefficients influ-

ence the precipitation to iron monosulphide. The second modification (neglecting activity coefficients) reduces the phosphate concentrations in the pore water. All other pore water species remain virtually unchanged by our model modifications.

## Appendix D:  Sensitivity analysis for vertically varying porosity

We used vertically constant porosity in our application of the model. Here we illustrate the effect of this simplification by comparison to a model with a realistic porosity profile, see Fig. A3. A porosity profile was measured at station Tromper Wieck during the April 2014 cruise (Lipka, 2018) and interpolated to the model depths. Below approx. 3 cm depth, where porosity is decreased in the realistic profile, we see enhanced pore water concentrations for the nutrients and for sulphide. This was to be expected because of the higher ratio between solid material and pore water volume. In contrast, iron concentrations are reduced in higher depths while manganese concentrations remain constant. While we do not observe qualitatively different behaviour, the differences between simplified and realistic-porosity model are significant, which means the model might benefit from using realistic porosity profiles.

## Appendix E:  Sensitivity analysis for vertical resolution in the sediments

In Fig. A4, we show the pore water profiles at the site Tromper Wiek, simulated with the original and with double resolution in the sediments. For numerical stability reasons, a time step for vertical diffusion of oxygen had to be reduced in the high-resolution run. Apart from that, the runs were identical. It can be seen that most of the vertical profiles look practically identical between the setups. For iron, phosphate and sulphide, we can see deviations which we, however, considered as acceptable if we keep the overall uncertainties in mind.

## Appendix F:  Numerical details of applying the Al-Hassan method

For solutes, we assume that the flux (positive upward) between two neighbouring cells takes the form

$$F_{k,k+1}^{solutes} \;\;=\;\; D\frac{\rho(c_{k+1} - c_k)}{l_{k,k+1}} \,, \tag{F1}$$

where $D$ is the effective diffusivity calculated as the sum of molecular diffusivity and bioirrigation diffusivity, see Eq. (5), $c_k$ denotes the concentration in the pore water of cell $k$ [mol/kg], and $l_{k,k+1}$ is the distance between the centres of the adjacent cells. This flux means a concentration change in the different grid cells given by

$$\frac{d}{dt}c_k \;\;=\;\; \frac{F_{k,k+1}^{b,solutes} - F_{k-1,k}^{b,solutes}}{\rho\Phi\Delta z_k} \tag{F2}$$

This allows us to construct a Matrix $M_B$ which transforms the vector of concentrations $(c_0, c_1, ..., c_{kmax})^T$ to its derivative, that is,

$$M_B \boldsymbol{c} \;\;=\;\; \frac{d}{dt}\boldsymbol{c}. \tag{F3}$$

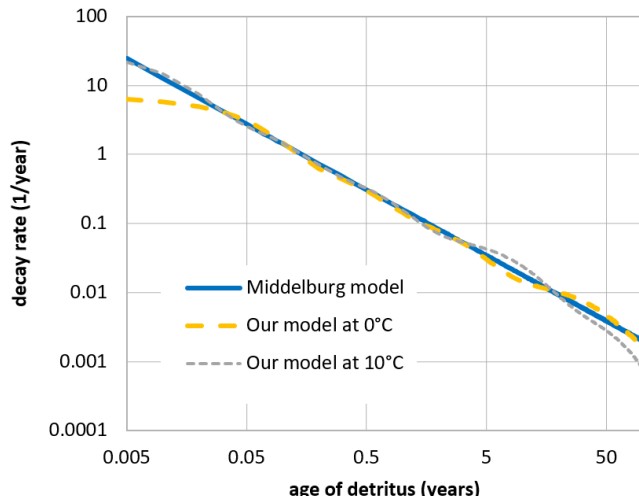

**Figure A1.** Decay rates of organic carbon in detritus depending on its time of creation. Comparison of the reactivity predicted by our model at different temperatures to the Middelburg decay rate prediction, see text.

So, after a time step of $\Delta t$, the concentration vector $\boldsymbol{c}^{new}$ can be determined as

$$\boldsymbol{c}^{new} \quad = \quad \exp(M_B \Delta t)\boldsymbol{c}. \tag{F4}$$

The matrix exponentiation can be defined as

$$\exp(M_B \Delta t) \quad = \quad \lim_{n \to \infty} \left( I + \frac{M_B \Delta t}{n} \right)^n. \tag{F5}$$

We approximate the limit by choosing $n = \Delta t/1$ s and use the method of Al-Hassan (2012) to compute $\boldsymbol{c}^{new}$. This method allows an efficient calculation of powers of tridiagonal matrixes with positive entries, and it is easy to see that $I + \frac{M_B \Delta t}{n}$ is of this shape if $n$ is chosen large enough.

An identical approach is used for the solids.

*Competing interests.* No competing interests are present.

*Acknowledgements.* This study is embedded in the KÜNO Project SECOS (03F0666A) funded by the German Federal Ministry for Education and Research (BMBF). The model optimisation runs were performed on the HLRN supercomputing facilities. Free software which supported this work includes R/RStudio and Freepascal Lazarus. Marko Lipka and Michael E. Böttcher wish to thank I. Schmiedinger, A. Köhler and I. Scherff for their invaluable help in the laboratory. They also wish to express their gratitude to B. Liu for helpful discussions on transport processes. Jana Woelfel and Gregor Rehder thank Peter Linke and Stefan Sommer for the long-lasting loan and technical support

(Sergiy Cherednichenko) of the two Mini Benthic chamber landers (GEOMAR, Kiel, Germany).

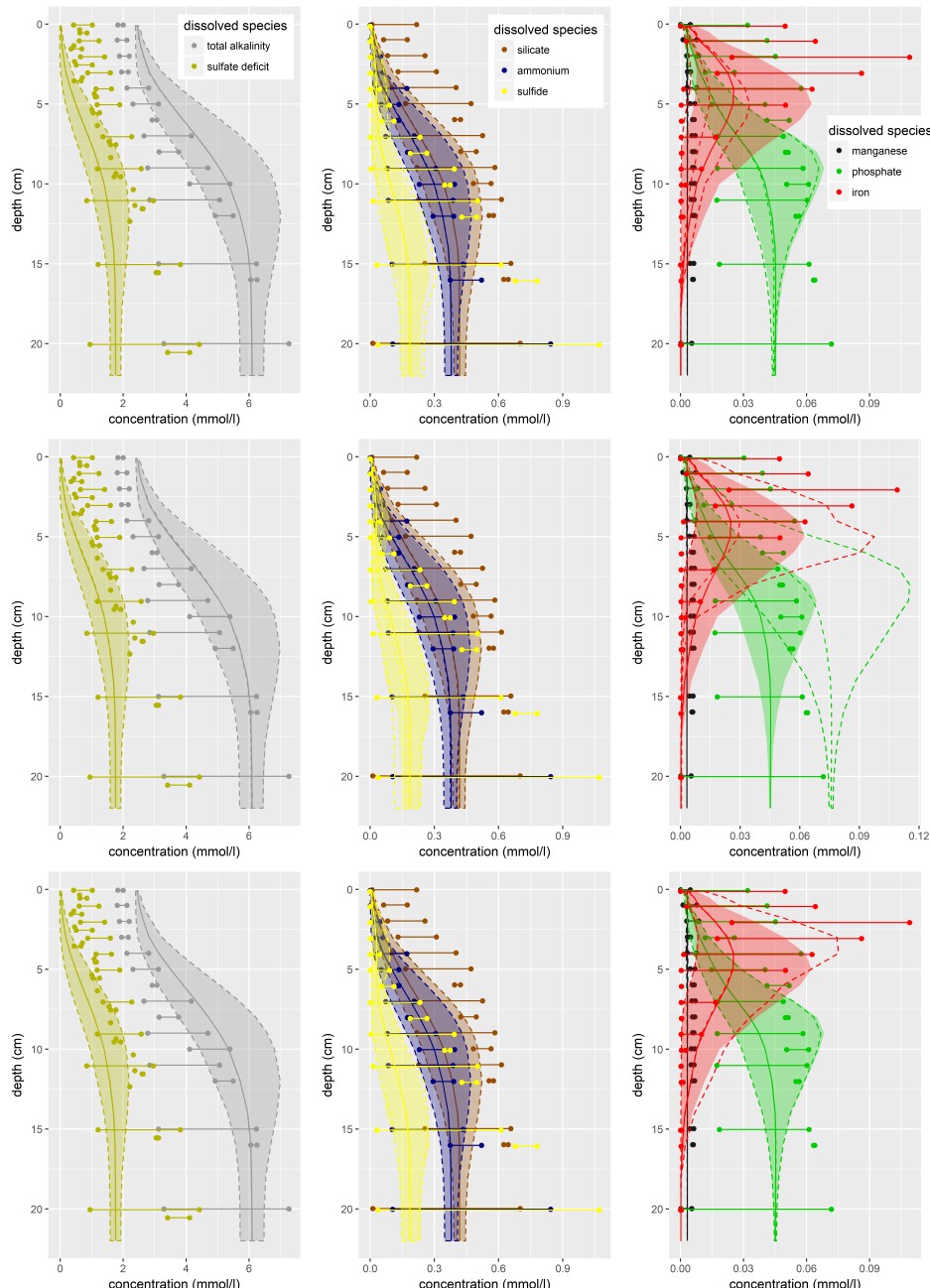

**Figure A2.** Pore water concentrations of several dissolved species at the silt station Tromper Wieck. Points and horizontal lines indicate the range of measurements. Solid curves and shading present the model results and indicate year-average concentrations and the seasonal range. Dashed curves show the same, but for a model version which neglects one of our improvements: Top row - without correcting the diffusivity of total alkalinity for hydroxide ions. Middle row - without correcting the solute concentrations by activity coefficients. Bottom row - without assuming adsorption to clay minerals.

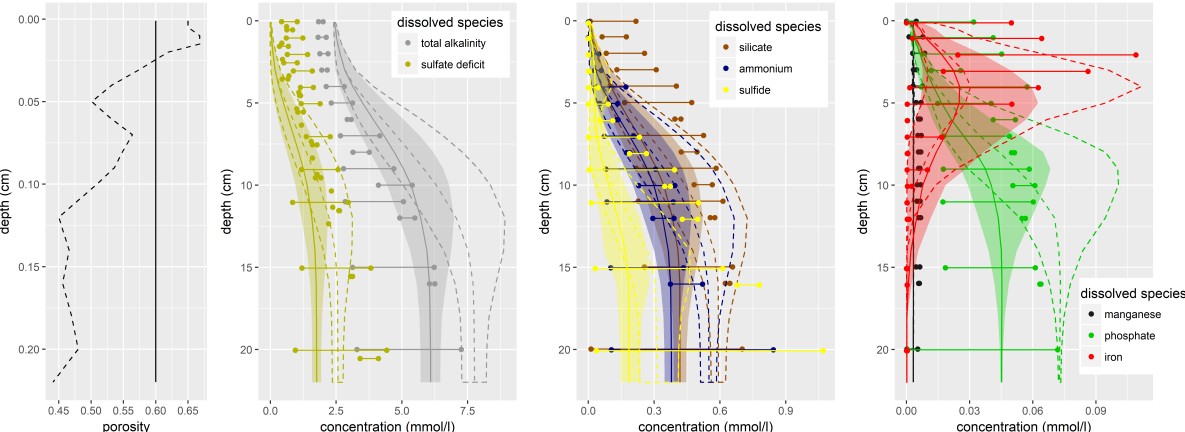

**Figure A3.** Left panel: Porosity at the silt station Tromper Wieck as assumed in the model (solid line) and a porosity profile for a realistic model setup, derived from the April 2014 cruise (dashed line). Three right panels: Pore water concentrations of several dissolved species at the silt station Tromper Wieck. Points and horizontal lines indicate the range of measurements. Solid curves and shading present the original model results and indicate year-average concentrations and the seasonal range. Dashed curves show the same, but for a model version with the realistic porosity profile.

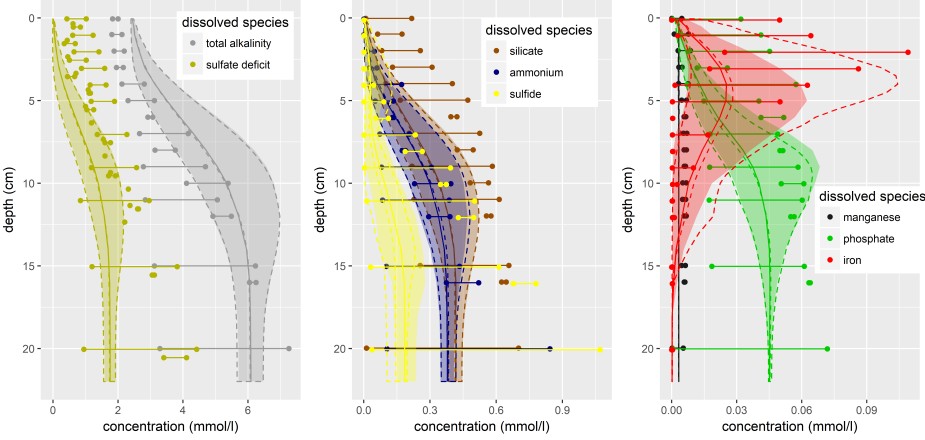

**Figure A4.** Pore water concentrations of several dissolved species at the silt station Tromper Wieck. Points and horizontal lines indicate the range of measurements. Solid curves and shading present the model results and indicate year-average concentrations and the seasonal range. Dashed curves show the same, but for a model version with double vertical resolution.

**Table A1.** stoichiometric composition of tracers

| tracer | C | Ca | Fe | H | Mn | N | O | P | S | Si | electric charge |
|---|---|---|---|---|---|---|---|---|---|---|---|
| t_no3 | | | | | | 1 | 3 | | | | -1 |
| t_lpp | 6.625 | | | 16.4375 | | 1 | 6.875 | 0.0625 | | | |
| t_spp | 6.625 | | | 16.4375 | | 1 | 6.875 | 0.0625 | | | |
| t_cya | 6.625 | | | 16.4375 | | 1 | 6.875 | 0.0625 | | | |
| t_zoo | 6.625 | | | 16.4375 | | 1 | 6.875 | 0.0625 | | | |
| t_det_? | 9.9375 | | | 22.875 | | 1 | 9.9375 | | | | |
| t_detp_? | | | | 0.28125 | | | 0.375 | 0.09375 | | | |
| t_don | | | | 4 | | 1 | | | | | +1 |
| t_poc | 1 | | | 2 | | | 1 | | | | |
| t_ihw | | | 1 | 3 | | | 3 | | | | |
| t_ipw | | | 1 | | | | 4 | 1 | | | |
| t_mow | | | | | 1 | | 2 | | | | |
| t_n2 | | | | | | 2 | | | | | |
| t_o2 | | | | | | | 2 | | | | |
| t_dic | 1 | | | | | | 2 | | | | |
| t_nh4 | | | | 4 | | 1 | | | | | +1 |
| t_no3 | | | | | | 1 | 3 | | | | -1 |
| t_po4 | | | | | | | 4 | 1 | | | -3 |
| t_h2s | | | | 2 | | | | | 1 | | |
| t_sul | | | | | | | | | 1 | | |
| t_so4 | | | | | | | 4 | | 1 | | -2 |
| t_fe2 | | | 1 | | | | | | | | +2 |
| t_ca2 | | 1 | | | | | | | | | +2 |
| t_mn2 | | | | | 1 | | | | | | +2 |
| t_sil | | | | 4 | | | 4 | | | 1 | |
| t_ohm_quickdiff | | | | 1 | | | 1 | | | | -1 |
| t_ohm_slowdiff | | | | 1 | | | 1 | | | | -1 |
| t_sed_? | 9.9375 | | | 23.0625 | | 1 | 9.9375 | | | | |
| t_sedp_? | | | | 0.28125 | | | 0.375 | 0.09375 | | | |
| t_ihs | | | 1 | | | | 3 | | | | |
| t_ihc | | | 1 | | | | 3 | | | | |
| t_ips | | | 1 | | | | 4 | 1 | | | |
| t_ims | | | 1 | | | | | | 1 | | |
| t_pyr | | | 1 | | | | | | 2 | | |
| t_mos | | | | | 1 | | 2 | | | | |
| t_rho | 1.6 | 0.6 | | | 1 | | 4.8 | | | | |
| t_i3i | | | 1 | 3 | | | 3 | | | | |
| t_iim | | | 1 | 2 | | | 2 | | | | |
| t_pim | | | | 3 | | | 4 | 1 | | | |
| t_aim | | | | 3 | 1 | | | | | | |
| h2o | | | | 2 | | | 1 | | | | |
| h3oplus | | | | 3 | | | 1 | | | | +1 |
| ohminus | | | | 1 | | | 1 | | | | -1 |
| i2i | | | 1 | 2 | | | 2 | | | | |

Tracer t_alk has been omitted since it just accumulates the changes to other tracers.

**Table A2.** decay rates of different classes of detritus

| detritus class | 1 | 2 | 3 | 4 | 5 | 6 |
|---|---|---|---|---|---|---|
| mass fraction | 26% | 16% | 16% | 16% | 8% | 18% |
| relative decay rate at $0°C$ (day$^{-1}$) | 0.0647 | 0.00924 | 0.00136 | 0.000108 | 0.0000162 | inert |

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
