# Peer review of "Ecological ReGional Ocean Model with vertically resolved sediments (ERGOM SED 1.0): Coupling benthic and pelagic biogeochemistry of the south-western Baltic Sea"

_Geoscientific Model Development, 2018_

## Referee Comment (RC1) · Anonymous Referee #1 · 13 Jun 2018

This paper presents the 1-dimensional numerical benthic-pelagic model resolving biogeochemical processes associated with organic matter degradation in water column, fluff layer, near-seafloor bioturbated marine sediments and solid sediments over a short timescale. The model parameters are constrained by the porewater profiles of dissolved chemical species and bioturbation rates collected at seven sites of Baltic Sea. The goal of the work is to provide a general model adapted to understand the role of different types of benthic sediments for the ecosystem of western Baltic Sea and considered as a basis for 3-dimensional model in the future. I do find the work to be novel

and important, however, I have some comments that the authors need to address.

Specific comments.

The model description section needs much more explanations; authors should provide more equations and possibly schemes. Sometimes I had a feeling that authors did not want to be understood.

Authors do not validate their model against benthic fluxes which are usually used as the major constraint. Modeled benthic fluxes should be reported and comparison of modelled benthic fluxes to their measured values should be provided.

Authors do not provide any result related to water column. For example depth dependent mean reactivity of sinking organic carbon can be reported.

Having 115 parameters to optimize, authors need to provide a reasonable explanation of how certain local optimum was chosen.

Minor comments.

P.5, L.26: 22 layers with 1mm at the surface is not enough. The grid should be much finer.

P.9, L.21-23: 3% per day, please explain how this number was estimated?

P.10, L.16-18: Please, provide the equation for DB(z).

Eq. 1,2 and 3: Replace $\varphi$ with $\varphi(z)$, c with c(t,z), DB with DB(z)

Eq. 3: Move $\varphi(z)$ out of differential as it is time independent

P.12, L.10-11: This is probably not true. Lateral migration assumes removal of organic particles of all kind but the same time it can be considered as a source of detritus from the other parts of the sea.

P.12, L.19-27: Please provide some general equation for plankton growth here.

Section 2.4.2: Please provide some general equation for phytoplankton respiration and mortality. Also, how do you account for day/night phytoplankton metabolism?

Section 2.4.3: Please provide some general equation for zooplankton growth here.

P.17, L.17: This simplification should be quantified. For example, you can run the model with constant and depth-dependent porosity and show that the results (benthic fluxes, porewater profiles) are similar.

Section 2.7.1: Very hard to understand, more detailed explanation is needed. Some equations/schemes would help.

P.22, L.25: Sedimentation rate is 0.00001 per day. Is this correct? I would say it should be 10 times higher.

P.23, L.15-16: With $\omega$ = 0.00001d-1 100y is not enough to fill the column with solid species, nor does it work with $\omega$ = 0.0001d-1. It basically means that sedimentation is neglected in the model.

Section 4.2: In this section authors need to specify the boundary conditions for each functional level (water column, fluffy layer and sediments). Mathematical formulation of boundaries (fixed concentration or gradient) is needed.

P.24, L.20-21: How the weighting function was applied?

P.25, L.4: AHR-ES abbreviation is given without explanation.

P.25, L.16: "We used 200 "individuals" ". Please, bring the formula to calculate the number of individuals required by AHR-ES.

P.25, L.4-18: What is the rational to put this in the paper? I think it is not needed as long as you do not compare all major evolutionary strategies.

P.25, L.27-28: "The optimisation converged after 30 iteration steps and reduced the error function from 6363 (the value obtained by previous manual tuning) to 4797". In

other words, this optimization provides the result which is just 25% better then original guess. Necessity of this optimization is questioned.

P.25, L.29-31: How many optima have been found? What can you say about sensitivity of the model to the different parameter groups?

P.26, L.18-20: This should be mentioned right after P.23, L.15-16.

P.26, L.31: 23km away? I would consider it as a different station. You should at least clearly mark the point representing this site on the plots.

P.27, L.11-12: Please provide modeled fluxes of dissolved species through sediment water interface and compare them to measured values or the values typical for each region.

---

## Referee Comment (RC2) · Anonymous Referee #2 · 13 Jun 2018

Hagen Radtke and co-authors present a 1-D benthic-pelagic ecosystem/biogeochemical model for coastal systems which is created by coupling the marine ecosystem model ERGOM to an early diagenetic model (Reed et al., 2011). In addition, some further model developments have been made to the sediment model. Most biogeochemical models currently either focus on the water column or the sediments. Therefore, a numerical representation suitable to study coupled processes between bottom waters and the surface sediments is a very useful tool. However, there are a number of weaknesses/issues in the manuscript to be resolved before

publication.

General comments:

1. Introduction and text structure: The introduction needs to be improved/restructured, better putting the model/work into context. Referring to another paper (i.e. Yakushev et al., 2017) for an overview of existing coupled models (pg. 2 ln. 13-14) is not sufficient. This part of the manuscript is critical for putting your work into context and for the motivation of your work! Why did you decide to use a new model? How does your model differ from those? Why is it better suited to your study site? I think, especially the explicit fluff layer (which is also know as the bottom boundary layer, I suppose – or are they different things?) deserves some more detail. Also the introduction should be restructured, e.g. starting with a better introduction of the importance of coupled benthic-pelagic processes and linking this to your site of interest (i.e. the German part of the Baltic Sea). This should motivate why the modelling exercise is needed (especially why using a coupled model and not just running a stand-alone sediment model if you are mainly interested in "the type of ecosystem services that coastal sediments can perform"). How can your new model help to improve our understanding of this environment (e.g. what are the most important processes here)? Also the results/conclusion section does not address these kind of questions. In general, the results section is rather short and a "story" behind your experiments or what you learn from them is missing. As this model is mainly developed to investigate benthic-pelagic interactions a discussion of simulated sediment-water interface fluxes and its comparison with observations would improve the validation of the model. Especially, as the conclusion states "... where the efflux of nutrients from the sediment strongly influences water column biogeochemistry, like in our study area."

It was not clear to me what questions you would like to answer with the model eventually? This could be included in a section on "Scope of applicability and model limitations" which is expected for a model development paper anyway and is currently missing.

2. The technical details of the implementation are incomplete: Just describing the processes/state variables qualitatively is not appropriate for a GMD paper which is supposed to be "detailed and complete". Include the most important equations and tables of parameters for e.g. (but not exclusively) the biogeochemical processes (2.4 + 2.5) in the main manuscript (e.g. compare ERSEM: Butenschön et al., 2016; PISCES: Aumont et al., 2015). Give values of parameters and references to justify your decision making. I know they are in the supplementary document but it is 189 pages long, therefore it is not easy to find what one is looking for and the very technical parameter names do not help either. I suggest not to use the code-names for the parameters in the text and equations. Give them more recognizable names and add Tables in the appendix which relates them to the code-names - if you wish (compare e.g. Aumont et al., 2015). For the diagenetic model: Add the 1-D mass conservation (general reaction-transport) equation and add table of reaction network for primary and secondary redox reactions and for precipitation/dissolution reactions (compare e.g. Reed et al., 2011; Jourabchi et al., 2005; Hülse et al., 2018).

The diffusivity (Section 2.3.1) and the initial and boundary conditions (Section 4.2) are from an unpublished MOM5 and ERGOM runs. More information on the model setups is needed as it is not possible to reproduce your results like this. This could go into the supplementary information together with the results needed to recreate your ERGOM SED results.

3. Model development: Make it more clear in the body of the manuscript what your model improvements are and describe them in more detail. Apart from abstract and conclusion this is not clearly mentioned and the explanation of the new developments are generally very short.

The coupling of the different modules (i.e. water column, fluff-layer, sediment) is obviously a new development as well but it is not clear to me how it is done. E.g. how do you calculate the bidirectional fluxes of dissolved species (give equations). Or the coupling of the fluff layer with sediment-column: You assume zero porosity? How are

solute species transported from the fluff to the sediments?

4. Text structure and referencing: The manuscript could benefit from another round of editing, looking at the structure and referencing (technique and missing citations). Examples of wrong referencing can be found: pg. 1 ln.7 ; pg. 2 ln. 5 + 15; pg. 10 ln. 15; pg. 22 ln. 5; pg. 27 ln. 1-2; etc. The breakdown of the text into paragraphs and linebreaks should be revisited. Linebreaks after just 1 or 2 sentences are often used (e.g. 2.3.7, 2.3.8, 2.3.9, 2.4.5, 2.5.2, 2.5.5 etc.) which does not help with the flow of the paper. Also some of the crosslinks given to other sections of the text are not correct (see specific comments).

Specific/technical comments:

Abstract: Abstract should include some information about the main findings/how the model performs.

ln. 11: what does SECOS stand for?

pg. 2 ln.8-9: Why are there so few coupled benthic-pelagic model studies?

pg. 3 ln. 16 – 20. You introduce here the seven study sites you are modelling and categorize them by their granulometric properties. Some information on how these three categories differ and what these differences mean for modelling the sites would be good.

pg. 5 2.1 Ancestor models + 2.2 Model compartments and state variables: A better short summary of the main, specific features of the ancestor models is needed for the reader to understand the model setup. Both sections could be combined. Then make it more clear would are the improvements done here to the sediment model.

pg. 5 footnote: Mention that these input files can be found in the specific stations folder. It took me a while to find it.

pg. 6 ln. 1-3: Does that mean, porosity is always constant in the sediment column?

Does the model also work with varying porosity?

pg. 6 ln. 10 – 15: The alkalinity description is unmotivated and too technical. It would be more clear if you describe in a sentence or two how alkalinity is calculated and add the change in parameters in parenthesis. Also, I can't find a clear explanation for the reasoning of the approach in Section 2.6 as promised here (pg. 6, ln. 15).

pg. 6 Section 2.3 Transport processes: There are a lot of often very short subsections. I would propose to have just 2 subsections: 2.3.1 Ocean 2.3.2 Fluff layer/Sediment. Then using different paragraphs for different processes to increase readability.

pg. 6 ln. 25: "... lateral transport processes have a major impact." please give reference

pg. 6 ln. 29-30 relax wintertime nutrients in the surface layer: Is this approach adopted from somewhere (Ref)? Does this lead to realistic results?

pg. 7 ln. 2: The parameterisation of lateral tranport is transport is discussed in this section (see 2.3.9 Parameterisation of lateral transport)

pg. 7 ln. 6-9: more information needed fo unpublished model run. KPP used without explanantion

pg. 9 Particle sinking: Is there a reference for the different sinking speeds?

pg. 9 ln. 14: replace "from" with "as"

pg. 9ln. 23: where does the rate of bioerosion come from? Ref?

pg. 10 ln. 10-15: coupling of fluff and sediments unclear Also why 3mm? What is a typical thickness, what influences it?

pg. 10 ln. 16-20: reference for the approach? Also, the oxygenation of the sediment column effects the depth/rate of bioturbation! Is this not represented in the model? I.e. is bioturbation possible even in the sulfidic zone? If you just fit your model to a specific site you probably change this manually but what are you doing when coupled to a 3D

ocean model?

pg. 11 ln. 6-9: Give equation for the exchange flux. Again the 3mm here...

pg. 11 ln. 22: replace "sediments" with settles or is deposited. I assume the sediment accumulation is taken as advective transport in the diagenetic model!? Clarify in the text.

pg.12 ln. 6-11: why transport away/towards different sites? explain/justify

pg. 12 Biogeochemical processes in the water column: add references for previously published ERGOM version you mean (ln. 16) and the equations for the processes described in the following subsections (2.4.1 – 2.4.5) with tables of parameters and their values as used in the model equations (see general comments).

pg. 13 ln. 18: rewrite to "from previous ERGOM versions". I find the use of the term detritus here confusing as you use organic carbon/material in the rest of your manuscript - Particulate organic carbon (POC) might be a better choice

pg. 13 ln. 19-23 / 24-26: The decay rate constants and the partitioning into reactivity-classes are probably the most important steps for the model output (e.g. Arndt et al., 2013, Hülse et al., 2018). Therefore, this deserves some more words and justification.

pg. 13 decay rate constants: where do the 0-degree values come from? - are they representative for your study area? - how do you know? - and how is it temperature dependent? Give equation!

pg. 13 Partitioning: Is the Middelburg approach not for OM at the sediment-water interface? Could you please clarify and give the equation used to calculate the fractions. Also what is the fraction of the non-decaying detrital?

pg.15 ln. 25-26: Give equation for conversion from SO4 to H2S depending on the diffusive CH4 flux from below the model domain

pg. 16 ln. 1: you DO favour the latter theory because your rate constants are independent of the TEA. Or am I wrong?

pg. 16 ln. 1-2: You say:"...we chose to adopt the decay rates proposed by Middelburg (1989), which may implicitly take the effect of the oxidant into account." I do not understand this statement. If I remember it right, the Middelburg (1989) rate model is the same for oxic and anoxic conditions. Also what are the values of your degradation rate constants? They are the most important parameters in the diagenetic model. List them e.g. in a table, together with the rate constants for the water column and the fractions for the OM partitioning.

pg. 16 ln. 3: change "their study". It is just one author.

pg. 16 ln. 4-7 preferential release of P: under which conditions is P preferentially released? Is this important for your study area? As you say on page 13 ln. 16: "anoxic conditions which, however, do not occur in our study area."

pg. 16 t_detp_n :refer to table A1. How do you get the t_detp_n and t_det_n numbers for H? I understand that detritus is 50% enriched in C and P and how the values for C, N, O, P are calculated. But H does not add up. Should H for t_det_n & t_sed_n not be 22.875?; pref_remin_p: what's the value and where does it come from?

pg. 16 2.5.2 Specific mineralisation processes: Add table for reaction network as mentioned in general comments

pg. 16 ln. 17-20: The description is very vage - list the station specific content e.g. in table 2 and what is the "small amount of reducible iron"? Quantify!

pg. 16 ln. 24-25: reference for statement

pg. 17 2.5.4 Pyrite formation: you did some model development here: add equations to make in more clear

pg. 18 ln. 8-9: state the formula

pg. 18 2.5.6 Reoxidation of reduced substances: There is a lot of information about

the reaction network here. A summary of all that in a table would be very helpful! Also just use either iron-II/III or Fe-II/III, don't mix them up

pg. 19 2.6 Carbon cycle: this is also a new model development – at least include the equations you use to calculate pH and pCO2.

pg. 20 ln. 4 mode splitting method: give some quick background what this is and a reference

pg. 21 ln. 7: style. Change to: . . . in the Southern Baltic Sea (see Fig. 1, we always present the stations from west to east).

pg. 21 ln. 9: units of salinity of 20 is missing pg. 21 ln. 21: replace "interface" with interfaces

pg. 22 ln. 14: replace "So" with thus

pg. 22 ln. 22: replace "sampled" with samples

pg. 23 4.2 Initial and boundary conditions: ERGOM model: more information needed for the unpublished model run as stated in general comments Is the relaxation approach of DIN and DIP adopted from somewhere? Does this lead to realistic results? pg. 24+25 4.3.2 Optimisation strategies: Why do you talk about the application of the AHR-ES algorithm which is in the end not used at all? Delete this part. pg. 25 ln. 19 - end: What parameters are changed? What is the range they are varied in (add table to appendix)? Why don't you show any results of the optimisation? Are the final parameter values realistic, e.g. compared with other models or data? What are the most important parameters?

pg. 26 4.4 Manual correction of sand and silt station: How exactly did you modify the parameters? This is needed for reproducibility of your results!

pg. 26 ln. 6: which detritus is meant here (just POC or POC with mineral particles)? - rephrase "out of the sediments"

pg. 26 - 5.1.1 Pore water profiles at mud stations: It should be easy to check with the data if the variability at site AB is because they are 23km apart.

pg. 30 ln. 1 phosphate is in the right panel

pg. 31. ln. 3-4: reference for this statement

pg. 31 ln. 6: rephrase

pg. 33 Conclusion: ignoring the N-cycle because it's not part of the SECOS project: This is a rather poor justification, especially as you say later: "... where the efflux of nutrients from the sediment strongly influences water column biogeochemistry, like in our study area." and "... denitrification [...] may strongly influence marine ecosystems".

pg. 37 Table A1: stoichiometry of t_h2s is wrong: it should have 2*H and 1*S

---

## Referee Comment (RC3) · Anonymous Referee #3 · 19 Jun 2018

The authors proposed a coupled framework for vertically resolved pelagic and benthic models. This framework is built from two existing models: ERGOM (Neumann et al. 2000) for the pelagic part, and the diagenetic model of Reed et al. 2011 for the diagenetic part. Besides the coupling framework, the benthic model has undergone several developments. The model is applied on 7 stations of distinct environments in the south western Baltic Sea (mud,sands,silts) and shows the ability to reproduce most of benthic observations.

The manuscript is clear, well illustrated and well written in general but lacks the

rigourous mathematical description one could expect from a model description manuscript. I understand the reasons for, and support, a large qualitative description that helps in getting a quick grab at which processes the model considers and which it discards, but this should be complemented by descriptive equations, eg. in appendixes. The reference to the user manual is not satisfactory, since it only contains automatically generated code, hard to read. In addition to this, I suggest major revision also to enhance the justification of the model developments presented here : enhancement of the benthic model, B-P coupling framework.

General Comments

* In the introduction and conclusion, the emphasis while presenting the ERGOMSED model is put on the online coupling between the Benthic and Pelagic part. However this online coupling is not valorised in the results and discussion section. This should be enhanced. Neither the pelagic part nor the solutes exchanges between the benthic and pelagic part are mentioned in the results, although the conclusion states that "In the long term, biogeochemical ocean models should aim at a process-resolving description of surface sediments. This is especially true for shallow ocean areas where the efflux of nutrients from the sediment strongly influences water column biogeochemistry, like in our study area." In the case that benthic fluxes, for any reason, are not available within SECOS (which would be surprising), ranges from the litterature could be used for comparison, and it would also be relevant to compare benthic fluxes to the lateral fluxes (inferred from the nudging procedure for the pelagic nutrients) to stress the relevance of such coupled framework.

* The fluff layer approach is an interesting feature of the coupling set-up, and a practical solution to handle solids lateral transport and exchanges between benthic and pelagic part. To my knowledge, the use of a fluff layer is not systematic in B-P coupled models and I would have found relevant to enforce introduction and discussion on this aspect.

* Appendix B supposedly justifies the inclusion of enhanced dynamics in the benthic

model. This should be more developed. In particular: 1) Has the same calibration procedure been applied "from scratch" after having switched off those processes; 2) Some of the "reduced" experiments actually seems to behave better than the reference simulation. Can the authors justify their modification in this context ?

Specific Comments

* P3L13 : I suggest to add a references on ecosystem services ( for instance : https://cices.eu/content/uploads/sites/8/2012/07/CICES-V43_Revised-Final_Report_29012013.pdf)

* From P9L17-19 and P10L10-15 I had understood that solid compounds were only transferred from the distinct fluff and upper sediment layers through bioturbation (which includes here also other mixing effects). However, at P11L29 we learn that accumulation (advection) also induces a transfer from the fluff to the sediments. This may be introduced earlier (eg. end of Sect. 2.3.3) and explained in more details.

* Tab 1. : Benthic tracers (both solids and dissolved) are defined in mol/m$^2$ which doesn't correspond with the definitions given in P10L6 and P10L25. Is there a general transfomation applied to get those in units of mol/volume of liquids/solids ?

* P10L10-20 : Bioturbation rate are defined for the sediment compartment. Is the uppermost value used for diffusion between the fluff layer and the uppormost sediment cell ? please precise.

* Sect 2.4: Interactions between phosphate and iron aren't described for the water column. Are t_ipw, t_ihw and t_mow only included to enable a lateral transport of resuspended solids, or is there possible biogeochemical transformation in the water column ?

*P24L3 : In general, it might be relevant to comment which parameters were considered for the calibration and which were adapted, which were considered as equals for all stations and which were considered to differ between stations.

* P24L15: It is not clear whether $\Delta_i$ is defined specific only to each variables, or specific also to each station, or also to each sampling depth. This is relevant as refferred to when discussing model performances.

* P26L3 : Should the first "bioirrigation" be replaced by "bioturbation" ?

* Sect 5.12 : As is true for numerous model of this type, application in sandy sediments might be limited to the the lack of consideration of processes specific to permeable sediments. "Whashout" is mentioned in Sect 2.3.3, but this isn't the only aspect of it. This should be discussed. For instance in this section. You might refer to the review from Huettel et al, 2013.

* P30L2-3 : Switch "higher" and "lower".

* P30L19 : I don't see a TOC maximum at the top of sediments for station DS. Concerning this last paragraph, you could maybe consider the fact that the inability of the model to provide a TOC profile increasing with depth is related to the absence of dynamics specific to permeable sediments , washout in particular ? I insist on this point since it represents a major challenge for BP coupling intended to be implemented on shelves with mixed sand/mud conditions. I don't ask that this be solved in this study, but the issue should be commented.

* Table A1 : t_h2s has one H and 2 S ? Is that an error in table or in the model ?

[Figure]

---

## Author Comment (AC1) · 1 Aug 2018

**Authors' response to referee comments**

Always given as: Referee comment – Authors' response – Changes to manuscript

**General Remark:**

First of all, we wish to thank the reviewers for their detailed reading of the manuscript, their constructive comments and their advice to modify and improve our manuscript!

There have been several comments asking us to provide quantitative description of selected processes in the main manuscript, while our intention was to keep it away from there and put it into the online supplement, for clarity and readability reasons. This decision was made after thorough consideration of potential advantages and disadvantages.

Showing some equations and model constants in the main text and hiding others, that is, giving an incomplete quantitative description, would lead to a somewhat unbalanced choice of which processes we think are important. On the other hand, giving a complete quantitative description would certainly expand the manuscript length towards an unreadable volume. This can be easily estimated by looking at the complete description of the model equations given in the supplement. Even if the quite technical description might deflate a bit when being translated into mathematical formulae, about 20-30 pages might remain. We decided not to present the whole set of equations because they are quite common and appear in a number of other models on the biogeochemistry of diagenesis in a very similar way. Presenting them would be essentially a repetition.

We agree with all three reviewers that a more thorough description of the non-common processes we added to the ancestor models is required. Therefore, we modified this part of our concept and now include a quantitative mathematical description of these ones. We hope that this compromise – qualitative description only of 'established' processes and thorough mathematical description of "un-common" processes, complemented by the precise technical description of the complete model in the supplement – is acceptable for the scientific community. Non-modelers might appreciate the possibility to get a quick qualitative overview of which processes are considered in the model and which are not. Modelers who wish to include the "non-common" processes into their own models will find a thorough description. Readers interested in applying our model will need to dig into the details anyway and will have to familiarize with our naming of the state variables etc., so the (rather technical) complete model description serves their potential needs.

Therefore, we appreciate and follow the reviewers' suggestions and include a thorough quantitative description of processes in the main text, but only for those processes that are not common in previous models.

**Reviewer 1 – specific comments**

The model description section needs much more explanations; authors should provide more equations and possibly schemes. Sometimes I had a feeling that authors did not want to be understood.

While we agree that more detailed descriptions are sometimes needed, we strongly reject the last speculation. The limited clarity rather arises from the attempt to avoid that the paper gets too long which might distract interested readers.

We readjust the balance between clarity and conciseness and add more explanations, making the paper longer than before. We especially add detail on (a) fluff layer representation and (b) added

biogeochemical processes. At the same time, we keep the quantitative description of the "old" processes taken from the ancestor models in the online supplement.

Authors do not validate their model against benthic fluxes which are usually used as the major constraint. Modeled benthic fluxes should be reported and comparison of modelled benthic fluxes to their measured values should be provided.

We invited two additional co-authors who added flux measurements by benthic chamber landers at the selected stations for additional model validation.

Authors do not provide any result related to water column. For example depth dependent mean reactivity of sinking organic carbon can be reported.

The reason for this is that the measurements taken during the sampling campaign focused on the sediment, so we have limited data to compare to.

We add graphs of mean reactivity of sinking organic carbon to the online supplement.

Having 115 parameters to optimize, authors need to provide a reasonable explanation of how certain local optimum was chosen.

The automatic method was started after manual calibration of the model. Since the optimisation method is deterministic, the local optimum is defined by this initial condition.

We clarify this in section 4.3.

**Reviewer 1 – minor comments**

P.5, L.26: 22 layers with 1mm at the surface is not enough. The grid should be much finer.

The resolution represents a compromise between accuracy and the need for a limited numerical effort. The latter is essential for the later application in a three-dimensional context.

For sensitivity analysis to the vertical resolution, we add a comparison to a model with doubled vertical resolution in every layer to the online supplement.

P.9, L.21-23: 3% per day, please explain how this number was estimated?

We add: "This number was estimated from calibration of a 3-dimensional Baltic Sea ecosystem model (Neumann et al. …) where the process showed to be critical for transporting organic matter to the deep basins below a depth of approx. … m. In these depths, a resuspension due to wave-induced shear stress is no longer possible."

P.10, L.16-18: Please, provide the equation for DB(z).

We give the requested equation.

Eq. 1,2 and 3: Replace ϕ with ϕ(z), c with c(t,z), DB with DB(z)

We did the requested replacements.

Eq. 3: Move φ(z) out of differential as it is time independent

We agree that this simplifies the equation, even if the presented formulation is still correct.

We did this in eq. 2 and 3

P.12, L.10-11: This is probably not true. Lateral migration assumes removal of organic particles of all kind but the same time it can be considered as a source of detritus from the other parts of the sea.

I guess this is a misunderstanding: Fluff material above sandy sediments partly decomposes there before being advected to deeper areas. Therefore, as an approximation, we assume that the quickest degradable parts will not arrive at the mud stations because they were already mineralized at the shallow sandy locations.

We changed the sentence to: "We assume that the quickest-degradable part of the detritus is already mineralised in the shallow coastal areas, before its lateral migration to the mud stations, and therefore exclude the first two classes from this artificial input."

P.12, L.19-27: Please provide some general equation for plankton growth here.

We decided after thorough consideration of advantages and disadvantages to provide the quantitative description of the "old" parts of the BGC model only in the online supplement. Giving some equations like this one and hiding others would be inconsistent. Giving all equations would massively increase the length of the manuscript and is therefore not an option for us.

No changes to manuscript

Section 2.4.2: Please provide some general equation for phytoplankton respiration and mortality. Also, how do you account for day/night phytoplankton metabolism?

See comment above. / We do not account for day/night metabolism, but represent phytoplankton as a Redfield-ratio state variable which will only grow when light is present. This is a simplification used in many ecosystem models.

We added the sentence: "This simplifying description of phytoplankton growth does not describe day/night metabolism."

Section 2.4.3: Please provide some general equation for zooplankton growth here.

See comments above

No changes to manuscript

P.17, L.17: This simplification should be quantified. For example, you can run the model with constant and depth-dependent porosity and show that the results (benthic fluxes, porewater profiles) are similar.

This simplification has been made due to limited spatial availability of data for the 3-dimensional approach. While detailed spatial maps of surface porosity exist, vertical profiles are rare.

We added a new appendix section showing a comparison between a constant-porosity profile and another one with vertically varying porosity.

Section 2.7.1: Very hard to understand, more detailed explanation is needed. Some equations/schemes would help.

We did not go into too much detail here since we expect that only few readers are actually interested in the numerical details.

We added a new appendix section describing the application of the Al-Hassan method.

P.22, L.25: Sedimentation rate is 0.00001 per day. Is this correct? I would say it should be 10 times higher. / P.23, L.15-16: With $\omega$ = 0.00001d-1 100y is not enough to fill the column with solid species, nor does it work with $\omega$ = 0.0001d-1. It basically means that sedimentation is neglected in the model.

Yes, you are right. This is an artificially small value, but different from zero for numerical purposes. It basically reflects the fact that the decay time scale of chlorophyll is much lower than the time scale for sediment accumulation.

We changed the sentence: "Experimentally derived chlorophyll decay constants of 0.01 d-1 for mud and 0.02 d-1 for sand (Morys, 2016) and an artificially small sedimentation rate $\omega$ of 0.00001 cm d-1 were used, the latter just reflecting the fact that chlorophyll decay is much faster than sedimentation."

Section 4.2: In this section authors need to specify the boundary conditions for each functional level (water column, fluffy layer and sediments). Mathematical formulation of boundaries (fixed concentration or gradient) is needed.

We added a mathematical description of the boundary conditions.

P.24, L.20-21: How the weighting function was applied?

This is already described in Equation 9.

No changes to manuscript.

P.25, L.4: AHR-ES abbreviation is given without explanation. / P.25, L.16: "We used 200 "individuals" ". Please, bring the formula to calculate the number of individuals required by AHR-ES. / P.25, L.4-18: What is the rational to put this in the paper? I think it is not needed as long as you do not compare all major evolutionary strategies.

Our intention to put it in here was to serve as a justification for using the simple strategy, illustrating that we have not used it because of its simplicity but because of its better performance compared to this more complicated method which had been suggested to us by colleagues. But probably this is not needed. Also, we thought it is useful information for others that we did not succeed applying this method.

We skip the description of the AHR-ES method (which we spell out as Adaptative Hierarchical Recombination – Evolutionary Strategies) and just mention that it was our first attempt but we did not succeed.

P.25, L.27-28: "The optimisation converged after 30 iteration steps and reduced the error function from 6363 (the value obtained by previous manual tuning) to 4797". In other words, this optimization provides the result which is just 25% better then original guess. Necessity of this optimization is questioned.

We don't know whether 25% is a little or a lot, because we would not expect anyway that a perfect optimization method arrives at an error of zero. There are (a) structural errors induced in the model by using a simplified set of processes and (b) errors due to a suboptimal choice of the parameters, and only the latter ones can be reduced by the optimisation. By "necessity is questioned" you probably mean a cost-benefit analysis? Traditionally, modelers just use a manual optimisation of parameters, we also did that and ended at the starting point for the additional automatic calibration, from where we could gain another 25%.

No changes to manuscript.

P.25, L.29-31: How many optima have been found? What can you say about sensitivity of the model to the different parameter groups?

Since the method is deterministic, only one optimum has been found.

We mention this fact here. Also, we add an additional appendix section giving the sensitivities of the penalty function to changes in individual model parameters.

P.26, L.18-20: This should be mentioned right after P.23, L.15-16.

We add the following sentence there: "While this period of 100 years is not sufficient to fill the considered 22 cm of sediment by accumulation, it is sufficient to almost reach a steady state in the pore water concentrations. While the sixth class of detritus, which is considered non-biodegradable, continues accumulating in the sediments after 100 years, those classes which affect the pore water concentrations decay on smaller time scales."

P.26, L.31: 23km away? I would consider it as a different station. You should at least clearly mark the point representing this site on the plots.

We use different symbols for the different locations in the plot now.

P.27, L.11-12: Please provide modeled fluxes of dissolved species through sediment water interface and compare them to measured values or the values typical for each region.

We invited two additional co-authors who added flux measurements by benthic chamber landers at the selected stations for additional model validation.

**Reviewer 2 – general comments**

1. Introduction and text structure: The introduction needs to be improved/restructured, better putting the model/work into context. Referring to another paper (i.e. Yakushev et al., 2017) for an overview of existing coupled models (pg. 2 ln. 13-14) is not sufficient. This part of the manuscript is critical for putting your work into context and for the motivation of your work!

We spell out an overview of existing models now, mostly repeating what Yakushev et al. compiled in addition to referring to it.

Why did you decide to use a new model? How does your model differ from those? Why is it better suited to your study site? I think, especially the explicit fluff layer (which is also know as the bottom boundary layer, I suppose – or are they different things?) deserves some more detail.

Thank you for this suggestion! The basic idea was to extend our existing ecosystem model ERGOM into the sediment, and the fluff layer transport was already implemented there. But you may be right in stating that emphasizing this difference to other early diagenetic models might be critical.

We add a paragraph after the description of existing bentho-pelagic models which answers this question.

Also the introduction should be restructured, e.g. starting with a better introduction of the importance of coupled benthic-pelagic processes and linking this to your site of interest (i.e. the German part of the Baltic Sea). This should motivate why the modelling exercise is needed (especially why using a coupled model and not just running a stand-alone sediment model if you are mainly interested in "the type of ecosystem services that coastal sediments can perform"). How can your new model help to improve our understanding of this environment (e.g. what are the most important processes here)?

Very good point that spelling out the importance of bentho-pelagic coupling in nature puts the work into context!

We start the introduction now with a paragraph on the importance of bentho-pelagic coupling for coastal ecosystems. Also we add an "outlook of an outlook" by explaining which type of questions might be answered one day with the 3-d version of the model, motivating this new modelling approach.

Also the results/conclusion section does not address these kind of questions. In general, the results section is rather short and a "story" behind your experiments or what you learn from them is missing.

It's a model description paper, not a presentation of scientific results. As such, it does not focus on a detailed story.

No changes to manuscript.

As this model is mainly developed to investigate benthic-pelagic interactions a discussion of simulated sediment-water interface fluxes and its comparison with observations would improve the validation of the model. Especially, as the conclusion states "... where the efflux of nutrients from the sediment strongly influences water column biogeochemistry, like in our study area."

We invited two additional co-authors who added flux measurements by benthic chamber landers at the selected stations for additional model validation.

It was not clear to me what questions you would like to answer with the model eventually? This could be included in a section on "Scope of applicability and model limitations" which is expected for a model development paper anyway and is currently missing.

We add this section explicitly as part of the "conclusions" section.

2. The technical details of the implementation are incomplete: Just describing the processes/state variables qualitatively is not appropriate for a GMD paper which is supposed to be "detailed and complete". Include the most important equations and tables of parameters for e.g. (but not exclusively) the biogeochemical processes (2.4 +2.5) in the main manuscript (e.g. compare ERSEM: Butenschön et al., 2016; PISCES: Aumont et al., 2015).

Please see our general reply to this at the first page.

We follow the suggestion and add a detailed mathematical description, but only for those processes which are new in the model.

Give values of parameters and references to justify your decision making. I know they are in the supplementary document but it is 189 pages long, therefore it is not easy to find what one is looking for …

Including everything in the main text would cause the same problem as in the supplementary document.

No changes to manuscript

… and the very technical parameter names do not help either. I suggest not to use the code-names for the parameters in the text and equations. Give them more recognizable names and add Tables in the appendix which relates them to the code-names - if you wish (compare e.g. Aumont et al., 2015).

I agree that the names for the process rates (shaped p_precursor_process_product) are very technical. For the parameters, this seems to be a matter of taste. I particularly dislike parameters called $\alpha_3$ or similar whose names give no indication of what they are, and I also very much dislike if they are called differently in the manuscript and in the code.

No changes to manuscript.

For the diagenetic model: Add the 1-D mass conservation (general reaction-transport) equation and add table of reaction network for primary and secondary redox reactions and for precipitation/dissolution reactions (compare e.g. Reed et al., 2011; Jourabchi et al., 2005; Hülse et al., 2018).

We add the general reaction-transport equation in the beginning of the model description section. Also, we add a reaction network table as suggested.

The diffusivity (Section 2.3.1) and the initial and boundary conditions (Section 4.2) are from an unpublished MOM5 and ERGOM runs. More information on the model setups is needed as it is not

possible to reproduce your results like this. This could go into the supplementary information together with the results needed to recreate your ERGOM SED results.

The model output for the different stations is already included in the online supplement, and the ERGOM-SED results can be reproduced with the information given there. We agree that adding more details on the 3-d model run is required, but reproducing the exact 3-d model results just from a description in a paper is impossible anyway.

We add more information on the 3-d model run which produced the data.

3. Model development: Make it more clear in the body of the manuscript what your model improvements are and describe them in more detail. Apart from abstract and conclusion this is not clearly mentioned and the explanation of the new developments are generally very short.

We add detailed information on the new processes.

The coupling of the different modules (i.e. water column, fluff-layer, sediment) is obviously a new development as well but it is not clear to me how it is done. E.g. how do you calculate the bidirectional fluxes of dissolved species (give equations). Or the coupling of the fluff layer with sediment-column: You assume zero porosity? How are solute species transported from the fluff to the sediments?

We give mathematical equations on these exchange processes.

4. Text structure and referencing: The manuscript could benefit from another round of editing, looking at the structure and referencing (technique and missing citations). Examples of wrong referencing can be found: pg. 1 ln.7 ; pg. 2 ln. 5 + 15; pg. 10 ln. 15; pg. 22 ln. 5; pg. 27 ln. 1-2; etc.

Thank you for the suggestion, we change these.

The breakdown of the text into paragraphs and linebreaks should be revisited. Linebreaks after just 1 or 2 sentences are often used (e.g. 2.3.7, 2.3.8, 2.3.9, 2.4.5, 2.5.2, 2.5.5 etc.) which does not help with the flow of the paper.

I prefer if paragraphs indicate when a line of thought has ended, but I agree that most others might prefer longer paragraphs.

We combine short paragraphs throughout the manuscript to improve the flow of reading.

Also some of the crosslinks given to other sections of the text are not correct (see specific comments).

Thank you for pointing this out!

We change the mistakes given and check all references again.

**Reviewer 2 – specific comments**

Abstract: Abstract should include some information about the main findings/how the model performs.

We add a description of model performance to the abstract.

ln. 11: what does SECOS stand for?

We spell it out as "The Service of Sediments in German Coastal Seas".

pg. 2 ln.8-9: Why are there so few coupled benthic-pelagic model studies?

Answering this would be pure speculation, so we will not do it in the paper. We guess it is because (a) the communities of early diagenetic modellers and pelagic ecosystem modellers are traditionally separate, (b) the spatial focus is different (single site versus whole ocean basin) and (c) for studies of pelagic processes, it is perfectly accepted to just state that the coarse representation of benthic processes is a weakness of the model, and that's much easier than trying to improve it.

No changes to manuscript.

pg. 3 ln. 16 – 20. You introduce here the seven study sites you are modelling and categorize them by their granulometric properties. Some information on how these three categories differ and what these differences mean for modelling the sites would be good.

We add a short summary of main biological differences between sand, silt and mud sediments.

pg. 5 2.1 Ancestor models + 2.2 Model compartments and state variables: A better short summary of the main, specific features of the ancestor models is needed for the reader to understand the model setup. Both sections could be combined. Then make it more clear would are the improvements done here to the sediment model.

We improve the description of these models. The improvements are pointed out better in the description of the physical and biogeochemical processes.

pg. 5 footnote: Mention that these input files can be found in the specific stations folder. It took me a while to find it.

We add this to the first footnote of its kind.

pg. 6 ln. 1-3: Does that mean, porosity is always constant in the sediment column? Does the model also work with varying porosity?

This simplification has been made due to limited spatial availability of data for the 3-dimensional approach. While detailed spatial maps of surface porosity exist, vertical profiles are rare.

We added a new appendix section showing a comparison between a constant-porosity profile and another one with vertically varying porosity.

pg. 6 ln. 10 – 15: The alkalinity description is unmotivated and too technical. It would be more clear if you describe in a sentence or two how alkalinity is calculated and add the change in parameters in parenthesis. Also, I can't find a clear explanation for the reasoning of the approach in Section 2.6 as promised here (pg. 6, ln. 15).

We add here: "Alkalinity is a quantity describing the buffering capacity of a solution against adding acids, it describes the amount of a strong acid that needs to be added to titrate it to a pH of 4.3." In Section 2.6, we add: "Total alkalinity changes if acidic or alkaline substances are added or removed. The substances occurring in our model equations which change alkalinity are OH-, H3O+ and PO43- ions." Also, we move the quantitative description of alkalinity change to Section 2.6.

pg. 6 Section 2.3 Transport processes: There are a lot of often very short subsections. I would propose to have just 2 subsections: 2.3.1 Ocean 2.3.2 Fluff layer/Sediment. Then using different paragraphs for different processes to increase readability.

We prefer to keep the short subsections since the length of the text now increases due to the description of the fluff layer processes.

No changes to manuscript.

pg. 6 ln. 25: "... lateral transport processes have a major impact." please give reference

We add three references, Schneider et al. 2010 and Emeis et al. and Christiansen et al. 2002.

pg. 6 ln. 29-30 relax wintertime nutrients in the surface layer: Is this approach adopted from somewhere (Ref)? Does this lead to realistic results?

No, this approach is not adopted. It by definition leads to realistic results for the wintertime nutrient concentrations, which mostly determine the export production of the surface layer.

pg. 7 ln. 2: The parameterisation of lateral tranport is transport is discussed in this section (see 2.3.9 Parameterisation of lateral transport)

We change the reference from 2.4 to 2.3.9

pg. 7 ln. 6-9: more information needed fo unpublished model run. KPP used without explanation

We add more information on the 3-d model run which produced the data. KPP is spelled out as K profile parametrisation.

pg. 9 Particle sinking: Is there a reference for the different sinking speeds?

They are taken from the previous ERGOM version, where they were just obtained by model calibration as described.

No changes to manuscript.

pg. 9 ln. 14: replace "from" with "as"

We do the suggested replacement.

pg. 9ln. 23: where does the rate of bioerosion come from? Ref?

They are taken from a previous ERGOM version (Neumann and Schernewski, 2008), where they were just obtained by model calibration.

We add "This value was obtained by calibration of a previous 3-d version of ERGOM (Neumann and Schernewski, 2008)." We also add a reference for the bioresuspension process (Graf and Rosenberg, 1996).

pg. 10 ln. 10-15: coupling of fluff and sediments unclear Also why 3mm? What is a typical thickness, what influences it?

We agree that a more mathematical formulation is required to clearly describe the coupling of fluff and sediments. For the 3mm, they are estimated from observed SPM concentrations during times when fluff was suspended due to exceedance of the critical bottom shear stress.

We add a mathematical formulation of the coupling between fluff and sediment. Also, we add the following:

"We, however, assume it to be perfectly compacted (phi = 0) to be able to apply the above equation to describe the exchange process, and therefore assume a thickness of 3 mm. This describes a volume estimate of SPM taken from this region: Typical SPM concentrations in the lowermost 40 cm of the water column are about 8 mg/l higher compared to the value 5 m above the sea floor (Christiansen et al. 2002). As the density of these particles is just slightly higher than that of the surrounding water, we can estimate their volume at approximately 3 l/m² which gives 3 mm of height if perfectly compacted. Assuming this perfect compaction is not a physical assumption but rather a numerical trick which we use to transport the fluff material into the sediments. In reality, the fluff layer may be up to a few centimetres thick, and the incorporation of organic matter into the sediment is done by macrofaunal activities, e.g. (van de Bund et al., 2001)."

pg. 10 ln. 16-20: reference for the approach? Also, the oxygenation of the sediment column effects the depth/rate of bioturbation! Is this not represented in the model? I.e. is bioturbation possible even in the sulfidic zone? If you just fit your model to a specific site you probably change this manually but what are you doing when coupled to a 3D ocean model?

There is no reference for the approach, it is just based on the assumptions that (a) bioturbation decreases with depth and (b) there is no bioturbation below a certain maximum depth. In the present model we have no "switching-off" of bioturbation in the sulfidic zone. This partly reflects the fact that certain bioturbators may generate local oxic zones inside a sulphidic environment, e.g. by ventilating their burrows, and we do not account for this spatial heterogeneity, but partly it is just an unrealistic simplification. An extension of the model where the vertical transport of solids is influenced by biogeochemistry would certainly be a desirable extension of the model.

We add: "The present formulation of the model has no explicit dependence of bioturbation depth on the availability of oxidants, i.e. bioturbation will take place in oxic as well as in sulphidic environments; adding this dependence should be essential if the model shall be applied to sulphidic areas."

pg. 11 ln. 6-9: Give equation for the exchange flux. Again the 3mm here...

We give this equation. We add: "In reality, the diffusive boundary layer thickness is on the order of 1 mm at low bottom shear situations and becomes even shallower if the bottom shear increases (e.g. Jorgensen and Des Marais, 1990). We choose a larger value because we need to account for the transport through the fluff layer as well. A future model version might include a dependence of this parameter on the bottom shear stress."

pg. 11 ln. 22: replace "sediments" with settles or is deposited. I assume the sediment accumulation is taken as advective transport in the diagenetic model!? Clarify in the text.

We do the replacement by "is deposited". We replace "as a downward movement" by "as a downward advection" in the following sentence to clarify.

pg.12 ln. 6-11: why transport away/towards different sites? explain/justify

We replace "movement" in the explanation above by "advection of fluff layer material" to clarify the explanation.

pg. 12 Biogeochemical processes in the water column: add references for previously published ERGOM version you mean (ln. 16) and the equations for the processes described in the following subsections (2.4.1 – 2.4.5) with tables of parameters and their values as used in the model equations (see general comments).

As described on top of this author's response document, it is by purpose that we do not include all equations and constants in the main text of the manuscript. We did not change anything in the water column BGC processes compared to the previous model version, so we do not give their quantitative description in our main text. We insist on keeping the details separate in the supplementary material.

We give references for previous ERGOM versions and make clear to which one we refer (Neumann et al. 2017) as our "ancestor model version".

pg. 13 ln. 18: rewrite to "from previous ERGOM versions". I find the use of the term detritus here confusing as you use organic carbon/material in the rest of your manuscript - Particulate organic carbon (POC) might be a better choice

The term "detritus" is used as this in all previous ERGOM model descriptions, so we use it here as well. "Particulate organic carbon" neglects the fact that our "detritus" contains N and P as well. "Particulate organic material" is also inaccurate as this would include living organisms, while detritus is the dead component only. This is why we use the footnote to exactly describe what we mean.

No changes to manuscript.

pg. 13 ln. 19-23 / 24-26: The decay rate constants and the partitioning into reactivity-classes are probably the most important steps for the model output (e.g. Arndt et al., 2013, Hülse et al., 2018). Therefore, this deserves some more words and justification.

The choice of classes is just designed to match the Middelburg model.

We give a justification of the chosen values in a new appendix section, where we compare the Middelburg decay rate over time with ours.

pg. 13 decay rate constants: where do the 0-degree values come from? - are they representative for your study area? - how do you know? - and how is it temperature dependent? Give equation!

The Middelburg equation does not include a temperature dependence, and the geographic locations at which their measurements were made include a large temperature range. Since we know that microbial decomposition is temperature dependent, we have to choose a baseline temperature, and the 0°C choice is indeed somewhat arbitrary. Luckily the model is not very sensitive to this choice, as a higher baseline temperature, meaning a lower decomposition rate of each class, would be compensated for by a shift in the class composition, leaving higher concentrations of quickly-degradable detritus classes which means an overall very similar total decomposition rate.

We give references (Thamdrup et al. 1998, Sawicka et al. 2012) for the temperature dependence and include these sentences: "The 0°C choice is somewhat arbitrary. Luckily the model is not very sensitive to this choice, as a higher baseline temperature, meaning a lower decomposition rate of each class, would be compensated for by a shift in the class composition, leaving higher concentrations of quickly-degradable detritus classes which means an overall very similar total decomposition rate."

pg. 13 Partitioning: Is the Middelburg approach not for OM at the sediment-water interface? Could you please clarify and give the equation used to calculate the fractions. Also what is the fraction of the non-decaying detrital?

Middelburg 1989 actually includes a graph showing the dependency of sulphate reduction rates with depth up to 1 m to support the exponential model.

We give details on the calculation in the new appendix section. We add a table giving decay rates and the corresponding mass fractions, including that of the non-decaying one of 18%.

pg.15 ln. 25-26: Give equation for conversion from SO4 to H2S depending on the diffusive CH4 flux from below the model domain

We give this stoichiometric equation together with all others in a table.

pg. 16 ln. 1: you DO favour the latter theory because your rate constants are independent of the TEA. Or am I wrong? / pg. 16 ln. 1-2: You say:"...we chose to adopt the decay rates proposed by Middelburg (1989), which may implicitly take the effect of the oxidant into account." I do not understand this statement. If I remember it right, the Middelburg (1989) rate model is the same for oxic and anoxic conditions.

Middelburg states that material which is decomposed later will be decomposed slower. This may be because the material itself is different, or because the oxidant is different. The Middelburg model includes both effects. Now in a mechanistic model we might want to separate the effects, but that's tricky because there is this controversial discussion. So what we do assume if we just apply the Middelburg model is that the time which a particle spends in the oxic zone, in the anoxic zone, in the sulphidic zone is similar in our setting and in Middelburg's experiments. In this case, the Middelburg model will include the correct slowing-down of degradation caused by the less efficient oxidant. If, in

contrast, our particle enters the sulphidic zone very quickly, the Middelburg model might predict a faster degradation since it anticipates the particle might still be in an oxic environment.

We add a footnote after "implicitly takes the effect of the oxidant into account": "Middelburg states that material which is decomposed later will be decomposed slower. This may be because the material itself is different, or because the oxidant is different. The Middelburg model includes both effects, and splitting them in a mechanistic model would mean preferring one theory or the other. So what we do assume if we just apply the Middelburg model is that the time which a particle spends in the oxic zone, in the anoxic zone, in the sulphidic zone is similar in our setting and in Middelburg's experiments. In this case, the Middelburg model will include the correct slowing-down of degradation caused by the less efficient oxidant."

Also what are the values of your degradation rate constants? They are the most important parameters in the diagenetic model. List them e.g. in a table, together with the rate constants for the water column and the fractions for the OM partitioning.

We add the requested table.

pg. 16 ln. 3: change "their study". It is just one author.

We change it.

pg. 16 ln. 4-7 preferential release of P: under which conditions is P preferentially released? Is this important for your study area? As you say on page 13 ln. 16: "anoxic conditions which, however, do not occur in our study area."

The study of Jilbert et al. 2011 states that this is the case in anoxic conditions, so the corresponding factor is applied in the absence of oxygen only in Reed et al. and in our model. Our model description is wrong here in stating that the factor is constant, in fact it is redox dependent, thank you for noticing this! Anoxic conditions occur inside the sediments in our region of interest, not at the sediment surface.

We change the sub-sentence to "as well as a constant factor_pref_remin_p which describes a redox-dependent ratio between the mineralisation speeds of OP and organic carbon and nitrogen. This factor is set equal to 1 under oxic conditions and greater than 1 under anoxic conditions."

pg. 16 t_detp_n :refer to table A1. How do you get the t_detp_n and t_det_n numbers for H? I understand that detritus is 50% enriched in C and P and how the values for C, N, O, P are calculated. But H does not add up. Should H for t_det_n & t_sed_n not be 22.875?; pref_remin_p: what's the value and where does it come from?

You are right, the value given here for H is wrong, in fact it is 22.875, as t_det_? is $(CH_2O)_{1.5*106/16}(NH_3)_1$ and t_detp_n is $(H_3PO_4)_{1.5*1/16}$. factor_pref_remin_p equals to 10, which is the geometric average of the values used in Reed et al. 2011, but we do not give values in the main text.

We change the value for H in Table A1 and add a reference to it.

pg. 16 2.5.2 Specific mineralisation processes: Add table for reaction network as mentioned in general comments

We add the desired table.

pg. 16 ln. 17-20: The description is very vage - list the station specific content e.g. in table 2 and what is the "small amount of reducible iron"? Quantify!

We add the station-specific content to Table 2 and add the number of 0.1 mass-% to the text.

pg. 16 ln. 24-25: reference for statement

We add the following reference: e.g. Sunagawa, Ichiro. 1994. „Nucleation, Growth And Dissolution Of Crystals During Sedimentogenesis and Diagenesis". In: Developments in Sedimentology, K. H. Wolf and G. V. Chilingarian, 51:19–47. Diagenesis, IV. Elsevier. https://doi.org/10.1016/S0070-4571(08)70435-7.

pg. 17 2.5.4 Pyrite formation: you did some model development here: add equations to make in more clear

We describe our additions in detail here, giving a quantitative mathematical description.

pg. 18 ln. 8-9: state the formula

We give the formula.

pg. 18 2.5.6 Reoxidation of reduced substances: There is a lot of information about the reaction network here. A summary of all that in a table would be very helpful! Also just use either iron-II/III or Fe-II/III, don't mix them up

We add a table of the reaction network. We replace iron-II and iron-III by Fe-II and Fe-III throughout the manuscript.

pg. 19 2.6 Carbon cycle: this is also a new model development – at least include the equations you use to calculate pH and pCO2.

We give the formulas we use for the iterative calculation process.

pg. 20 ln. 4 mode splitting method: give some quick background what this is and a reference

We combine the two sentences: "The equations which determine the temporal evolution of the state variables are solved by a mode splitting method, i.e. concentration changes due to physical and biogeochemical processes are applied alternately in separate sub-timesteps. For a discussion of this method and alternatives we refer to Butenschön et al. (2012)"

pg. 21 ln. 7: style. Change to: ... in the Southern Baltic Sea (see Fig. 1, we always present the stations from west to east).

We change this following your suggestion.

pg. 21 ln. 9: units of salinity of 20 is missing

We add g/kg as unit.

pg. 21 ln. 21: replace "interface" with interfaces / pg. 22 ln. 14: replace "So" with thus / pg. 22 ln. 22: replace "sampled" with samples

We do the requested replacements.

pg. 23 4.2 Initial and boundary conditions: ERGOM model: more information needed for the unpublished model run as stated in general comments Is the relaxation approach of DIN and DIP adopted from somewhere? Does this lead to realistic results?

No, this approach is not adopted. It by definition leads to realistic results for the wintertime nutrient concentrations, which mostly determine the export production of the surface layer.

We add more information on the 3-d model run which produced the data.

pg. 24+25 4.3.2 Optimisation strategies: Why do you talk about the application of the AHR-ES algorithm which is in the end not used at all? Delete this part.

We delete it and just mention that we used it and were not successful, since we believe this may be an important piece of information for others.

pg. 25 ln. 19 - end: What parameters are changed? What is the range they are varied in (add table to appendix)? Why don't you show any results of the optimisation? Are the final parameter values realistic, e.g. compared with other models or data? What are the most important parameters?

We add a table of original and changed parameters to a new appendix section which also includes a discussion of model sensitivity to the different parameter choices.

pg. 26 4.4 Manual correction of sand and silt station: How exactly did you modify the parameters? This is needed for reproducibility of your results!

We add: "This modification meant raising bioturbation and bioirrigation intensity by a factor of 10 at each station. Afterwards we reduced the parameter r_fluffy_moveaway which describes the rate at which fluff layer material is transported to the deeper areas until realistic concentrations in the pore water profiles were reached."

pg. 26 ln. 6: which detritus is meant here (just POC or POC with mineral particles)? - rephrase "out of the sediments"

We rephrase "by keeping detritus out of the sediments" to "by an unrealistically low incorporation of reactive particulate material into the sediments".

pg. 26 - 5.1.1 Pore water profiles at mud stations: It should be easy to check with the data if the variability at site AB is because they are 23km apart.

We use different symbols for the different locations in the plot now.

pg. 30 ln. 1 phosphate is in the right panel

We correct this to "the profiles of ammonium and phosphate (middle and right panel in Fig. 6)"

pg. 31. ln. 3-4: reference for this statement

We add the following reference: Meysman, Filip J.R., Volodymyr S. Malyuga, Bernard P. Boudreau, and Jack J. Middelburg. 2008. „A Generalized Stochastic Approach to Particle Dispersal in Soils and Sediments". Geochimica et Cosmochimica Acta 72 (14): 3460–78. https://doi.org/10.1016/j.gca.2008.04.023.

pg. 31 ln. 6: rephrase

Rephrased to "In Figure 8a, we compare measured bioturbation diffusivities DB to those used in the model."

pg. 33 Conclusion: ignoring the N-cycle because it's not part of the SECOS project: This is a rather poor justification, especially as you say later: "... where the efflux of nutrients from the sediment strongly influences water column biogeochemistry, like in our study area." and "... denitrification [...] may strongly influence marine ecosystems".

Within the SECOS project, no measurements of e.g. nitrification or denitrification rates have been performed. This prevents us from a comparison of the model performance regarding the N cycle with direct measurements.

We rephrase the sentence to "For example, the nitrogen cycle was not compared to observations, which is due to the fact that the project SECOS in which this work was done did not focus on it and so the required observations of nitrification or denitrification rates were missing."

pg. 37 Table A1: stoichiometry of t_h2s is wrong: it should have 2*H and 1*S

We correct this error in the manuscript, thank you for noticing!

**Reviewer 3 – general comments**

* In the introduction and conclusion, the emphasis while presenting the ERGOMSED model is put on the online coupling between the Benthic and Pelagic part. However this online coupling is not valorised in the results and discussion section. This should be enhanced. Neither the pelagic part nor the solutes exchanges between the benthic and pelagic part are mentioned in the results, although the conclusion states that "In the long term, biogeochemical ocean models should aim at a process-resolving description of surface sediments. This is especially true for shallow ocean areas where the efflux of nutrients from the sediment strongly influences water column biogeochemistry, like in our study area." In the case that benthic fluxes, for any reason, are not available within SECOS (which would be surprising), ranges from the litterature could be used for comparison, and it would also be relevant to compare benthic fluxes to the lateral fluxes (inferred from the nudging procedure for the pelagic nutrients) to stress the relevance of such coupled framework.

We do include a comparison to measured exemplified bentho-pelagic fluxes now as an additional model validation.

* The fluff layer approach is an interesting feature of the coupling set-up, and a practical solution to handle solids lateral transport and exchanges between benthic and pelagic part. To my knowledge, the use of a fluff layer is not systematic in B-P coupled models and I would have found relevant to enforce introduction and discussion on this aspect.

We put more emphasis on this coupling approach now by mentioning its novelty and adding a more detailed mathematical description of the coupling.

* Appendix B supposedly justifies the inclusion of enhanced dynamics in the benthic model. This should be more developed. In particular: 1) Has the same calibration procedure been applied "from scratch" after having switched off those processes;

No, it has not, since the calibration procedure was very time consuming. So the comparison is somewhat unfair in this context.

We mention this in Appendix B now.

2) Some of the "reduced" experiments actually seems to behave better than the reference simulation. Can the authors justify their modification in this context ?

Both the model without correction for the diffusion of alkalinity and the model without adsorption to clay minerals gives higher concentrations of Fe-II in the pore water, which is in this case closer to observations. However, for both processes we know they exist, so leaving them out would mean a contradiction to our approach of a mechanistic representation. This approach is now pointed out stronger in the introduction.

**Reviewer 3 – specific comments**

* P3L13 : I suggest to add a references on ecosystem services ( for instance : https://cices.eu/content/uploads/sites/8/2012/07/CICES-V43_Revised-Final_Report_29012013.pdf)

We add the suggested reference.

* From P9L17-19 and P10L10-15 I had understood that solid compounds were only transferred from the distinct fluff and upper sediment layers through bioturbation (which includes here also other mixing effects). However, at P11L29 we learn that accumulation (advection) also induces a transfer from the fluff to the sediments. This may be introduced earlier (eg. end of Sect. 2.3.3) and explained in more details.

We give a more thorough mathematical description of the vertical transport processes now, which includes the representation of sediment growth as a downward advection.

* Tab 1. : Benthic tracers (both solids and dissolved) are defined in mol/m2 which doesn't correspond with the definitions given in P10L6 and P10L25. Is there a general transfomation applied to get those in units of mol/volume of liquids/solids ?

Yes there is, and the conversion is absolutely straightforward.

We give this transformation in the framework of the mathematical description of the vertical transport processes now.

* P10L10-20 : Bioturbation rate are defined for the sediment compartment. Is the uppermost value used for diffusion between the fluff layer and the uppormost sediment cell ? please precise.

Yes it is.

In our new formulation we are more precise and explicitly describe the flux between the compartments.

* Sect 2.4: Interactions between phosphate and iron aren't described for the water column. Are t_ipw, t_ihw and t_mow only included to enable a lateral transport of resuspended solids, or is there possible biogeochemical transformation in the water column ?

Indeed oxidation of Fe-II and Mn-II, reduction of Fe-III and Mn-IV, and adsorption of phosphate to iron oxyhydroxides may also happen in the water column.

We added the corresponding sections to the description of water column processes.

*P24L3 : In general, it might be relevant to comment which parameters were considered for the calibration and which were adapted, which were considered as equals for all stations and which were considered to differ between stations.

We add a table stating this to a new appendix section which also describes the sensitivity of the model to a variation in the parameters.

* P24L15: It is not clear whether $\Delta_i$ is defined specific only to each variables, or specific also to each station, or also to each sampling depth. This is relevant as refferred to when discussing model performances.

It is neither individual to a station nor to a sampling depth.

We add a footnote stating this.

\* P26L3 : Should the first "bioirrigation" be replaced by "bioturbation" ?

Yes indeed. Thank you for noticing!

We replace „estimated bioirrigation rates" by "bioturbation rates"

\* Sect 5.12 : As is true for numerous model of this type, application in sandy sediments might be limited to the the lack of consideration of processes specific to permeable sediments. "Whashout" is mentioned in Sect 2.3.3, but this isn't the only aspect of it. This should be discussed. For instance in this section. You might refer to the review from Huettel et al, 2013.

Thank you for this suggestion!

We add some discussion on the specifics of permeable sediments and refer to the suggested review article.

\* P30L2-3 : Switch "higher" and "lower".

Corrected, thanks!

\* P30L19 : I don't see a TOC maximum at the top of sediments for station DS.

That's because the measurements we show in the background of the DS graph are mistakenly duplicated from the ST station.

We correct this in the figure.

Concerning this last paragraph, you could maybe consider the fact that the inability of the model to provide a TOC profile increasing with depth is related to the absence of dynamics specific to permeable sediments , washout in particular ? I insist on this point since it represents a major challenge for BP coupling intended to be implemented on shelves with mixed sand/mud conditions. I don't ask that this be solved in this study, but the issue should be commented.

Very good point, we add some discussion on this. But this need not necessarily be washout, also nonlocal transport of fluff material into higher depths by bioturbating organisms might explain TOC profiles increasing with depth.

\* Table A1 : t_h2s has one H and 2 S ? Is that an error in table or in the model ?

Corrected, thank you for noticing!